# Near-Unity Nitrate to Ammonia conversion via reactant enrichment at the solid-liquid interface

Wanru Liao[1,2,9], Jun Wang[1,2,9], Yao Tan[1], Xin Zi[1], Changxu Liu[3], Qiyou Wang[1], Li Zhu[4], Cheng-Wei Kao[5], Ting-Shan Chan[5], Hongmei Li[1], Yali Zhang[6], Kang Liu[1], Chao Cai[1], Junwei Fu[1], Beidou Xi[7]✉, Emiliano Cortés[4]✉, Liyuan Chai[8] & Min Liu[1,8]✉

Electroreduction of nitrate ($NO_3^-$) to ammonia ($NH_3$) is a promising approach for addressing energy challenges. However, the activity is limited by $NO_3^-$ mass transfer, particularly at reduction potential, where an abundance of electrons on the cathode surface repels $NO_3^-$ from the inner Helmholtz plane (IHP). This constraint becomes pronounced as $NO_3^-$ concentration decreases, impeding practical applications in the conversion of $NO_3^-$-to-$NH_3$. Herein, we propose a generic strategy of catalyst bandstructure engineering for the enrichment of negatively charged ions through solid-liquid (S-L) junction-mediated charge rearrangement within IHP. Specifically, during $NO_3^-$ reduction, the formation of S-L junction induces hole transfer from Ag-doped $MoS_2$ (Ag-$MoS_2$) to electrode/electrolyte interface, triggering abundant positive charges on the IHP to attract $NO_3^-$. Thus, Ag-$MoS_2$ exhibits a ~28.6-fold $NO_3^-$ concentration in the IHP than the counterpart without junction, and achieves near-100% $NH_3$ Faradaic efficiency with an $NH_3$ yield rate of ~20 mg h$^{-1}$ cm$^{-2}$ under ultralow $NO_3^-$ concentrations.

Ammonia ($NH_3$), one of the most common industrial chemicals, is crucial for the production of agricultural fertilizers and holds immense potential as a green hydrogen-rich fuel[1–3]. The current global ammonia demand exceeds 180 million tons annually, primarily fulfilled through industrial-scale production of energy-intensive Haber-Bosch (H-B) process[1,4]. Within this process, steam-reformed hydrogen ($H_2$) undergoes reaction with nitrogen ($N_2$) under elevated temperature (~500 °C) and pressure (>100 atm)[5,6], which not only accounts for approximately 1.4% of global carbon dioxide ($CO_2$) emissions, but also necessitates the consumption of 2% of the world's total energy supply[7,8]. Recently, electrocatalytic methodologies have surfaced as a viable clean energy pathway for decentralized ammonia synthesis at room temperature, accommodating a range of infrastructure scales and potentially powered by locally sourced renewable energy sources[9,10]. Despite the substantial global demand that may sustain the traditional ammonia production route in the near future, the electrochemical ammonia synthesis can be promising complementary process to the Haber−Bosch technology for contributing to decarbonizing ammonia production[11,12].

[1]Hunan Joint International Research Center for Carbon Dioxide Resource Utilization, State Key Laboratory of Powder Metallurgy, School of Physics, Central South University, Changsha 410083 Hunan, P. R. China. [2]School of Chemistry and Pharmaceutical Engineering, Changsha University of Science and Technology, Changsha, Hunan, P. R. China. [3]Centre for Metamaterial Research & Innovation, Department of Engineering, University of Exeter, Exeter, UK. [4]Nanoinstitut München, Fakultät für Physik, Ludwig-Maximilians-Universität München, München, Germany. [5]National Synchrotron Radiation Research Center, Hsinchu, Taiwan. [6]Key Laboratory of Land Surface Pattern and Simulation, Institute of Geographic Sciences and Natural Resources Research, Chinese Academy of Sciences, Beijing, P. R. China. [7]State Key Laboratory of Environmental Criteria and Risk Assessment, Chinese Research Academy of Environmental Sciences, Beijing, P. R. China. [8]School of Metallurgy and Environment, Central South University, Changsha, P. R. China. [9]These authors contributed equally: Wanru Liao, Jun Wang. ✉e-mail: xibd@craes.org.cn; Emiliano.Cortes@lmu.de; minliu@csu.edu.cn

Recently, diverse electrochemical approaches have been explored to address the varied demands for NH$_3$ production in the future energy landscape, encompassing the electrochemical N$_2$ reduction reaction (NRR)[13–15], lithium-mediated NRR[16–19] and nitrate reduction reaction (NO$_3$RR)[20–22]. Among them, the present NH$_3$ yield via NO$_3$RR surpasses those of NRR by two to three orders of magnitude, mainly due to the relatively lower dissociation energy of the N=O bond (204 kJ mol$^{-1}$) of nitrate anion (NO$_3^-$) compared to the N≡N bond (941 kJ mol$^{-1}$)[23,24]. Besides, NO$_3^-$ exhibits a ubiquitous presence within contaminated groundwater and industrial effluent[25,26], with their availability further augmented through the oxidation process of atmospheric N$_2$ as the source of nitrogen[9,27,28]. Consequently, the NO$_3$RR pathway has emerged as one of the most potential in renewable NH$_3$ production.

While continuous progress has been made in NO$_3$RR under high NO$_3^-$ concentrations (>100 mM)[11,29–31], achieving large throughput and effective NO$_3^-$ reduction in groundwater with concentrations below 10 mM remains a challenge, presenting a limited NH$_3$ yield rate below 5 mg h$^{-1}$ cm$^{-2}$ and a Faradaic efficiency (FE) below 85%[22,32–34]. On the other hand, a considerable portion of the groundwater, stemming from both agricultural and industrial sources[35–40], has a diluted concentration, necessitating a platform suitable for operation at low concentration. More importantly, the removal process of NO$_3^-$ continuously reduces its concentration, inevitably reaching the low concentration regime with reduced performance.

To mitigate the degradation in low concentration, significant efforts have been devoted to concentrating NO$_3^-$ locally around electrodes, employing various strategies such as porous carbon framework encapsulation[32], built-in electric field regulation[41], and nitrogen-vacancy engineering[42]. However, these endeavors primarily focus on manipulating catalyst properties for NO$_3^-$ enrichment, with limited exploration of interfacial NO$_3^-$ transfer under operational conditions. Particularly, at the reduction potential, an excess of electrons accumulates on the cathode surface, leading to a repulsion of negatively charged NO$_3^-$ ions from the inner Helmholtz plane (IHP), where the catalytic reactions occur[27]. This ubiquitous effect of charge repulsion hinders the mass transfer and sets up an intrinsic bottleneck for the catalytic process.

Here, we overcome the fundamental limitation through utilizing a solid-liquid (S-L) junction[43,44] to manipulate the charge distribution within the IHP region. By employing proper bandstructure engineering of the catalyst (p-type Ag-doped MoS$_2$, Ag-MoS$_2$), the enhanced interfacial charge transfer endows reinforced S-L junction to facilitate the NO$_3^-$ enrichment in IHP (Fig. 1), showcasing a ~ 27.6-fold increase in NO$_3^-$ concentration compared to the counterpart without junction. As a proof of concept, the Ag-MoS$_2$ platform demonstrates impressive performance, including a record-high NH$_3$ yield rate of ~ 20 mg h$^{-1}$ cm$^{-2}$ and NH$_3$ FE of ~ 100%, outperforming previous results obtained under low NO$_3^-$ levels[22,32,34,36]. Furthermore, the high NO$_3^-$-to-NH$_3$ conversion efficiency (99.3%) enables the removal of nitrate from 100 mM to low levels of 0.54 mM within 3 h, in sharp contrast with the reference (still relatively high level under 48 h operation). Notably, we successfully convert NO$_3^-$ into high-purity NH$_4$Cl products with near-unity efficiency by coupling the NO$_3$RR with an air stripping process, demonstrating the potential of large throughput ammonia production in a sustainable way.

## Results

Based on the merits of appropriate band gap (1.2–1.9 eV), excellent conductivity, adjustable Fermi level position, and two-dimensional characteristics of large specific surface area, MoS$_2$ was selected as the platform for the electrocatalysis, with intentionally doped Ag as the enabler for the desired S-L junction formation. The Ag-doped MoS$_2$ (Ag-MoS$_2$) catalyst was grown in situ on a carbon cloth using a one-step hydrothermal method (Supplementary Fig. 1, see "Methods"). MoS$_2$ as a reference was synthesized through a similar route, excluding the addition of a silver source. Scanning electron microscopy (SEM), transmission electron microscopy (TEM), and high-resolution TEM (HRTEM) revealed a nanoflower morphology of Ag-MoS$_2$ with (100) crystal plane orientation (Fig. 2a–c and Supplementary Fig. 2), maintaining a morphology similar to that of MoS$_2$ (Supplementary Figs. 3, 4). Energy-dispersive X-ray (EDX) elemental mapping illustrated the uniform distribution of Ag species on MoS$_2$ nanoflowers (Fig. 2d). The X-ray diffraction (XRD) patterns of Ag-MoS$_2$ and MoS$_2$ were similar, indicating no formation of secondary phases after Ag doping (Fig. 2e). Inductively coupled plasma optical emission spectrometry (ICP-OES) determined the optimal Ag content in Ag-MoS$_2$ to be 3.49 wt% (Supplementary Table 1).

To analyze the surface chemical states of the catalysts, high-resolution X-ray photoelectron spectroscopy (XPS) was performed

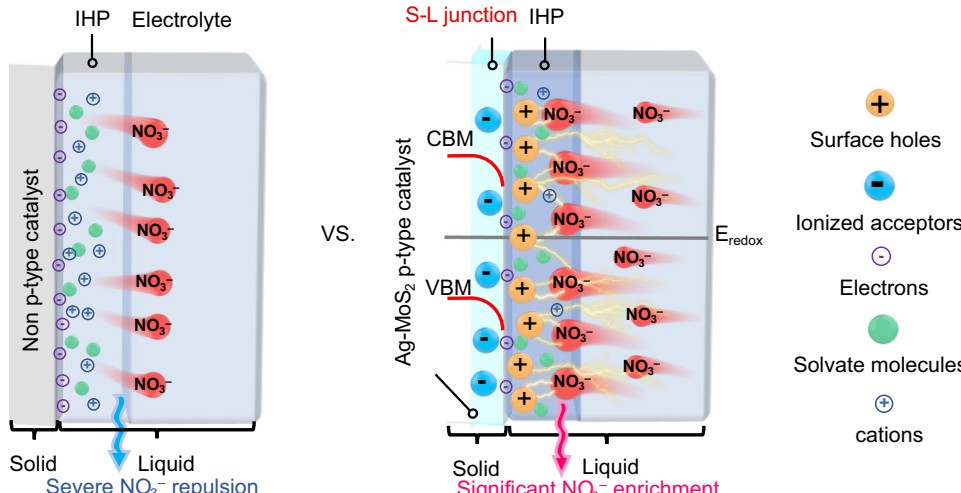

**Fig. 1 | Solid-liquid (S-L) junction-mediated charge rearrangement to attract NO$_3^-$.** Schematic diagram of S-L junction-mediated NO$_3^-$ enrichment within the IHP region over Ag-MoS$_2$ at the reduction potential. $E_{redox}$ represents the theoretical redox potential of NO$_3^-$/NH$_3$. CBM and VBM express the conduction band minimum and valence band maximum of Ag-MoS$_2$, respectively. Note: IHP contains surface holes, solvate molecules, and NO$_3^-$ anions and cations. Electrons accumulate on the electrode surface under negative polarization. Ionized acceptors exist in the S-L junction (see detailed discussions in the supplementary discussions).

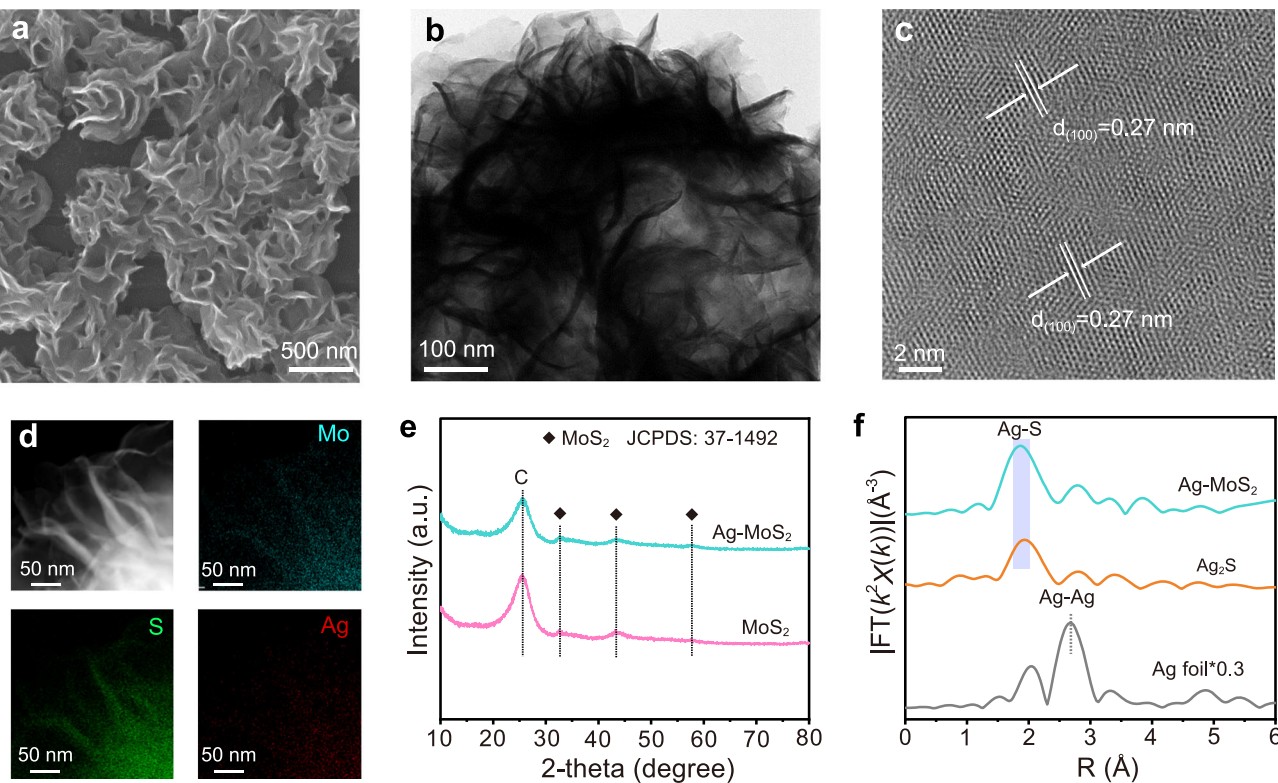

**Fig. 2 | Synthesis and structural characterizations of Ag-MoS₂. a** SEM, (**b**) TEM, (**c**) HRTEM, and (**d**) EDX mapping of Ag-MoS₂. **e** XRD patterns of Ag-MoS₂ and MoS₂. **f** Fourier transformed $k^2$-weighted EXAFS spectra of Ag-MoS₂ and reference samples.

(Supplementary Fig. 5). The Mo 3$d$ and S 2$p$ XPS spectra indicated the presence of $Mo^{4+}$ and $S^{2-}$ in both Ag-MoS₂ and MoS₂[45,46]. In the Ag 3$d$ spectra, characteristic peaks at 368.16 and 374.13 eV were assigned to the Ag-S bond[47,48]. To further elucidate the coordination structure of Ag species, X-ray absorption fine structure (XAFS) was investigated. The Mo $K$-edge X-ray absorption near edge structure (XANES) spectra exhibited a higher pre-edge absorption energy of Ag-MoS₂ than those of Mo foil and MoS₂ (Supplementary Fig. 6a), implying that the doping Ag species induced decreased electron density on the Mo site. From the Ag $K$-edge XANES spectra, the absorption energy of Ag-MoS₂ located between those of Ag foil and Ag₂S references (Supplementary Fig. 6b), indicating the valence state of the doped Ag within 0 to 1. The Fourier-transformed extended XAFS (EXAFS) of Ag-MoS₂ exhibited characteristic peaks of the Ag-S bond at 1.9 Å, confirming the coordination of Ag with S atoms (Fig. 2f). These comprehensive results provide clear evidence of the synthesis of Ag-doped MoS₂ catalysts.

To investigate the construction and regulation of the solid-liquid (S-L) junction, we conducted ultraviolet photoelectron spectra (UPS) and Mott-Schottky (M-S) measurements. The UPS results (Supplementary Fig. 7) demonstrated a significant increase in the work function of Ag-MoS₂ (5.47 eV) compared to MoS₂ (4.98 eV), indicating a downward-shifted Fermi level after Ag doping. In addition to the UPS method, KPFM measurements were carried out to collaboratively demonstrative the increase of work function by Ag doping (Supplementary Figs. 8, 9)[49]. The surface potential difference between Ag-MoS₂ and FTO substrate was ~ 230 mV, while 20 mV for MoS₂. Correspondingly, the work functions of Ag-MoS₂ and MoS₂ were 5.13 and 4.92 eV, respectively. The variation trend was in consistent with UPS results, indicating a relatively high work function of Ag-MoS₂. M-S plots further revealed the Fermi level potential ($E_F$) of Ag-MoS₂ (1.07 V $versus$ RHE) and MoS₂ (0.53 V $versus$ RHE), respectively (Fig. 3a), surpassing the theoretical redox potential of $NO_3^-/NH_3$ ($E_{redox}$, 0.27 V $versus$ RHE under neutral conditions (pH 6.71))[50-53]. The calculated carrier concentration of Ag-MoS₂ is $1.14 \times 10^{19}$ cm⁻³, 1.8 times higher than that of

MoS₂ ($6.3 \times 10^{18}$ cm⁻³, Supplementary Table 2). Thus, the Ag dopant as the electron acceptor to increase the carrier concentration and work function of MoS₂ was verified.

Upon contact with the $NO_3^-$-containing electrolyte, the potential difference between the $E_F$ and the $E_{redox}$ ($E_F$-$E_{redox}$) triggered charge transfer between the catalyst surface and the inner Helmholtz plane (IHP), forming positively charged IHP on the electrolyte side (Supplementary Fig. 10). Specifically, p-type MoS₂ catalysts exhibited the characteristic of hole conduction. Under the drive of $E_F$-$E_{redox}$, the holes (majority carrier) in the valance band would transfer from the semiconductor surface to the electrolyte, inducing charge rearrangement within the IHP region (Supplementary Fig. 10a)[54]. Meanwhile, the high carrier concentration ($1.14 \times 10^{19}$ cm⁻³) and thin space charge layer width (5.88 nm) of Ag-MoS₂ can contribute to the increase of probability for hole tunneling effect (Supplementary Fig. 10b and Supplementary Tables 2, 3), which benefited positive charges distribution within IHP. Compared to MoS₂ with a limited $E_F$-$E_{redox}$ (0.23 V), the larger $E_F$-$E_{redox}$ value of Ag-MoS₂ (0.76 V) provoked 0.53 V more downward band bending, intensifying the built-in electric fields at S-L junctions (see detailed quantitative methods and discussion in Supplementary Figs. 11–13).

To explore the influence of S-L junction on the charge distribution within the IHP region, alternating current voltammetry (ACV) measurements were employed (Fig. 3b)[55,56]. ACV was a non-destructive technique which had been widely utilized to record the adsorption behavior change in the IHP range[57,58]. Compared to MoS₂, Ag-MoS₂ exhibited a negatively shifted potential of zero charges (PZC, 0.57 V $versus$ RHE). PZC was adopted as an indicator for the IHP structure evolution, and the negative PZC indicated the adsorption of anion on the IHP region of Ag-MoS₂ (Fig. 4 and Supplementary Figs. 14, 15)[59,60]. To investigate the charge distribution under reduction potential, surface capacitance (charge distribution within the solid-liquid interface) of the catalysts was obtained from Nyquist plots at − 0.6 V $versus$ RHE (Supplementary Fig. 16). The capacitance of Ag-MoS₂ (9.5 μC cm⁻²) was lower than that of

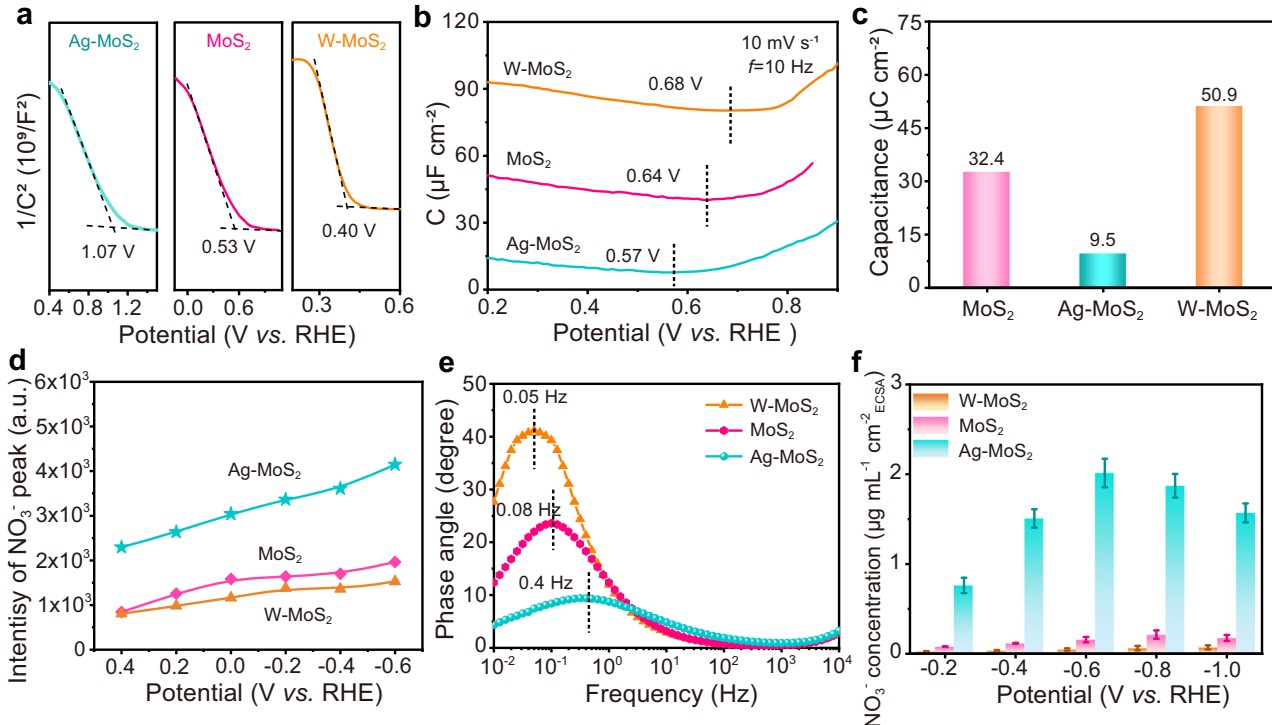

**Fig. 3 | S-L junction-mediated NO$_3^-$ enrichment within IHP region. a** Mott-Schottky plots of Ag-MoS$_2$, MoS$_2$, and W-MoS$_2$ catalysts. **b** The non-Faradaic capacitance-potential curves for the diverse catalysts. The potential of zero charge (PZC) describes the condition when the capacitance on a surface is minimal. **c** The fitted surface capacitance of the three catalysts. **d** The peak intensity of absorbed NO$_3^-$ from in situ Raman spectra of catalysts at various potentials. **e** Bode plots of catalysts at the reduction potential of −0.6 V versus RHE in 10 mM NO$_3^-$ electrolyte. **f** The ECSA normalized NO$_3^-$ adsorption capacity at different applied potentials on catalysts.

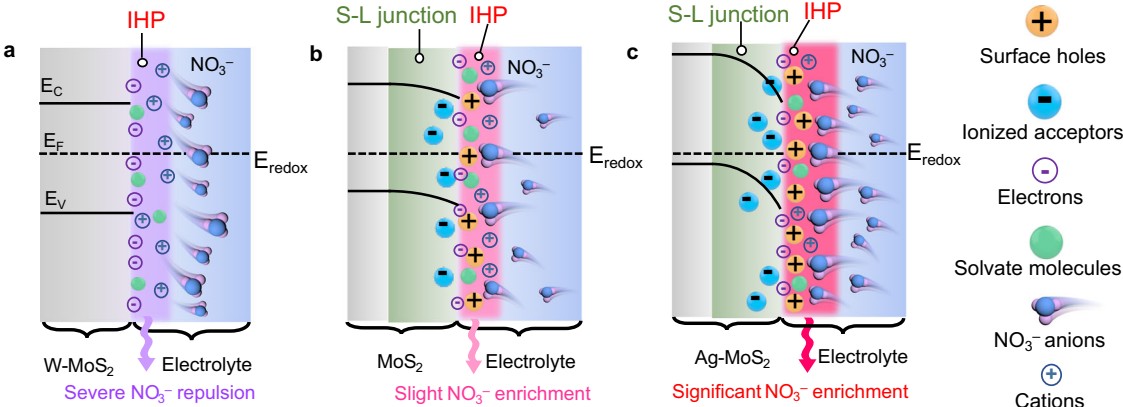

**Fig. 4 | NO$_3^-$ distribution on IHP region over MoS$_2$-based catalysts. a** W-MoS$_2$ without S-L junction, (**b**) MoS$_2$ with weak S-L junction, and (**c**) Ag-MoS$_2$ with strong S-L junction at the reduction potential.

MoS$_2$ (32.4 µC cm$^{-2}$), indicating significant positive charges distribution of Ag-MoS$_2$ (Fig. 3c). High valence metals had been reported to serve as electron donors to regulate the $E_F$ away from VBM, for enhancing the n-type characteristics of semiconductors[61]. Thus, we also prepared the reference samples with tungsten (W) doping, to demonstrate the effect of the S-L junction. In contrast, a n-type W-doped MoS$_2$ (W-MoS$_2$) counterpart with no obvious S-L junction (Fig. 3a and Supplementary Figs. 7 and 17–21) had higher PZC value (0.67 versus RHE, Fig. 3b) and capacitance (50.9 µC cm$^{-2}$, Fig. 3c), suggesting extensive anions packed on its IHP. These results confirmed that Ag doping can effectively enhance the built-in electric field at the S-L junction, which consequently induced abundant positive charges distributed on the IHP of Ag-MoS$_2$ at the reduction potential.

To investigate the S-L junction-mediated NO$_3^-$ enrichment effect, in situ Raman spectra of the three catalysts were carried out (Supplementary Fig. 22). Apart from the observed H$_2$O peak (at 1605 cm$^{-1}$)[62–64], the NO$_3^-$ peak (at 1046 cm$^{-1}$)[65,66] was detected at the potentials from open circuit potential (OCP) to −0.6 V versus RHE, indicating NO$_3^-$ adsorption near the catalyst surface. Notably, Ag-MoS$_2$ with a stronger NO$_3^-$ peak than those of MoS$_2$ and W-MoS$_2$ at each reduction potential (Fig. 3d) demonstrated a pronounced NO$_3^-$ enrichment effect. In Bode phase plots (Fig. 3e), compared with W-MoS$_2$ (0.05 Hz) and MoS$_2$ (0.08 Hz), the higher frequency of Ag-MoS$_2$ (0.4 Hz) implied a facilitated interfacial charge transfer process, which can be ascribed to its intensified NO$_3^-$ accumulation within the IHP region[67,68]. Elemental mapping displayed obvious N and O signals on Ag-MoS$_2$ nanoflowers

during the electrolysis of −0.6 V *versus* RHE in $NO_3^-$-contained electrolyte, whereas they were relatively shallow or inconspicuous on $MoS_2$ and W-$MoS_2$ (Supplementary Figs. 23–25), disclosing a large number of $NO_3^-$ migrated to the surface.

To quantify the $NO_3^-$ concentration, the $NO_3^-$ enrichment capacity of the three catalysts was measured by ion chromatography (IC) at applied potentials (Fig. 3f and Supplementary Fig. 26, see "Methods"[69]). By normalizing the $NO_3^-$ adsorption capacity with respect to the electrochemical active surface area (ECSA), the concentration of $NO_3^-$ adsorbed on Ag-$MoS_2$ presented a normal distribution trend as the potential ranging from − 0.2 to −1.0 V *versus* RHE (Fig. 3f and Supplementary Fig. 27), and reached the peak at − 0.6 V *versus* RHE (2.0 μg mL$^{-1}$ cm$^{-2}_{ECSA}$). Due to the different Fermi level position, the adsorbed $NO_3^-$ concentration of $MoS_2$ and W-$MoS_2$ peaked at more negative potentials of − 0.8 and −1.0 V *versus* RHE, respectively. As the Ag dopant acted as the electron acceptor to increase the carrier concentration (Supplementary Table 2), it induced an intensified surface band bending when the Ag-$MoS_2$ was contacted with the $NO_3^-$ contained electrolyte. Thus, the maximum adsorbed $NO_3^-$ concentration of Ag-$MoS_2$ surpassed $MoS_2$ and W-$MoS_2$ by factors of 8.8 and 27.6, respectively, at their optimal adsorption potentials. The $NO_3^-$ adsorption concentration was also normalized by specific surface area (SSA), this trend aligned with the ECSA normalization results (Supplementary Fig. 28), highlighting superior intrinsic $NO_3^-$ adsorption capability of Ag-$MoS_2$. The lowest surface capacitance at −0.6 V *versus* RHE suggested massive positive charges within IHP (Supplementary Fig. 29), contributing to the peaked $NO_3^-$ concentration at that potential. While the excessive potential (above −0.6 V) triggered severe downward shift of $E_F$ into the valence band of Ag-$MoS_2$ (Supplementary Fig. 30 and Supplementary Table 4, see details in the corresponding discussion), forming a degenerate semiconductor with metallic property, as proved by the decreased arc radius at high-frequency region. This could severely destroy the S-L junction to impact the enrichment of $NO_3^-$. We also implemented finite element method simulations for the $NO_3^-$ distribution on the three catalyst surfaces through COMSOL, which matched the experimental results demonstrated (Supplementary Fig. 31 and Supplementary Table 5).

Under the S-L junction-induced $NO_3^-$ enrichment feedback, we evaluated the $NO_3RR$ performance in a standard three-electrode system at ambient temperature and pressure. In this reaction system, $NH_4^+$, $NO_3^-$, and $NO_2^-$ were monitored and quantified by coloration and $^1H$ nuclear magnetic resonance (NMR) spectroscopy[70] (Supplementary Figs. 32–35). A typical industrial and agricultural groundwater-relevant $NO_3^-$ concentration of 10 mM was reasonably used in the electrolyte for the standard electrochemical tests[35,37,38,69]. Linear sweep voltammetry (Supplementary Fig. 36) curves of W-$MoS_2$, $MoS_2$, and Ag-$MoS_2$ all presented a higher current density between −0.5 and − 1.65 V *versus* RHE in the $NO_3^-$ electrolyte relative to $NO_3^-$-free solutions. Particularly, the most significant current difference occurred on Ag-$MoS_2$, revealing its promising $NO_3RR$ activity.

Following the chronoamperometry measurements at various applied potentials tested in a flow cell reactor (Supplementary Fig. 37), the $NO_3RR$ performance was assessed for the three catalysts with a 0.5 h duration. Of these, Ag-$MoS_2$ displayed the superior performance (Fig. 5a, b), with a near unit $NH_3$ Faradaic efficiency (FE) of 99.7% at − 0.6 V *versus* RHE (~ 200 mA cm$^{-2}$) and an optimal $NH_3$ yield rate of 20.1 mg h$^{-1}$ cm$^{-2}$ at −1.0 V *versus* RHE (~ 340 mA cm$^{-2}$). The optimal $NH_3$ yield rate value was about 3.4-folds and 1.7-folds than those of W-$MoS_2$ and $MoS_2$, respectively. Similarly, the peak $NH_3$ FE of Ag-$MoS_2$ was approximately 3.2-folds and 1.8-folds than those of W-$MoS_2$ and $MoS_2$, respectively. These results displayed the superior $NO_3RR$ performance of Ag-$MoS_2$ compared to its counterparts. Moreover, the performance surpassed most state-of-the-art $NH_3$ activity ever reported at a low $NO_3^-$ system (Fig. 5c and Supplementary Table 6)[22,32,34,71,72]. The electrochemically active surface area-normalized $NH_3$ yield rate also

clarified the excellent intrinsic activity of Ag-$MoS_2$ (Supplementary Figs. 27 and 38). The blank experiments without adding $NO_3^-$ to the electrolyte or working at OCP produced a negligible amount of $NH_3$ (Supplementary Fig. 39), excluding the possible interference on the quantification results. Besides, the possible contribution of Ag nanoparticles on the $NO_3^-$ enrichment and $NO_3RR$ performance of $MoS_2$ has been eliminated (Supplementary Figs. 40–43).

To reveal the significance of $NO_3^-$ enrichment for superior $NO_3RR$ performance under a low-$NO_3^-$ concentration system, EIS measurements were conducted (Fig. 5d and Supplementary Fig. 16). In the Nyquist plots, semicircles with similar radii at the high-frequency region of the three samples indicated nearly equivalent conductivity. At the low-frequency region, W-$MoS_2$ without S-L junction showed a straight line with a slope of 45°, suggesting that $NO_3^-$ mass transfer was the rate control step for $NO_3RR$. In contrast, the smaller arc radius over Ag-$MoS_2$ illustrated the enhanced reaction rate of $NO_3RR$ (from 16.41 to 6.92 Ω cm$^{-2}$, Supplementary Table 7). The lowest phase angle of Ag-$MoS_2$ in bode plots also testified its accelerated catalytic kinetics, ascribing to the distinct $NO_3^-$ enrichment effect (Fig. 3e). In addition, from the $NO_3RR$ performance at a wide range of $NO_3^-$ concentrations (Supplementary Fig. 44), the optimal Ag-$MoS_2$ in $K_2SO_4$ system (Supplementary Figs. 45–46) exhibited its broad adaptability and achieved an excellent $NH_3$ FE at − 0.6 V *versus* RHE (> 90%, in Supplementary Fig. 47), even at ultra-low concentration (1 mM). While the $NH_3$ yield rate of Ag-$MoS_2$ was significantly higher than those of W-$MoS_2$ and $MoS_2$ in low $NO_3^-$ concentration (Supplementary Fig. 44, insert), it gradually became comparable as the concentration increased. These results highlighted the importance of $NO_3^-$ enrichment to overcome the constraint of low-concentration $NO_3^-$ electroreduction to $NH_3$, via the S-L junction-mediated effect.

To examine the $NO_3^-$ removal capability of Ag-$MoS_2$, the conversion tests with an initial 100 mM $NO_3^-$ were carried out in an H-cell reactor, and the remaining products were measured. Impressively, the $NO_3RR$ on the three samples followed typical quasi-first-order kinetics, and the $NO_3^-$ to $NH_3$ selectivity of Ag-$MoS_2$ was up to 99.3% within only 3 h of electrolysis at − 0.6 V versus RHE ($NO_3^-$ concentration decreased sharply from 100 mM to 0.54 mM, Fig. 5e and Supplementary Fig. 48), manifesting that nearly all the N sources were converted into $NH_3$ (Supplementary Fig. 49). Comparatively, the $NO_3^-$ to $NH_3$ conversion efficiency of W-$MoS_2$ and $MoS_2$ were as low as ~16% and ~30%, and the residual $NO_3^-$ still remained a relatively high level of 4.64 mM and 11.78 mM even after 48 h (Supplementary Fig. 50). Also, Ag-$MoS_2$ behaved with stable performance and robust structure, as demonstrated by the steady eight-cycling $NO_3RR$ process at 25 °C (Fig. 5f), the negligible morphology change, tiny phase structure difference, and slight chemical valence state shift after the electrolysis (Supplementary Figs. 51, 52).

Furthermore, we demonstrated the practicality of the proposed setup for efficient $NH_3$ collection. Here, we integrated electrocatalysis with an air-stripping method for the recovery of high-purity ammonia products at − 1.0 V *versus* RHE (Fig. 5g). After the $NO_3RR$ process, the cathodic electrolyte was transferred to a conical flask and adjusted to an alkaline state. Due to the high $NH_3$ vapor pressure under alkaline conditions, the produced $NH_3$ was efficiently extracted at 70 °C using an air stripping method (see details in Methods)[73,74]. Consequently, approximately 98.9% of the $NH_3$ vapor was successfully stripped out from the electrolyte (Fig. 5h), indicating favorable $NO_3^-$ conversion and simultaneous high-efficiency $NH_3$ generation. Subsequently, around 97.3% of the outflowing $NH_3$ gas was collected in an HCl solution, and approximately 97.1% of $NH_4Cl$ powder was finally obtained after rotary evaporation. The high-purity $NH_4Cl$ products were confirmed by XRD measurement (Supplementary Fig. 53), highlighting their potential as fertilizer for agricultural production.

Next, to investigate the practical application prospect, complex electrolyte (which included $CO_3^{2-}$, $Na^+$, $Cl^-$, $K^+$, $NO_3^-$, and $SO_4^{2-}$) was prepared to simulate the nitrate-containing wastewater condition[75].

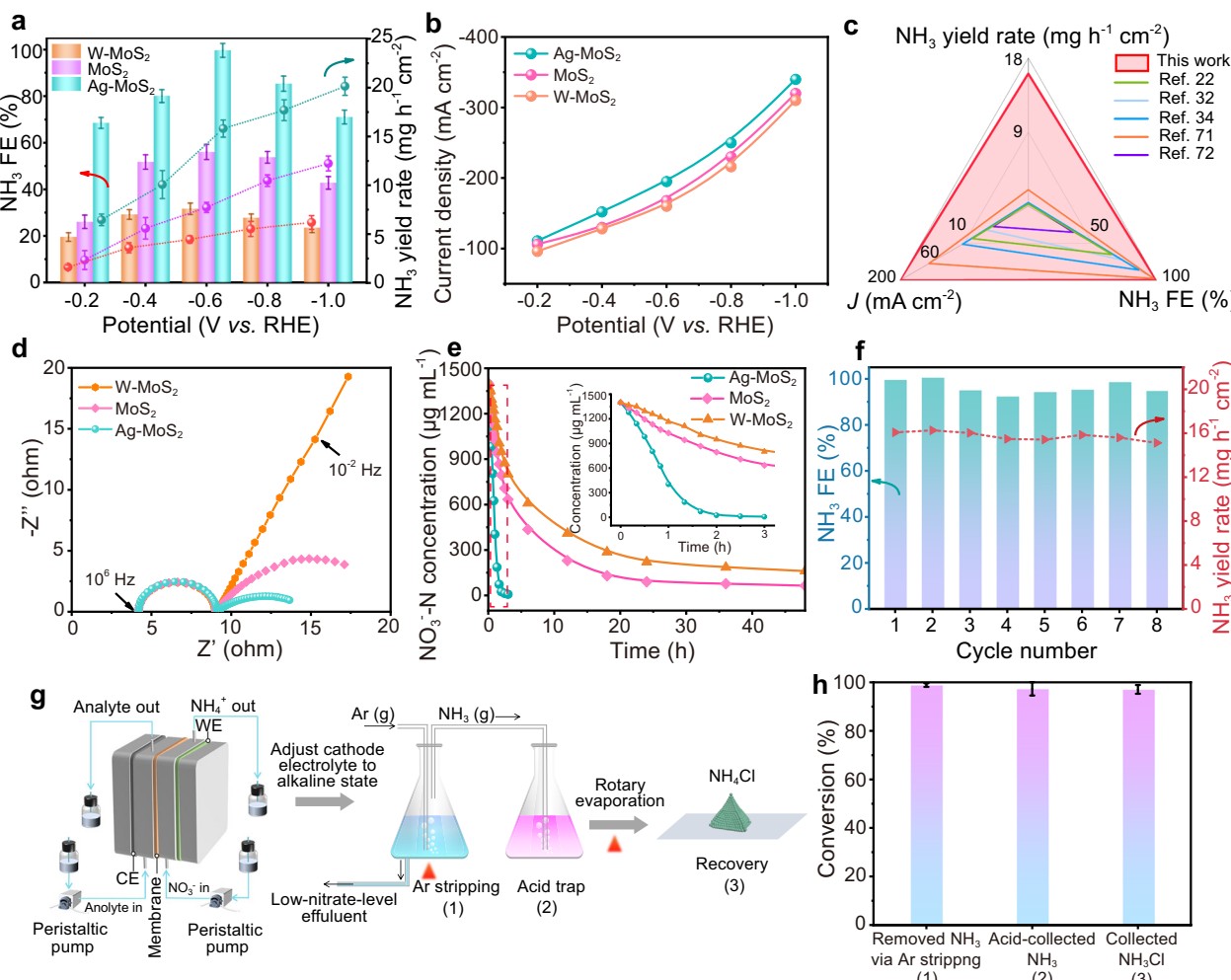

**Fig. 5 | NO₃RR performance under ultralow NO₃⁻ concentrations. a** NH₃ yield rate and NH₃ FE of catalysts in 10 mM NO₃⁻ electrolyte at various applied potentials. **b** I–V plots of catalysts in 0.5 M K₂SO₄ with 10 mM NO₃⁻ electrolyte at various potentials for 0.5 h electrolysis. **c** A summary of recent NO₃RR works on NH₃ yield rate, NH₃ FE and current density under ultralow NO₃⁻ concentrations (NO₃⁻ concentration ≤10 mM)[22,32,34,71,72]. **d** EIS tested at −0.6 V versus RHE in 10 mM NO₃⁻ electrolyte. **e** NO₃⁻ removal in initial 0.5 M K₂SO₄ with 100 mM NO₃⁻ electrolyte at −0.6 V versus RHE in H-cell reactor. After 3 h of electrolysis, only 0.54 mM of

NO₃⁻−N. Insert was the enlarged vision within 3 h. **f** NO₃RR performance stability over Ag-MoS₂ measured in a 0.5 M K₂SO₄ with 10 mM NO₃⁻ electrolyte at −0.6 V versus RHE. **g** Schematic of the ammonia product synthesis process from 100 mM NO₃⁻ electrolyte to NH₄Cl at −1.0 V versus RHE. **h** The conversion efficiency of different steps for the ammonia product synthesis process. Numbers on the x-axis indicated the corresponding conversion steps in panel (g). Error bars indicate the relative standard deviations of the mean (n = 3).

The NH₃ Faradaic efficiency over Ag-MoS₂ in simulated wastewater was 76.7%, while the NH₃ yield rate reached 16.02 mg h⁻¹ cm⁻² (Supplementary Fig. 54a). The decreased NO₃RR performance can be ascribed to the presence of interfering ions that influenced the targeted adsorption of NO₃⁻. Although the NO₃RR performance in wastewater decreased to a certain extent, the NH₃ yield rate and Faradaic maintained 80% and 76% of the optimal performance for simple electrolyte. In the recovery process, approximately 73.9% of NH₄Cl powder was finally obtained (Supplementary Fig. 54b). Also, we have evaluated the preparation cost of Ag-MoS₂ catalyst, whose price with 1 cm² was as low as 0.475 ¥ (Supplementary Table 8). These results suggested that the Ag-MoS₂ has the potential for the conversion of NO₃⁻ to ammonia product in actual industrial and agricultural wastewater.

To gain a deeper understanding of the NO₃RR mechanism over Ag-MoS₂ catalysts, we employed in situ attenuated total reflection infrared spectroscopy (ATR-IR) to capture intermediates and monitor the reaction. As the applied potential ranged from 0.4 to −1.0 V versus RHE (Fig. 6a), the detected NO₂ peaks (at ~1530 cm⁻¹)[76] in the spectra of Ag-MoS₂ indicated the deoxygenation of NO₃⁻. The pronounced characteristic peaks of N-H (at 3200–3380 cm⁻¹)[77–79], -H (at ~2050 cm⁻¹)[80], and

NH₄⁺ (at ~1450 cm⁻¹)[81] verified the effective hydrogenation of nitrogen oxide intermediates to ammonia on a highly protonated surface. In comparison, the NO₂, -H, and NH₄⁺ signals were not prominent in the spectra of MoS₂ and W-MoS₂ (Supplementary Figs. 55, 56), indicating that the relatively slow reaction rates of the two catalysts resulted in less accumulation of related species on their surfaces. Online differential electrochemical mass spectrometry (DEMS) was further performed to detect molecular intermediates and products (Fig. 6b and Supplementary Figs. 57, 58). The mass-to-charge ratio (m/z) signals of NH₃ (17), H₂ (2), N₂ (28), NO (30), NH₂OH (33), and N₂O (44) were detected and tracked. Among the three catalysts, the NH₃ intensity was the strongest, while other signals (H₂, N₂, NO, and N₂O) shrunk over Ag-MoS₂, which demonstrated the promotion of NH₃ generation under the S-L junction-induced NO₃⁻ enrichment on the Ag-MoS₂ surface. Further, we calculated the electronic band structures and partial density of states (PDOS) through density functional theory (DFT). After Ag doping, the newly emerged band tail states at VBM extended the valence band of Ag-MoS₂ (Fig. 6c, d and Supplementary Fig. 59)[82,83], leading to the downward shift of the Fermi level. While, W doping induced upward shift of the Fermi level away from the VBM (Supplementary Fig. 60). These phenomena

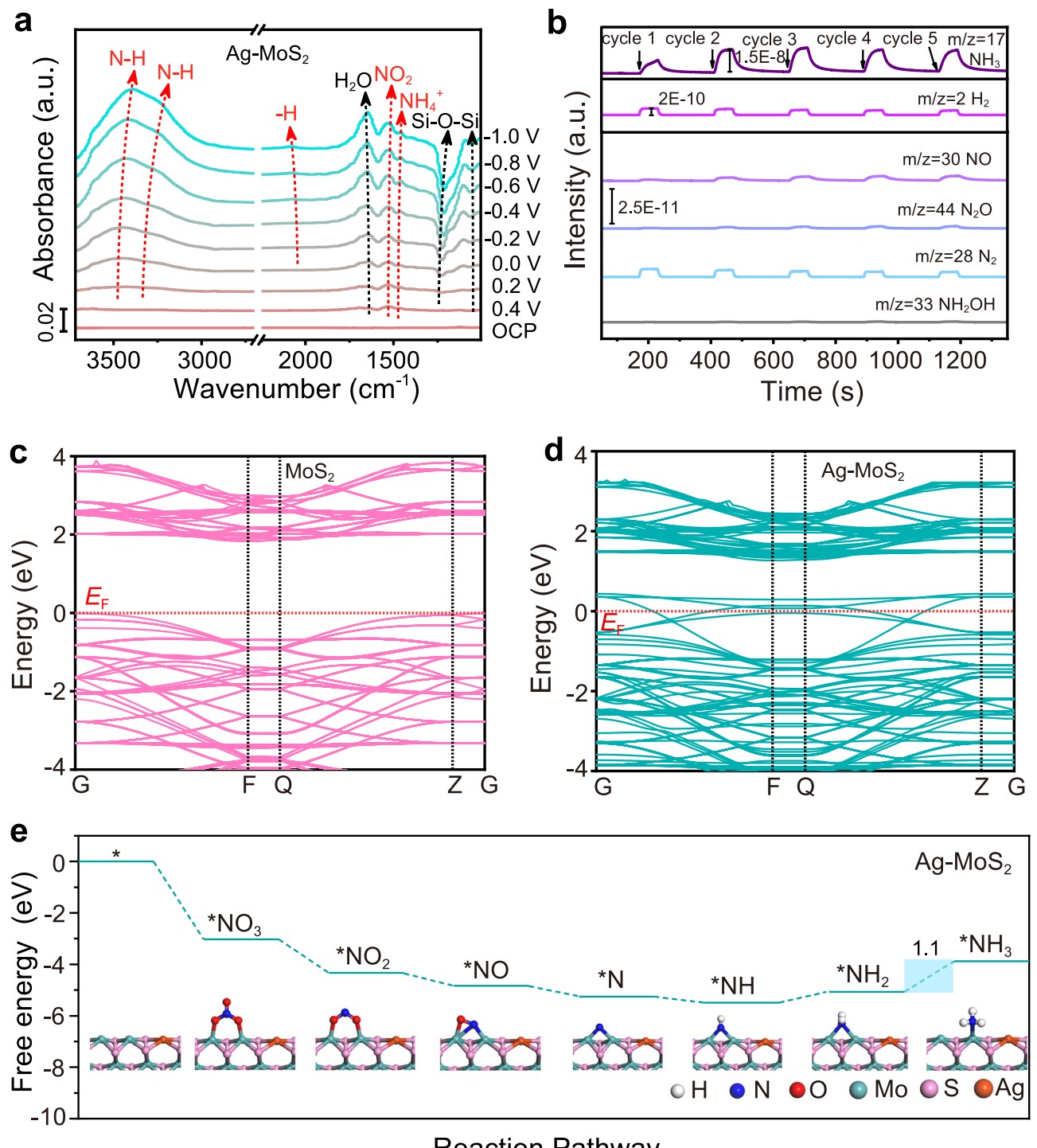

**Fig. 6 | Mechanistic study on NO₃⁻ electroreduction. a** In situ ATR-IR spectra of Ag-MoS₂. The Si-O signal was derived from the reduction of surface SiO₂ on the Si semi-cylindrical prism substrate under the applied potentials. **b** Online differential electrochemical mass spectrometry (DEMS) measurements of NO₃RR over Ag-

MoS₂ under the potential of −0.6 V *versus* RHE. Electronic band structures of (**c**) MoS₂ and (**d**) Ag-MoS₂. **e** Gibbs free energy diagram of various intermediates generated during NO₃RR over Ag-MoS₂ at pH = 7.

were in consistent with UPS and M-S results, unveiling that Ag doping benefited the downward band bending within S-L junction and thus the charge transfer between IHP and the catalyst. The optimal reaction pathways of Ag-MoS₂ were also calculated (*NO₃ → *NO₂ → *NO → *N → *NH → *NH₂ → *NH₃) by comparing multiple possible branches (Fig. 6e and Supplementary Figs. 61–63). All three samples underwent the process of adsorption of NO₃⁻, deoxygenation of the N species, and hydrogenation of the N species to synthesize NH₃, in which the

*NH₂ → *NH₃ process was the potential-determining step (PDS). The Gibbs free energy change (ΔG) of this step was 1.1, 0.95, and 1.02 eV for Ag-MoS₂, MoS₂, and W-MoS₂ catalysts at pH = 7. The ΔG of Ag-MoS₂ was comparable to those of W-MoS₂ and MoS₂, suggesting that Ag doping had little contribution to the ΔG of PDS. Although MoS₂ and W-MoS₂ exhibited reduced ΔG of PDS at corresponding optimal NO₃⁻ adsorption potentials (Supplementary Figs. 61 and 64, 65), while, their NO₃RR performance were far below Ag-MoS₂ (Fig. 5a). These results further

indicated the decisive role of Ag doping-induced surface $NO_3^-$ enrichment effect in promoting $NO_3RR$ activity in low concentration system.

## Discussion

In summary, we proposed a strategy to enrich charged chemicals in the vicinity o of the electrodes through catalyst bandstructure engineering, overcoming the bottleneck of mass transfer posed by intrinsic charge repulsion in electrocatalysis. We systematically demonstrated the S-L junction-mediated charge rearrangement effect on the IHP region, boosting electrocatalytic reduction of low-concentration $NO_3^-$ to $NH_3$. COMSOL simulations and experimental measurements, including in situ Raman spectra, IC, EIS, and ACV, elucidated that the construction of S-L junction induced positively charged IHP, significantly enriching $NO_3^-$ anions concentration (by a factor of 27.6) to break the mass transfer limitation at the reduction potential.

As a proof of concept, an optimized Ag-$MoS_2$ was introduced as a model platform for the $NO_3RR$ flow cell system. We evaluated the comprehensive performance through the decontamination of nitrate polluted solution into drinkable level, which requires efficient catalytic process within a broad range of concentration and high conversion rate. By virtue of the unparalleled performance at ultra-low concentrations ($\sim 20$ mg h$^{-1}$ cm$^{-2}$, FE of nearly 100% under concentration of 10 mM), the catalyst with S-L junction succeeded in removing $NO_3^-$ to low level within 3 h, while the counterpart without the junction cannot meet the level after treatment of 48 h. Considering the practicality of the utilization of the high-value-added ammonia product, we implement additional experimental efforts to show a near-unity efficiency by the conversion of $NO_3^-$ to a high-purity $NH_4Cl$, aiming to achieve sustainable development.

Efficient anion accumulation on the reaction interface is crucial for a wide range of renewable electricity-driven reduction applications. Thus, the strategy we proposed holds promise for various electrochemical anion reduction reactions, including $NO_3^-$, $NO_2^-$, $AsO_4^{3-}$, and $Cr_2O_7^{2-}$, etc, inspiring innovative design for water treatment and remediation in an environmentally friendly way. Such investigations could illuminate avenues for both fundamental research and practical applications across various scientific domains, ranging from physics, chemistry to environmental sciences.

## Methods

### Chemicals

Sodium molybdate dehydrate ($Na_2MoO_4 \cdot 2H_2O$, $\geq 99.95\%$), thiourea ($CH_4N_2S$, $\geq 99\%$), silver nitrate ($AgNO_3$, $\geq 99.8\%$), hydroxylamine hydrochloride ($NH_2OH \cdot HCl$, $\geq 99.99\%$), hexadecyl trimethyl ammonium bromide ($C_{19}H_{42}BrN$, CTAB, $\geq 99\%$), sodium tungstate dihydrate ($Na_2WO_4 \cdot 2H_2O$, $\geq 99.5\%$), salicylic acid ($C_7H_6O_3$, $\geq 99\%$), trisodium citrate dihydrate ($Na_3C_6H_5O_7 \cdot 2H_2O$, $\geq 99\%$), sulfanilamide ($C_6H_8N_2O_2S$, $\geq 99.8\%$), $p$-dimethylaminobenzaldehyde (($CH_3)_2NC_6H_4CHO$, PDAB, $\geq 98\%$), sodium nitroferricyanide (III) dihydrate ($Na_2Fe(CN)_5NO \cdot 2H_2O$, $\geq 99.98\%$), N-(1-Naphthyl) ethylenediamine dihydrochloride ($C_{12}H_{14}N_2 \cdot 2HCl$, $\geq 98\%$), potassium sulfate ($K_2SO_4$, $\geq 99\%$), potassium nitrate-$^{15}N$ ($^{15}KNO_3$, $\geq 99\%$), dimethyl sulfoxide ($C_2H_6SO$, DMSO-$d6$, $\geq 99.9\%$), maleic acid ($C_4H_4O_4$, $\geq 99\%$), and ammonium chloride ($^{14}NH_4Cl$, $\geq 99.8\%$), ammonium-$^{15}N$ chloride ($^{15}NH_4Cl$, $\geq 98.5\%$) were purchased from Aldrich Chemical Reagent Co., Ltd. Sodium hydroxide (NaOH, $\geq 96\%$), sodium hypochlorite (NaClO, analytical pure), hydrochloric acid (HCl, 38%), potassium nitrate ($KNO_3$, $\geq 99.999\%$), potassium nitrite ($KNO_2$, $\geq 97\%$), and ethanol ($C_2H_6O$, $\geq 99.7\%$) were purchased from Sinopharm Chemical Reagent Co., Ltd. All chemicals were used as received without further purification. The water used in this research was purified through a Millipore system.

### Preparation of Ag-$MoS_2$

For the synthesis of Ag-$MoS_2$ in a typical hydrothermal process, 4.27 mmol of $Na_2MoO_4 \cdot 2H_2O$, 18.40 mmol of $CH_4N_2S$, a certain amount of $AgNO_3$, and 10.43 mmol of $NH_2OH \cdot HCl$ were dissolved in 50 mL of deionized water under vigorous stirring for 0.5 h. 0.18 g of CTAB was then added to the mixed solution, and the pH value of the mixture was adjusted to 6. After that, carbon cloth with an area of $4 \times 4$ cm$^2$ and a thickness of 0.35 mm was immersed into the homogeneous solution. The mixture was subsequently transferred into a 100 mL Teflon-lined stainless-steel autoclave and heated at 180 °C for 24 h in an oven. After cooling down to room temperature naturally, the products were washed with ethanol and deionized (DI) water repeatedly, followed by vacuum-freeze drying. The obtained sample was Ag-$MoS_2$ grown on carbon cloth (catalyst mass of $\sim 5$ mg cm$^{-2}$). Ag-$MoS_2$ catalysts with various Ag-doped amounts were prepared according to the above procedure by changing the additional amount of $AgNO_3$. Specifically, 0.213 mmol, 0.426 mmol, 0.639 mmol and 0.852 mmol $AgNO_3$ were added to achieve 1.87 wt%, 3.49 wt%, 5.91 wt% and 6.91 wt% Ag doping amounts, respectively. These catalysts were denoted as Ag-$MoS_2$-5, Ag-$MoS_2$-10, Ag-$MoS_2$-15 and Ag-$MoS_2$-20 (Supplementary Table S1).

### Preparation of $MoS_2$

In a typical synthesis of $MoS_2$, 4.27 mmol of $Na_2MoO_4 \cdot 2H_2O$, 18.40 mmol of $CH_4N_2S$, and 10.43 mmol of $NH_2OH \cdot HCl$ were dissolved in 50 mL of deionized water under vigorous stirring for 0.5 h. 0.18 g CTAB was then added to the mixed solution, and the pH value of the mixture was adjusted to 6. After that, carbon cloth with an area of $4 \times 4$ cm$^2$ and a thickness of 0.35 mm was immersed into the homogeneous solution. The mixture was subsequently transferred into a 100 mL Teflon-lined stainless-steel autoclave and heated at 180 °C for 24 h in an oven. After cooling down to room temperature naturally, the products were washed with ethanol and DI water repeatedly, followed by vacuum-freeze drying. The obtained sample was $MoS_2$ grown on carbon cloth (catalyst mass of $\sim 5$ mg cm$^{-2}$).

### Preparation of W-$MoS_2$

To systhesize W-$MoS_2$, 4.27 mmol of $Na_2MoO_4 \cdot 2H_2O$, 18.40 mmol of $CH_4N_2S$, 0.426 mmol of $Na_2WO_4 \cdot 2H_2O$, and 10.43 mmol of $NH_2OH \cdot HCl$ were dissolved in 50 mL of deionized water under vigorous stirring for 0.5 h. 0.18 g of CTAB was then added to the mixed solution, and the pH value of the mixture was adjusted to 6. After that, carbon cloth with an area of $4 \times 4$ cm$^2$ and a thickness of 0.35 mm was immersed into the homogeneous solution. The mixture was subsequently transferred into a 100 mL Teflon-lined stainless-steel autoclave and heated at 180 °C for 24 h in an oven. After cooling down to room temperature naturally, the products were washed with ethanol and DI water repeatedly, followed by vacuum-freeze drying. The obtained sample was W-$MoS_2$ grown on carbon cloth (catalyst mass of $\sim 5$ mg cm$^{-2}$).

### Electrochemical testing

Before the $NO_3RR$ tests, the pretreatment of Nafion 117 membrane (area: a circular shape with a diameter of 2 cm and a thickness of $\sim 183$ μm) was as follows: the membrane was first oxidized in 5% $H_2O_2$ solution at 80 °C for 1 h, next boiled in DI water for 1 h, then used 0.5 M $H_2SO_4$ at 80 °C for 1 h, finally washed with DI water.

The $NO_3RR$ tests were performed on an electrochemical workstation (PARSTAT 4000) with a typical three-electrode flow cell, including as-prepared catalyst electrode (area of $1 \times 1$ cm$^2$, catalyst mass of $\sim 5$ mg cm$^{-2}$), $IrO_2$ electrode (area of $1 \times 1$ cm$^2$), and a saturated calomel electrode as the working electrode, counter electrode, and reference electrode, respectively. Nafion 117 membrane was used to separate the anodic cell and cathodic cell. The flow rate of electrolyte (0.5 M $K_2SO_4$ with 10 mM $NO_3^-$) was set at 2 mL min$^{-1}$ in cathodic and anodic chambers. All potentials reported in this work were referred to the RHE scale via calibration by the following equation: E(versus RHE) = E(versus SCE) + 0.244 + 0.0591 × pHvalue. The three

batches of catalysts (Ag-MoS$_2$ (AgNO$_3$ amount of 0.426 mmol), MoS$_2$ and W-MoS$_2$) were synthesized, respectively, and their NO$_3$RR performance at various potentials were tested. The error bars were the mean values standard deviation according to the obtained data. For the chronoamperometry measurement, the potential was applied from − 0.2 to − 1.0 versus RHE. LSV was carried out in a voltage window from 0.25 to − 1.65 V versus RHE at scan rates of 10 mV·s$^{-1}$. EIS of the samples was performed at the frequency region from $10^6$ to $10^{-2}$ Hz with an amplitude of 20 mV at various applied potentials. During the electrochemical testing process, no iR compensation was performed. Mott-Schottky measurements were carried out at the frequency of 1000 Hz under dark conditions to obtain the flat band potential and carrier concentration of the samples, in which the dielectric constant of MoS$_2$ was 7.6[75,84].

## Detection and quantification of NH$_3$ using UV-vis

The concentration of NH$_3$ after chronoamperometry measurements with different potentials was detected by the salicylic acid method[85]. The cathodic electrolyte after electrolysis was collected and diluted to the detection range. Then, 2 mL of diluted liquid was added to 2 mL of 1 M NaOH solution containing 5 wt% C$_7$H$_6$O$_3$ and 5 wt% Na$_3$C$_6$H$_5$O$_7$·2H$_2$O. 1 mL of 0.05 M NaClO and 0.2 mL of C$_5$FeN$_6$Na$_2$O solution (1 wt%) were subsequently mixed with the aforementioned solution and shaken well. After standing for 2 h, UV-vis spectrophotometer measurements were performed with the range from 500 to 800 nm, and the absorbance value at the wavelength of 655 nm was recorded. The concentration-absorbance curve was calibrated using standard ammonium chloride solution with concentrations of 0.5, 1.0, 3.0, 5.0, 10.0, 20.0 and 30.0 μg mL$^{-1}$ in 0.5 M K$_2$SO$_4$ with NO$_3^-$. The fitting curve ($y = 0.10x + 0.01$, $R^2 = 0.999$) exhibited a good linear relation of absorbance value with NH$_4^+$ concentration.

## Detection and quantification of NO$_3^-$ using UV-vis

The cathodic electrolyte after electrolysis was collected and diluted to the detection range. In the process, 5 mL of diluted sample solution was mixed with 0.1 mL of 1 M HCl. After standing for 20 min, the UV-vis absorbance at the wavelength ranging from 215 to 280 nm was recorded[86,87]. The absorbance difference between the wavelengths at 220 and 275 nm was calculated using the equation: $A = A_{220\,nm} - A_{275\,nm}$. The concentration-absorbance difference curve was calibrated using standard KNO$_3$ solution with 1, 5, 10, 20, 30 and 50 μg mL$^{-1}$ concentrations. The fitting curve ($y = 0.034x - 0.036$, $R^2 = 0.999$) exhibited a good linear relation of absorbance value with NO$_3^-$ concentration.

## Detection and quantification of NO$_2^-$ using UV-vis

The preparation of the color reagent was as follows[86,87]: First, 0.5 g of C$_6$H$_8$N$_2$O$_2$S was dissolved in 50 mL of 2.0 M HCl solution, which was labeled as reagent A. 20 mg of N-(1-naphthyl) ethylenediamine dihydrochloride was then dispersed in 20 mL of DI water, which was labeled as reagent B. After 0.1 mL of reagent A mixed with 5 mL of standard or diluted sample solutions, 0.1 mL of reagent B was finally injected into the homogeneous solution, shaking up and standing for 30 min. The UV-vis absorbance at the wavelength ranging from 400 to 640 nm was recorded, in which the characteristic absorption peak of NO$_2^-$ was identified at 540 nm. The concentration-absorbance difference curve was calibrated using standard KNO$_2$ solution with concentrations of 0.2, 0.5, 1.0, and 2.0 μg mL$^{-1}$. The fitting curve ($y = 0.446x + 0.039$, $R^2 = 0.999$) exhibited a good linear relation of absorbance value with NO$_2^-$ concentration.

## Detection and quantification of NH$_3$ using $^1$H NMR

$^{14}$NO$_3^-$ and $^{15}$NO$_3^-$ isotope labeling experiments were carried out on a Bruker AVANCE III HD NMR spectrometer (600 MHz) to support the UV-vis results[12]. After electrolysis, the pH value of the cathodic electrolyte was adjusted to 2 with 1 M HCl. 0.5 mL of the homogeneous solution was then mixed with 0.1 mL DMSO-d6 with 0.04% C$_4$H$_4$O$_4$. $^1$H NMR was recorded to quantitatively determine NH$_3$ concentration according to the corresponding standard curves.

## Electrochemical in situ Raman spectroscopy.

In situ Raman spectroscopy was measured by inVia Reflex (Renishaw, UK) with a 532 nm laser as the excitation source[20]. The electrochemical process was performed in the three-electrode custom-made Teflon reactor with a quartz window, including Ag/AgCl, Pt wire, and catalysts coated on the Au electrode as the reference electrode, counter electrode, and working electrode, respectively. The spectra were recorded by the potential from OCP to − 0.6 V versus RHE.

## Electrochemical in situ ATR-IR spectroscopy.

ATR-IR spectroscopy was tested on a Nicolet iS50 FT-IR spectrometer equipped with an MCT detector and cooled by liquid nitrogen[20]. The electrochemical process was performed in the three-electrode custom-made reactor, including Ag/AgCl and Pt wire as the reference electrode and counter electrode, respectively. The working electrode was prepared as follows: The Si semi-cylindrical prism was first polished with Al$_2$O$_3$ powder and sonicated in acetone and deionized water. The Si was placed in a piranha solution at 60 °C for 20 min to remove the organic contaminants. Then the reflecting surface of Si was deposited in the Au precursor mixture at 60 °C for 10 min, and the Au-coated Si conductive substrate was obtained. The catalyst ink was finally coated on the substrate-reflecting surface. After that, the spectra were recorded by the potential from 0.4 V to − 1.0 V versus RHE. The spectrum collected at OCP was as background subtraction.

## Electrochemical online DEMS tests.

The online DEMS tests[21] were measured in the three-electrode customized reactors containing 0.5 M K$_2$SO$_4$ with 10 mM NO$_3^-$ electrolyte, including catalysts coated on breathable film with Au plating layer, Pt wire, and saturated Ag/AgCl electrode as the working electrode, the counter electrode and the reference electrode, respectively. The potential of OCP and − 0.6 V versus RHE were applied alternately with an interval of 3 min after the baseline of the mass spectrometry kept steady. During the electrochemical process, the differential mass signals were recorded when the gaseous products formed on the electrode surface. The operation process was repeated five cycles to avoid accidental errors.

## Direct ammonia product synthesis.

By coupling NO$_3$RR with the Ar stripping process, the cathodic electrolyte after electrolysis was adjusted to alkaline state, and sealed in a conical flask at 70 °C with flowing 100 sccm Ar gas to purge the NH$_3$ gas out. The outlet stream was injected into 2 M HCl to capture the NH$_3$ product. The NH$_3$ amount in all the solutions was evaluated by the salicylic acid method mentioned above, and the removal efficiency and collection efficiency were calculated as following equations[12], respectively:

$$\text{Removed NH}_3 \text{ via Ar stripping} = 1 - \frac{\text{NH}_3 \text{ left after Ar stripping (mol)}}{\text{initial NH}_3 \text{ (mol)}} \tag{1}$$

$$\text{Acid collected NH}_3 = \frac{\text{NH}_3 \text{ in acid trap (mol)}}{\text{removed NH}_3 \text{ via Ar stripping (mol)}} \tag{2}$$

To estimate the efficiency of the NH$_4$Cl product, the HCl solution with the trapped NH$_3$ was dried by rotary evaporator at 70 °C. The generated NH$_4$Cl was measured by a balance and analyzed by XRD. The collection efficiency of NH$_4$Cl from the acid trap was calculated by the

following equation:

$$\text{Collected NH}_4\text{Cl from acid trap} = \frac{\text{collected dried out NH}_4\text{Cl (mol)}}{\text{acid collected NH}_3\text{ (mol)}}$$

(3)

**DFT calculations.** All calculations were carried out by spin-polarized DFT with the Vienna Ab initio Simulation Package (VASP)[88,89]. Electron exchange-correlation was expressed by the Perdew-Burke-Ernzerhof (PBE) functional within the generalized gradient approximation (GGA)[90]. To describe the ionic cores, the projector augmented wave (PAW) pseudopotential was applied[91,92]. The plane wave energy cutoff, and convergence criterion for electronic energy and forces were set as 450 eV, $10^{-5}$ eV, and 0.02 eV/Å. van der Waals (VDW) interactions were corrected using the D3 method of Grimme[93]. To compare the electronic structures of $MoS_2$, $Ag\text{-}MoS_2$ and $W\text{-}MoS_2$, we constructed a $2 \times 2 \times 2$ $MoS_2$ supercell containing 16 Mo atoms and 32 S atoms. Substituting one Mo atom with an Ag atom is referred to as $Ag\text{-}MoS_2$, while substituting one $MoS_2$ atom with a W atom is referred to as $W\text{-}MoS_2$. For investigating the reaction pathways of nitrate ions on three catalytic surfaces, we built a nanoribbon containing 20 Mo atoms and 40 S atoms. The bottom two layers were fixed to simulate the edge of a single-layer $MoS_2$. For $Ag\text{-}MoS_2$ and $W\text{-}MoS_2$, we replaced one edge Mo atom with an Ag atom and a W atom, respectively. To avoid periodic interlayer interactions, we introduced a vacuum layer of 20 Å in the Z and Y directions. Aqueous phase $H_2O$ and $NO_3^-$ were as the energetics references.

## Data availability

Full data supporting the findings of this study are available within the article and its Supplementary Information, as well as from the corresponding author upon reasonable request. Source data are provided in this paper.

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

## Acknowledgements

We gratefully thank the Foundation for Innovative Research Groups of the National Natural Science Foundation of China (No. 52121004 for L.C.), National Natural Science Foundation of China (Grant No. 22376222 for M.L., 52372253 for J.F., 52202125 for C.C.), the Science and Technology lnnovation Program of Hunan Province (2023RC1012 for M.L.), Central South University Research Program of Advanced Interdisciplinary Studies (Grant No. 2023QYJC012 for M.L.), Central South University Innovation-Driven Research Program (Grant No. 2023CXQD042 for J.F.), and Hunan Provincial Natural Science Foundation (2025JJ60117 for W.L.). We would like to acknowledge the help from Beam Lines BL01C1 in the National Synchrotron Radiation Research Center (NSRRC, Hsinchu, Taiwan) for various synchrotron-based measurements. We also acknowledge funding and support from the Deutsche Forschungsgemeinschaft (DFG) under Germany´s Excellence Strategy – EXC 2089/1 – 390776260 cluster of excellence e-conversion, the Bavarian program Solar Technologies Go Hybrid (SolTech) and the Center for NanoScience (CeNS). L.Z. acknowledges the LMU-CSC program for a doctoral fellowship at LMU. We are grateful for resources from the High Performance Computing Center of Central South University.

## Author contributions

W.L., J.W., and M.L. conceived the research and supervised the project. W.L. and J.W. performed the experiments. W.L., J.W., M.L., X.Z., Q.W., C.K., T.C., H.L., Y.Z., C.C., J.F., L.Z., L.C., and B.X. analyzed the data. Y.T. and K.L. carried out the DFT simulations. W.L., J.W., C.L., and E.C. wrote the draft. All authors discussed the results and contributed to the writing of the manuscript.

## Funding

## Competing interests

The authors declare no competing interests.
