## [Peer Review File · Nature Communications]

Near-Unity Nitrate to Ammonia Conversion via Reactant Enrichment at the Solid-Liquid Interface

Corresponding Author: Professor Emiliano Cortes

Version 0:

Reviewer comments:

Reviewer #1

(Remarks to the Author)

The manuscript reports a method to construct a Ag-doped MoS₂ catalyst for nitrate reduction. Because the band bending at interface, a negative charge accumulation at the catalyst surface, which induces cations at Helmholtz layers. However, through the manuscript, there is several contradictory descriptions about what type of charges are accumulated at Helmholtz layer. The authors' aim is to induce an increase in the NO₃⁻ concentration. Many technologies are used to confirm that. However, the authors failed to relate the excess charge to anions. The physical mechanism is not stated clearly. Only a COMSOL simulation is provided to demonstrate the enrichment of NO₃⁻. In particular, from the Line 167 where the authors stated the increase in positive charge at IHP to the experiments in the following two paragraph, no discussion and explanation is provided. In addition, there are many errors in the manuscript. I would suggest a major revision.

The authors claim extensive free electrons packed on its IHP. The statement is wrong. There are no free electrons in IHP. Ions can usually appear at IHP. In the abstract, "Particularly, at the reduction potential, an excess of free electrons accumulates on the inner Helmholtz plane (IHP), the solution side of the double layer where catalytic reactions occur". The statement of the authors is wrong. There is no excess of free electrons but excess charge. Free electrons are usually not able to exist in an aqueous electrolyte. In addition, "free electrons" are used for metals or in vacuum. At the electrode/electrolyte interfaces, the electrodes acquire more electrons upon negative polarization and induce the contact adsorption of partially dissociated cations at atomic length. Away from the surface of electrodes, there are outer Helmholtz plane(OHP) and diffusive layer, respectively.

A p-type semiconductor is used as the cathode. Upon negative polarization, the conductivity decreases dramatically, which is not favorable for electrocatalysis.

Figure 2f shows the EXAFS spectra of Ag-MoS₂ and reference samples. However, the K-edge XANES spectra are now shown. Please provide and discuss them.

The work function of Ag-doped WS₂ is 5.47eV which is greater than that of MoS₂ (4.98V). Previous studies show that precious metal doped MoS₂ induced the decrease in working function(Chem Mater 2019;31(2):429). Please explain the reason. It should be confirmed by Kelvin probe force microscope. Doping induces mid-gap states which decrease the work functions.

Figure 1 shows all negative ions at IHP, which is not correct. Because the surface of electrodes accumulates electrons under negative potentials, which polarize electrolytes and attract water molecule dipole or cations at IHP. From OHP to diffuse layer, the concentration of excess counter-ions decreases and approaches to the bulk concentration. According to the classic model, the IHP usually are water molecules or counter-ions of excess charge at electrode surface. Negative polarization leads to the accumulation of electrons at the surface of cathodes, which prefers to attract cations toward cathodes. In some cases, there may be some contact adsorption of cations, water, and anions at IHP. The contact adsorption mainly depends on the chemical natures of adsorbents rather than charge, which is called superequivalent adsorption. As far as the absolute values are concerned, the excess charge at electrode (Q_{ec}) may be less than that of contact adsorption

(Qca). The charge at OHP (Q_{ohp}) is less than the total excess charge of solution from IHP to bulk (Q_s), which is equal to Q_{ec} .

Figure S24 show very different concentration distribution for three samples. However, No range normal to the surface was indicated. It is hard to have an idea of length. Are they at equilibrium or please indicate the time at which the distribution was shown.

In line 190, severe downward shift of E_f into the valence band of Ag-MoS₂ (Figure S23). The figures are not related to the description. The authors used the diameter of semicircle to derive that negative polarization leads to the formation of degenerate semiconductor. However, the voltage is too low too negative. Is that the bubbling effects which stirred the solution.

Figure S29 show the comparison with and without NO₃⁻. The LSV is like the HER. There should be a well-designed control to indicate what ratio of the HER occurred? By decreasing the concentration of NO₃⁻, you should find a diffusion-limited peak which is the sign of NO₃RR. The straight LSV lines do not tell the truth.

Figure 3e indicates a y-axis title of "peal", which may be "peak". Figure 3c has an x-axis title of Potential?

Reviewer #2

(Remarks to the Author)

The manuscript by Liao et al. presents an interesting hypothesis on the design of electrocatalysts for electroreduction of nitrate to ammonia. However, the presented data do not seem to provide a robust confirmation of the authors' interpretations. Moreover, the overall experiment design is questionable. Under these circumstances, this manuscript cannot be recommended for publication.

1. Introductory explanations of the importance of the nitrate-to-ammonia reduction technology should be revised, as this process cannot be considered neither as a sustainable source of ammonia nor as an alternative to the Haber-Bosch process in the way it is presented [10.1038/s41929-023-01060-w].

2. Directly linked to the comment above is the questionable purpose of this study. What is the practical reason for the transformation of dilute nitrate solutions into dilute solutions of similarly toxic ammonium? The authors put a lot of emphasis on remediation and achieving WHO drinkable limits. However, if this would be the aim of the work, the target product should be genuinely benign N₂, not ammonium.

3. Further, the choice of conditions used to study the nitrate-to-ammonia are very hard to understand. Why using non-buffering K₂SO₄ electrolyte, which provides essentially no control over the reaction conditions and is very challenging to translate into a practical technology? Practical nitrate to ammonia reduction requires the use of alkaline conditions, which can be integrated into a PEM-based device as recently demonstrated [10.1038/s41929-024-01200-w]. However, the use of alkaline conditions for a Mo-based electrode might not be a fruitful idea due to their limited stability under these conditions. This further questions the impact of the presented results.

4. It is important to recognise that MoS₂ produces only ~33% less ammonia than AgMoS₂, i.e. differences in the rate of the target process are not substantial. Hence, even if the proposed effects are real (see next comment), they are not sufficiently strong to form a basis for a strategy to design effective electrocatalysts for the nitrate-to-ammonia reduction reaction.

5. The evidence for the increase in the local, near-electrode concentration of nitrate due to the Ag doping of MoS₂ is unconvincing:

5.1. Comprehensive control data for high-surface area Ag nanoparticles deposited on a different support, e.g. carbon black, are needed. Probably, simple silver nps would produce even better results?

5.2. Increased Raman signals on Ag/MoS₂ can be just a plasmonic, SERS, effect.

5.3. Increased NO₃⁻ adsorption can be just due to the preferential adsorption of nitrate on the surface of silver nanoparticles, which has nothing to do with localised concentration increase.

5.4. If the differences in the local NO₃⁻ concentrations were as substantial as presented, why the rates of the nitrate reduction are essentially the same for MoS₂ and Ag-MoS₂, the difference being only some preference of the latter for the formation of ammonium?

Overall, none of the methods employed probes the phenomenon in question. The reviewer recognises that this is exceptionally hard, but this does decrease the current very high level of speculations in the presented interpretations.

6. Standard potential for the nitrate-to-ammonia redox couple is approximately 0.7 V vs. SHE, not 0.3 V vs. RHE at pH 7.

7. How can the authors explain the difference in the double-layer capacitance measured by ACV and dc voltammetry, which show opposite trends?

8. Qualitative interpretations of the non-linear capacitance measured by ACV can hardly produce meaningful results,

especially in terms of the PZC. Interpretation of the potential corresponding to the capacitance minimum in terms of PZC is highly questionable. Moreover, the presented data are within the experimental error given the poorly defined shape of the fundamental harmonic.

9. Qualitative interpretations of impedance data in terms of the phase angle dependence only is very hard to understand and accept. EIS should be modelled with a physicochemically sensible model and only these data can provide some clues on the differences in the performance.

10. The apparent Warburg component for W-MoS₂ can be just an onset of another, unresolved “semicircle”. Given that the Nyquist plot is not orthonormal (as it should be), it is also hard to validate the apparent 45deg slope.

11. “In comparison, the NO₂, -H, and NH₄⁺ signals were absent in the spectra of MoS₂ and W-MoS₂ (Figures S42-43).” – how ammonium is then formed on these catalysts?

12. The reviewer has failed to find examples of chronoamperograms. It is not even clear what was the duration of the experiments. These data should be exemplified in the main text for the most important experiments, and the rest of the data should be presented in the SI.

Figure S30 is critically important and should be shown in the main text.

13. Error bars are said to correspond to “relative standard deviations by three repeated tests”. While these repeated experiments are important, they are not sufficient to prove the reproducibility and reliability of the data. To prove that the trends are real, key materials (Ag-MoS₂ and MoS₂, at minimum) should be synthesised as three independent samples, each should be investigated, and the data should be reported as mean +/- standard deviation.

Reviewer #3

(Remarks to the Author)

This study explores the use of a p-type Ag-MoS₂ catalyst for the electrochemical nitrate reduction reaction (NO₃RR). Ammonia production via NO₃RR is a crucial topic for energy conversion and decarbonizing chemical manufacturing, as it plays a key role in balancing the global nitrogen cycle and reducing carbon emissions. While most previous works have studied on NO₃RR catalysts under relatively high nitrate concentrations, the Ag-MoS₂ catalyst presented here demonstrates decent performance under low nitrate concentrations, which is particularly advantageous for real wastewater treatment. However, the central claim that a p-type catalyst can enhance local nitrate concentration is not entirely reasonable, and several other concerns remain unaddressed in the manuscript. Therefore, this work is not recommended for publication in Nature Communications at least in its current form.

1. The central claim of this work—that a p-type catalyst can generate positive charges at the solid-liquid junction to attract nitrate ions, making it an effective catalyst for NO₃RR, especially under low nitrate concentrations—is highly questionable. While the authors have provided sufficient analysis showing that Ag-MoS₂ exhibits p-type characteristics and can produce more positive charges at the S-L junction compared to MoS₂ and W-MoS₂ under OCV, which I find convincing, this only implies that Ag-MoS₂ might attract more nitrate ions when all three catalysts are operated under the same reduction potential. However, this does not necessarily mean that Ag-MoS₂ will be a better NO₃RR catalyst. I would challenge the authors with the argument that by applying a less negative potential to MoS₂, it could achieve a similar charging environment in the IHP, thereby attracting the same amount of nitrate ions as Ag-MoS₂.

Therefore, the real question is: when all the catalysts are operated at their respective optimal potentials (where they can all attract sufficient nitrate ions for the reaction), which catalyst structure exhibits the lowest energy barrier in the 8-step electron transfer process of NO₃RR? This crucial point, however, is not discussed in the manuscript. The only relevant data provided is in Fig. 5e, but it does not include a comparison with the other two catalysts, nor does it address the optimal applied potentials required to attract sufficient nitrate or the energy barriers under those conditions.

If the authors agree with the idea above, a reasonable logical flow could be: 1) p-type/n-type catalysts create different charges at the solid-liquid junction, 2) therefore, the three catalysts require different applied biases to attract sufficient nitrate ions, 3) under the desired applied potentials, one catalyst (presumably Ag-MoS₂) has the lowest energy barrier for NO₃RR, making it the most effective catalyst among the three. The authors should restructure the whole manuscript and accordingly, with additional experimental and simulation details to support and validate these claims.

Some other concerns:

2. Relative to the points above, the manuscript does not clearly identify which step in the NO₃RR process is considered the rate-determining step.

3. In Fig. 3g, the authors compared the three catalysts only at the same voltage. It would be more informative to also test the other two catalysts across different potentials, as they may have different optimal voltages for nitrate absorption.

4. The authors provided only EXAFS data in Fig. 3f, while XANES analysis for the three catalysts is crucial for validating their electronic structures.

5. The Comsol simulation section lacks sufficient detail and explanation. Specifically, the inputs used to simulate the surface boundary parameters for the three catalysts are not provided.

6. The authors state that the Nyquist plot for W-MoS₂ “showed a straight line with a slope of 45°, suggesting that NO₃⁻ mass transfer was the rate-limiting step for NO₃RR.” This conclusion requires further clarification. Additionally, to confirm whether the plot exhibits a straight line rather than a large semicircle, a scan at lower frequencies should be conducted.

7. The authors state, “We observed a nearly constant reaction rate within a NO₃⁻-N concentration from 1400 µg/mL (100 mM) to 7.6 µg/mL (0.54 mM) (indicated by the slope of the cyan curve in Figure 4e), demonstrating the excellent performance of Ag-MoS₂.” This statement is ambiguous. Most studies suggest that NO₃RR follows a pseudo-first-order reaction and is

dependent on nitrate concentrations. If the reaction rate is constant, do the authors imply a zero-order reaction, suggesting that it may be independent of nitrate concentration? This conclusion should be examined more carefully.

Version 1:

Reviewer comments:

Reviewer #1

(Remarks to the Author)

After carefully reviewing the rebuttal and revisions, my concerns have grown further. The foundational aspects of the manuscript appear to be unreliable. Specifically, I have issues with the following key points: (1) the formation of band bending, (2) the misinterpretation of hole behavior and surface anion properties, and (3) the explanation of charge transfer direction and conduction mechanisms. I will outline my concerns in more detail below. Overall, I am not in agreement with the authors' proposed mechanisms or their key findings.

The authors report work functions of 4.98 eV for Ag-MoS₂ and 4.92 eV for MoS₂ using the Kelvin probe. However, they subsequently state that the Fermi levels of Ag-MoS₂ and MoS₂ are 1.07 V and 0.53 V vs. RHE, respectively. The discrepancy between these values of two samples is too large to be physically plausible. Additionally, the authors claim that the potential difference between the Fermi level (E_f) and the redox potential (E_{redox}) triggers charge transfer from the inner Helmholtz plane to the materials. This statement forms the cornerstone of their argument for band bending. However, in the Supporting Information, the authors assert that the Fermi level (E_F) of a p-type semiconductor is higher than the redox potential of NO₃⁻/NH₃ ($E_{redox} = 0.27$ V vs. RHE), suggesting electron transfer from the electrolyte to the surface of Ag-MoS₂ and MoS₂. This raises a critical question: how can NO₃⁻ donate electrons to Ag-MoS₂ and MoS₂? The authors do not address the underlying mechanism of this electron donation, and they overlook the importance of work function differences and the oxidation of NO₃⁻ in constructing the band bending.

In line 186, the authors use EIS to measure capacitance, which they then use to quantify charge distribution. However, they fail to clarify whether the capacitance arises from the depletion region, surface adsorption, or electrochemical reactions at -0.6 V vs. RHE. This is a crucial point that needs to be addressed for a proper understanding of their results.

In the rebuttal, the authors claim that the valence band of MoS₂ is primarily composed of S atom orbitals. This is incorrect. The valence band of MoS₂ is mainly formed by the 4d orbitals of Mo, not the orbitals of S atoms (see PHYSICAL REVIEW B, 2001, 64, 235305; Nature Communications 5, 4673 (2014)). The holes in the Mo 4d orbitals are responsible for conduction, while unsaturated edge S atoms contribute to surface states that narrow the bandgap at the interface, but they do not play a direct role in bulk conduction.

The authors suggest that surface sulfur (S) atoms may act as hole carriers, but they incorrectly describe them as positively charged. Surface S atoms are still negatively charged as anions and are not capable of becoming positively charged. While surface S may act as a hole carrier, it remains negatively charged and cannot be assigned a positive charge. The mechanism based on this erroneous assumption does not hold up.

The statement that "the downward band bending promotes hole transfer from bulk to surface" is fundamentally incorrect. Downward band bending would push holes toward the bulk, not to the surface. This misinterpretation of band bending further undermines the manuscript's core argument.

The authors mention a carrier concentration of 10¹⁸ cm⁻³, which typically places the material in a range from semiconductor to insulator. They then claim superior conductivity based on EIS measurements, but these results are largely influenced by carbon additives and do not address the underlying issue of poor electric conductivity. Additionally, this comparison is unfair, as the voltage is primarily applied to the depletion region, and EIS does not account for overpotential losses, which are critical in this context.

Finally, the authors should explicitly state the dielectric constant used in the calculation of carrier concentration in the experimental section. This is a crucial parameter that needs to be clearly defined.

Reviewer #3

(Remarks to the Author)

The authors have resolved the major concerns raised, and I recommend the manuscript for publication.

Reviewer #4

(Remarks to the Author)

This article presents a novel Ag-doped MoS₂ catalyst (Ag-MoS₂) for the efficient conversion of nitrate (NO₃⁻) to ammonia (NH₃). The authors suggest that doping with Ag enhances the p-type characteristics of MoS₂, leveraging its hole conduction

properties to generate more positive charges at the solid-liquid interface, which in turn attracts nitrate anions. This increased enrichment of reactants at the solid-liquid interface improves reaction efficiency, achieving a nitrate-to-ammonia conversion efficiency close to 100%. Three reviewers provided valuable feedback, offering insightful analyses and highly professional suggestions regarding the selection of experimental conditions, the reliability of results, the clarity of chart representations, and the experimental design. The authors addressed the reviewers' comments in detail and revised the article accordingly. After these revisions, the article is data-rich, logically coherent, and I recommend its publication. However, there are still a few minor points that need attention:

1. On page 11, the authors describe how they quantified the NO_3^- enrichment capacity based on methods outlined in the Supporting Information (SI). They measured the overall concentration of absorbed NO_3^- directly. However, aside from the enhanced adsorption capacity due to the p-type properties of the catalyst improved by Ag doping, other factors such as the electrochemical active surface area (ECSA) and specific surface area of the catalyst may also influence the nitrate adsorption capacity. I believe it is insufficient for the authors to rely solely on ion chromatography to detect adsorbed nitrate. It would be beneficial to also test and compare the specific surface area and ECSA of the different catalysts coated on carbon paper.
 2. The authors could strengthen their work by comparing their results with existing literature, especially with other high-performance NO_3RR catalysts. Including intuitive comparison charts would help highlight the innovations and advantages of this study.
 3. Regarding Comment 2 from Reviewer #2, the authors proposed that nitrate is transformed, and the resulting ammonium is recovered as a potentially valuable product. If the production of valuable ammonium is a primary application of this research, it would be helpful for the authors to discuss the economic value of ammonium, the cost of the catalyst, the scalability of the preparation method, and potential challenges in large-scale applications. Additionally, the authors noted that one of the key applications of this work is in treating agricultural and industrial wastewater. In light of this, it is recommended that the authors simulate conditions more representative of actual wastewater treatment or ammonia production, such as using more complex electrolyte compositions.
 4. Regarding Comment 3 from Reviewer #2, the reviewer pointed out that K_2SO_4 , which does not facilitate ion exchange, cannot effectively control the reaction conditions. The authors confirmed through verification experiments that the pH changes significantly before and after the reaction when K_2SO_4 is used as the electrolyte. In response, the authors suggested that the ionic effect and the good conductivity of SO_4^{2-} contribute to better performance, whereas PBS has poorer conductivity. However, could it be possible to prepare an electrolyte composed of both K_2SO_4 and PBS? By using PBS to control the pH during the reaction process, K_2SO_4 could potentially enhance the solution's conductivity.
 5. Regarding Comment 1 from Reviewer #3, although the authors have supplemented their analysis with data on the different adsorption capacities of the three samples for NO_3^- at various potentials, the core issue raised by Reviewer #3—the comparison of energy barriers for different catalysts at their respective optimal potentials—has not yet been fully addressed. The authors should provide a more detailed discussion of this issue, for example, by calculating the changes in reaction free energy at different potentials to clearly demonstrate the advantages of Ag-MoS₂.
- Aside from the few minor issues mentioned above, I believe the authors have responded to the reviewers' comments thoroughly. The additional data provided is reasonable and effective, and after making the minor revisions, this manuscript can be recommended for publication.

Version 2:

Reviewer comments:

Reviewer #1

(Remarks to the Author)

The authors addressed all the questions. It is ready for acceptance now.

Reviewer #4

(Remarks to the Author)

The authors have done all reviewer's comments. This manuscript can be considered for publication.

Manuscript number: NCOMMS-24-49780-T

Title: Near-Unity Nitrate to Ammonia Conversion via Reactant Enrichment at the Solid-Liquid Interface

A Point-to-Point Response to Reviewer's Comments

Dear Editor and Reviewers,

Thank you for taking time and effort to carefully examine our manuscript. The comments are highly appreciated and helpful to improve this work. We have incorporated the corresponding changes into the manuscript (**highlighted in the revised manuscript**) and supplementary information in order to address the editor's concerns and the requests of the reviewers. These changes are specified and discussed in a point-to-point response to the reviewers' comments, as shown below:

Reviewer #1:

The manuscript reports a method to construct a Ag-doped MoS₂ catalyst for nitrate reduction. Because the band bending at interface, a negative charge accumulation at the catalyst surface, which induces cations at Helmholtz layers. However, through the manuscript, there is several contradictory descriptions about what type of charges are accumulated at Helmholtz layer. The authors' aim is to induce an increase in the NO₃⁻ concentration. Many technologies are used to confirm that. However, the authors failed to relate the excess charge to anions. The physical mechanism is not stated clearly. Only a COMSOL simulation is provided to demonstrate the enrichment of NO₃⁻. In particular, from the Line 167 where the authors stated the increase in positive charge at IHP to the experiments in the following two paragraph, no discussion and explanation is provided. In addition, there are many errors in the manuscript. I would suggest a major revision.

Response: Thank you very much for your detailed comments, which played a crucial role in improving the quality of our paper. In response to the raised questions, we have made extensive revisions to the article, including **1)** revised the schematic diagrams related to the species distribution in IHP, **2)** supplemented the XANES and KPFM results as well as the COMSOL details, **3)** re-performed the EIS and LSV tests, **4)** and provided the explanations about the working principle and the selection of p-type 2D MoS₂ catalyst as the cathode for NO₃RR. We hope our modifications can fulfill the requirements of the reviewer.

Comment 1: The authors claim extensive free electrons packed on its IHP. The statement is wrong. There are no free electrons in IHP. ions can usually appear at IHP. In the abstract, “Particularly, at the reduction potential, an excess of free electrons accumulates on the inner Helmholtz plane (IHP), the solution side of the double layer where catalytic reactions occur”. The statement of the authors is wrong. There is no excess of free electrons but excess charge. Free electrons are usually not able to exist in an aqueous electrolyte. In addition, “free electrons” are used for metals or in vacuum. At the electrode/electrolyte interfaces, the electrodes acquire more electrons upon negative polarization and induce the contact adsorption of partially dissociated cations at atomic length. Away from the surface of electrodes, there are outer Helmholtz plane(OHP) and diffusive layer, respectively.

Response: Thanks so much for your constructive comments. We agree with your opinion that there is no excess of free electrons in an aqueous electrolyte. The questions raised by the reviewer are answered from the following three parts:

a) Surface electron distribution on conductor materials. As the reviewer mentioned, under negative polarization, abundant electrons would accumulate on the traditional conductor material surface. Thus, at the electrode/electrolyte interface, the long-range electrostatic interaction could induce the contact adsorption of cations (*J. Am. Chem. Soc.* 2021, 143, 14158–14168).

b) Hole conduction characteristic of p-type catalyst. To achieve the anion attraction on cathode under negative polarization, the p-type semiconductor catalysts are taken into consideration in this work. First, unlike the traditional conductor materials (with abundant free electrons), the p-type catalysts have the characteristics of hole conduction (*Adv. Mater.* 2023, 35, 2206939; *Adv. Mater.* 2018, 30, 1706262.). Usually, the Fermi level of p-type catalysts is located near the valence band maximum. During the contact between p-type semiconductor and solution, interfacial charge transfer can induce downward band bending and built-in electric field on catalyst side, based on the potential difference between Fermi level and redox pair ($\text{NO}_3^-/\text{NH}_3$), to promote the hole transfer from bulk catalyst to surface.

c) The reason for choosing 2D MoS₂. Moreover, typical p-type MoS₂ exhibits unique two-dimension (2D) structure, which expose abundant edge S sites (*ACS Catal.* 2022, 12, 1, 8-17; *Energy Environ. Sci.* 2017, 10, 593-603). As the valence band of MoS₂ is mainly composed of the orbits of S atom, these unsaturated coordination edge S sites act as the traps to capture holes (as the free carriers) on the surface of MoS₂. When contacting with electrolyte, abundant positive holes distribute on the IHP region, and the downward band bending promotes the hole transfer from bulk to surface. Even under negative polarization, the remained positive holes within IHP can attract NO_3^- anion (Figure S10).

Besides, the adsorption energy of NO_3^- on optimized Ag-MoS₂ is -3.0 eV (Figure 5e), which favors the formation of chemical bonds at atomic length. These are the reasons for selecting Ag-MoS₂ as the cathode material for NO_3^- enrichment, to achieve superior NO₃RR performance under low concentration condition.

Thus, to better illustrate the selection of Ag-MoS₂ catalysts, we have corrected the description related to “free electrons”, modified corresponding description, added the characteristics of p-type Ag-MoS₂ catalysts, and revised some schematic diagrams (Figures 1, 3h, S10-11). We hope these changes can satisfy and address the reviewer’s concerns.

Figure S10. Interfacial thermal equilibrium process when the Ag-MoS₂ contact with the NO₃⁻ contained solution. Downward band bending at solid-liquid (S-L) junction under (b) no potential and (c) reduction potential. E_{redox} and E_F represent the theoretical redox potential of NO₃⁻/NH₃ and the Fermi level potential. CBM and VBM express the conduction band minimum and valence band maximum of semiconductor, respectively.

Figure 5e. Gibbs free energy diagram of various intermediates generated during NO₃RR over Ag-MoS₂.

Figure S11. (a) Interfacial thermal equilibrium process when MoS₂ contact with the NO₃⁻ contained solution. Downward band bending at solid-liquid (S-L) junction under (b) no potential and (c) reduction potential.

Figure 1. Schematic diagram of solid-liquid (S-L) junction-mediated NO_3^- enrichment within IHP region over Ag-MoS₂ at the reduction potential. E_{redox} represents the theoretical redox potential of $\text{NO}_3^-/\text{NH}_3$. CBM and VBM express the conduction band minimum and valence band maximum of Ag-MoS₂, respectively. Note: IHP contains surface trapped holes, solvate molecules, and NO_3^- anions, and cations. Electrons accumulate on the electrode surface under negative polarization. Ionized acceptors exist in the S-L junction (see detailed discussions in supplementary discussions).

Figure 3h. Schematic diagram of NO_3^- enrichment on IHP region over different catalysts at the reduction potential.

Corresponding revision:

- Figure 1 has been revised (Page 6, Revised manuscript).
- Figure 3h has been revised (Page 13, Revised manuscript).
- Figures S10-11 have been revised (Pages 19-20, Revised supplementary information).
- The corresponding content in the abstract has been revised to “However, the activity of this process is limited by NO_3^- mass transfer, particularly at reduction potential,

where an abundance of electrons on the cathode surface repels NO_3^- anions from the inner Helmholtz plane (IHP).” (Page 2, Revised manuscript).

➤ The corresponding content in the abstract has been revised to “Specifically, during NO_3^- reduction, the formation of the S-L junction induces hole transfer from p-type Ag-doped MoS_2 (Ag-MoS_2) to electrode/electrolyte interface, triggering abundant positive charges on the IHP to attract NO_3^- .” (Page 2, Revised manuscript).

➤ The corresponding content in the introduction has been revised to “Particularly, at the reduction potential, an excess of electrons accumulates on the cathode surface, leading to a repulsion of negatively charged NO_3^- ions from the inner Helmholtz plane (IHP), where the catalytic reactions occur²⁷.” (Pages 4-5, Revised manuscript).

➤ The corresponding content “suggesting extensive anions packed on its IHP” has been added (Page 10, Revised manuscript).

➤ The corresponding content “P-type MoS_2 exhibited the characteristic of hole conduction, and the 2D structure endowed abundant edge S sites as traps to capture positive holes⁶⁰⁻⁶². Thus, the positive charges distributed within IHP can be ascribed to the surface trapped holes.” has been added (Page 10, Revised manuscript).

➤ The corresponding content “The p-type 2D MoS_2 exhibits the characteristic of hole conduction, its abundant edge S sites can act as traps to capture holes. When immersing the p-type semiconductor Ag-MoS_2 into the NO_3^- -contained solution, the potential difference between the E_F of Ag-MoS_2 and the E_{redox} induces the hole transfer from bulk catalyst to the surface S sites (Figure S10a).” has been added (Page 19, Revised supplementary information).

***Comment 2:** A p-type semiconductor is used as the cathode. Upon negative polarization, the conductivity decreases dramatically, which is not favorable for electrocatalysis.*

Response: We thank the reviewer for the comments, and the questions raised by the reviewer are answered from the following two parts:

a) The great conductivity of 2D MoS_2 catalyst. We agree with the reviewer’s opinion that the conductivity of some p-type semiconductors is poor, which can be ascribed to

the wide band gap and low carrier concentration. While, for two-dimensional MoS₂ catalyst, the narrow band gap (~1.3 eV) and high carrier concentration (~10¹⁸ cm⁻³) jointly contribute to the excellent conductivity (*Nano Lett.* 2014, 14, 8, 4628–4633; *Nat. Commun.* 2015, 6, 7666. *Nano Res.* 2021, 14, 2255–2263). In the EIS analysis, the fitted R_s of MoS₂, Ag-MoS₂, and W-MoS₂ are as low as 4.25, 4.18, and 4.21 $\Omega\text{ cm}^{-2}$, verifying their superior conductivity (Table S6). Besides, the p-type MoS₂ has been widely applied as cathode for electrocatalysis including HER, NRR, ORR, and CO₂RR etc (*Adv. Funct. Mater.* 2017, 27, 1702300.).

Table S6. Fitting results for resistances of various catalysts at -0.6 V *versus* RHE in a 0.5 M K₂SO₄ with 10 mM NO₃⁻ electrolyte according to EIS equivalent circuit diagram.

Resistance Sample tests	R_s ($\Omega\text{ cm}^{-2}$)	R_{bulk} ($\Omega\text{ cm}^{-2}$)	R_{ct} ($\Omega\text{ cm}^{-2}$)	Warburg ($\Omega\text{ cm}^{-2}$)
W-MoS ₂	4.21 (0.71%)*	4.80 (0.88%)	47.5 (3.83%)	45.47 (1.96%)
MoS ₂	4.25 (0.76%)	4.83 (0.62%)	15.41 (4.25%)	/
Ag-MoS ₂	4.18 (0.45%)	4.89 (0.59%)	6.04 (3.17%)	/

* The content in parentheses after each value represents the relative standard deviation of the fitting result.

b) Hole conduction characteristic for attracting NO₃⁻. In addition to the excellent conductivity, the typical 2D p-type Ag-MoS₂ exhibits abundant unsaturated coordinated edge S sites (Figure R1). As the p-type semiconductors have the feature of hole (hole as the positive charge) conduction, the edge S sites can act as the traps or surface states to capture holes (*Sci. Rep.* 2021, 11, 3958; *ACS Appl. Mater. Interfaces* 2018, 10, 12, 10580–10586; *Nano Lett.* 2017, 17, 12, 7962–7967). These holes bounded by the surface states may exist in the outermost layer of the catalyst and be in contact with the electrolyte.

Figure R1. Schematic diagram of planar atomic structure of two-dimensional MoS₂. In which the orange hotspot area indicates the holes gathered on the edge S.

Meanwhile, the Fermi level potential of Ag-MoS₂ (1.06 V *vs.* RHE) is much higher than the redox potential of NO₃⁻/NH₃ pair (0.27 V *vs.* RHE), and this potential difference drives the hole transfer from inner Ag-MoS₂ to the surface, which intensified the accumulation of trapped holes on the electrode/electrolyte interface (Figure S10). Thus, the abundant holes can attract the NO₃⁻ anion under the electrostatic interaction, to favor the enrichment of reactant NO₃⁻. Thus, we consider that the selection of p-type Ag-MoS₂ as the cathode for NO₃RR is reasonable.

Figure S10. Interfacial thermal equilibrium process when the Ag-MoS₂ contact with the NO₃⁻ contained solution. Downward band bending at solid-liquid (S-L) junction under (b) no potential and (c) reduction potential. E_{redox} and E_{F} represent the theoretical redox potential of NO₃⁻/NH₃ and the Fermi level potential. CBM and VBM express the conduction band minimum and valence band maximum of semiconductor, respectively.

Corresponding revision:

➤ The corresponding content “The typical 2D p-type Ag-MoS₂ exhibits abundant unsaturated coordinated edge S sites. As the p-type semiconductors have the feature of hole (hole as the positive charge) conduction, the edge S sites can act as the traps or

surface states to capture holes (*Sci. Rep.* 2021, 11, 3958; *ACS Appl. Mater. Interfaces* 2018, 10, 12, 10580–10586; *Nano Lett.* 2017, 17, 12, 7962–7967). These holes bounded by the surface states may exist in the outermost layer of the catalyst and be in contact with the electrolyte. Meanwhile, the Fermi level potential of Ag-MoS₂ (1.06 V vs. RHE) is much higher than the redox potential of NO₃⁻/NH₃ pair (0.27 V vs. RHE), and this potential difference drives the hole transfer from inner Ag-MoS₂ to the surface, which intensified the accumulation of trapped holes on the electrode/electrolyte interface. Thus, the abundant holes can attract the NO₃⁻ anion under the electrostatic interaction, to favor the enrichment of reactant NO₃⁻.” has been added (Page 9, Revised supplementary information).

Comment 3: *Figure 2f shows the EXAFS spectra of Ag-MoS₂ and reference samples. However, the K-edge XANES spectra are not shown. Please provide and discuss them.*

Response: Thanks for raising the concern about the XANES spectra. We have supplemented the data and added the corresponding description in the manuscript.

The Mo K-edge X-ray absorption near edge structure (XANES) spectra exhibit the higher pre-edge absorption energy of Ag-MoS₂ than those of Mo foil and MoS₂ (Figure S6a), implying that the doping Ag species induces decreased electron density on the Mo site. From the Ag K-edge XANES spectra, the absorption energy of Ag-MoS₂ located between those of Ag foil and Ag₂S references (Figure S6b), indicating the valence state of the doped Ag within 0 to 1. Thus, there is a significant electronic interaction between the introduced Ag species and MoS₂.

Figure S6. (a) Mo K-edge XANES spectra and (b) Ag K-edge XANES spectra of Ag-MoS₂ and reference samples.

Corresponding revision:

- **Figure S6** has been added (Page 15, Revised supplementary information).
- The corresponding content has been added “The Mo K-edge X-ray absorption near edge structure (XANES) spectra exhibited the higher pre-edge absorption energy of Ag-MoS₂ than those of Mo foil and MoS₂ (Figure S6a), implying that the doping Ag species induced decreased electron density on the Mo site. From the Ag K-edge XANES spectra, the absorption energy of Ag-MoS₂ located between those of Ag foil and Ag₂S references (Figure S6b), indicating the valence state of the doped Ag within 0 to 1.” (Page 8, Revised manuscript).

Comment 4: The work function of Ag-doped MoS₂ is 5.47eV which is greater than that of MoS₂ (4.98V). Previous studies show that precious metal doped MoS₂ induced the decrease in working function (Chem Mater 2019;31(2):429). Please explain the reason. It should be confirmed by Kelvin probe force microscope. Doping induces mid-gap states which decrease the work functions.

Response: Thanks so much for your comment. The questions raised by the reviewer are answered by the following three parts:

a) First, the low-valence state Ag atom can act as the electron acceptor, to enhance the hole concentration and thus the p-type characteristic of the semiconductors (*Nano Energy* 2018, 46, 212-219; *Nanoscale* 2018, 10, 15980-15988; *Sci. Rep.* 2014, 4, 6627). In the density of states (DOS) results (**Figure S56**), the Ag dopant induces significant band-tail states near the valence band maximum (VBM), which lead to the downward shift of the Fermi level close to the VBM. From the M-S plots (**Figure 3a**), a relatively low slope of Ag-MoS₂ further indicates its higher carrier concentration. After calculation, the carrier concentration of Ag-MoS₂ is $1.14 \times 10^{19} \text{ cm}^{-3}$ (**Table S2**), 1.8 times higher than that of pure MoS₂ ($6.3 \times 10^{18} \text{ cm}^{-3}$).

In addition, the flat band potential obtained from the M-S plot can reflect the Fermi level position. Clearly, the flat band potential of Ag-MoS₂ is 1.06 V vs. RHE, much

higher than that of MoS₂ (0.53 V vs. RHE). A higher flat band potential suggested that the Fermi level position is downward shifted and close to VBM. Thus, the tested work function of Ag-MoS₂ is higher than that of pure MoS₂ in the present study.

Figure 3a. Mott-Schottky plots of Ag-MoS₂, MoS₂, and W-MoS₂ catalysts.

Figure S56. The partial density of states (PDOS) of (a) Ag-MoS₂, (b) MoS₂ and (c) W-MoS₂.

Table S2. Calculated carrier concentration of the three samples by M-S plots.

Samples	Carrier concentration (cm ⁻³)
W-MoS ₂	3.25×10 ¹⁸
MoS ₂	6.3×10 ¹⁸
Ag-MoS ₂	1.14×10 ¹⁹

b) We have carefully studied the literature provided by the reviewer (*Chem. Mater.* 2019, 31(2): 429), in which the author investigated the influence of precious metal single atoms (SA) on the work function of 2D MoS₂. According to their KPFM results, compared with MoS₂, the work function of Pt_{SA}-MoS₂ is slightly decreased from 4.635 to 4.61 eV. This is attributed to the extra electrons donation from Pt_{SA}. While, different

from the effect of Pt_{SA}, the Ag dopant act as the electron acceptor to increase the hole concentration and the p-type response of MoS₂, which have been confirmed by UPS, M-S plots and DOS results.

c) In addition, according to the reviewer's suggestion, we have supplemented the KPFM test of MoS₂ and Ag-MoS₂ catalysts. The prepared samples were dispersed on conductive FTO substrate for test. As seen from Figures S8-S9, the potential difference between bulk MoS₂ and FTO substrate is 20 mV, while 80 mV between Ag-MoS₂ and FTO substrate. As the work function of FTO substrate is 4.9 eV (*Sci. Rep.* 2016, 6, 20399; *ACS Appl. Mater. Interfaces* 2014, 6, 8, 5367–5373), the work functions of MoS₂ and Ag-MoS₂ measured by KPFM technique are 4.92 eV and 4.98 eV, respectively. Combined with UPS, M-S plot, KPFM experiments and DOS theoretical calculation, it is reasonable that Ag-MoS₂ exhibits a relatively high work function than that of MoS₂.

Figure S8. (a) Surface potential and (b) corresponding line profiles along the arrow lines of MoS₂.

Figure S9. (a) Surface potential and (b) corresponding line profiles along the arrow lines of Ag-MoS₂.

Corresponding revision:

- **Figure 3a** has been revised (Page 13, Revised manuscript).
- **Table S2** has been added (Page 71, Revised supplementary information).
- The corresponding content “The calculated carrier concentration of Ag-MoS₂ is $1.14 \times 10^{19} \text{ cm}^{-3}$, 1.8 times higher than that of MoS₂ ($6.3 \times 10^{18} \text{ cm}^{-3}$, Table S2). Thus, the Ag dopant as the electron acceptor to increase the carrier concentration and work function of MoS₂ was verified.” has been added (Page 9, Revised manuscript).
- The corresponding content “Mott-Schottky measurements were carried out at the frequency of 1000 Hz under dark condition to obtain the flat band potential and carrier concentration of the samples.” has been added (Page 23, Revised manuscript).
- **Figures S8-9** have been added (Pages 17-18, Revised supplementary information).
- The corresponding content “In addition to the UPS method, KPFM measurements were carried out to collaboratively demonstrative the increase of work function by Ag doping (Figures S8-9)⁴⁹. The surface potential difference between Ag-MoS₂ and FTO substrate was 80 mV, while 20 mV for MoS₂. The work functions of Ag-MoS₂ and MoS₂ were 4.98 and 4.92 eV. The variation trend was in consistent with UPS results, indicating a relatively high work function of Ag-MoS₂.” has been added (Page 9, Revised manuscript).
- The corresponding content “The calculated carrier concentration of Ag-MoS₂ is $1.14 \times 10^{19} \text{ cm}^{-3}$, 1.8 times higher than that of MoS₂ ($6.3 \times 10^{18} \text{ cm}^{-3}$, Table S2). Thus, the Ag dopant as the electron acceptor to increase the carrier concentration and work function of MoS₂ was verified.” has been added (Page 9, Revised manuscript).
- The corresponding content “The Kelvin probe force microscopy (KPFM) experiment was conducted under room temperature and atmospheric pressure. A Bruker icon instrument with an SCM-PIT test probe was utilized. The samples on FTO were scanned at a rate of 0.5 Hz with an image size of 256×256 . The testing was performed on a vibration isolation table to ensure the accuracy of the results.” has been added (Page 8, Revised supplementary information).

***Comment 5:** Figure 1 shows all negative ions at IHP, which is not correct. Because the surface of electrodes accumulates electrons under negative potentials, which polarize electrolytes and attract water molecule dipole or cations at IHP. From OHP to diffuse layer, the concentration of excess counter-ions decreases and approaches to the bulk concentration. According to the classic model, the IHP usually are water molecules or counter-ions of excess charge at electrode surface. Negative polarization leads to the accumulation of electrons at the surface of cathodes, which prefers to attract cations toward cathodes. In some cases, there may be some contact adsorption of cations, water, and anions at IHP. The contact adsorption mainly depends on the chemical natures of adsorbents rather than charge, which is called superequivalent adsorption. As far as the absolute values are concerned, the excess charge at electrode (Q_{ec}) may be less than that of contact adsorption (Q_{ca}). The charge at OHP (Q_{ohp}) is less than the total excess charge of solution from IHP to bulk (Q_s), which is equal to Q_{ec} .*

Response: We sincerely thank the reviewer for the constructive comments. We agree with your opinion that the distribution of all negative ions on IHP is not correct. Our original intention is to more intuitively display the difference between solid and liquid phases. While, other species which should distribute in IHP have been overlooked. Thus, according to your suggestion, we have modified Figure 1.

Specifically, all possible species within S-L junction and IHP region, including surface trapped holes, ionized acceptors, free electrons on catalyst surface, solvate molecules, and cations, are added in the new version of Figure 1. For traditional conductor electrocatalysts, free electrons packed on the catalyst surface under negative polarization, leading to the attraction of cations and the repulsion of the NO_3^- reactant significantly.

Unlike traditional catalysts, p-type MoS_2 exhibited the characteristic of hole conduction, and its abundant edge active S sites can act as the trap to capture the free holes in MoS_2 . Under this condition, the positive holes could accumulate on the interface between MoS_2 and electrolyte. A more positive Fermi level of Ag- MoS_2 could

cause an increased difference between the Fermi level of Ag-MoS₂ and the redox potential of NO₃⁻/NH₃, leading more holes transferred from bulk MoS₂ to surface and being captured by edge S sites. Thus, compared to non p-type catalyst with classic double layer model, the positive holes of Ag-MoS₂ can attract more NO₃⁻ for NO₃RR.

Also, all the corresponding schematic diagrams have been revised and updated (Figures 1, 3h, and S10-11).

Figure 1. Schematic diagram of solid-liquid (S-L) junction-mediated NO₃⁻ enrichment within IHP region over Ag-MoS₂ at the reduction potential. E_{redox} represents the theoretical redox potential of NO₃⁻/NH₃. CBM and VBM express the conduction band minimum and valence band maximum of Ag-MoS₂, respectively. Note: IHP contains surface trapped holes, solvate molecules, and NO₃⁻ anions, and cations. Electrons accumulate on the electrode surface under negative polarization. Ionized acceptors exist in the S-L junction (see detailed discussions in supplementary discussions).

Figure 3h. Schematic diagram of NO₃⁻ enrichment on IHP region over different catalysts at the reduction potential.

Figure S10. (a) Interfacial thermal equilibrium process when the Ag-MoS₂ contact with the NO₃⁻ contained solution. Downward band bending at solid-liquid (S-L) junction under (b) no potential and (c) reduction potential. E_{redox} and E_F represent the theoretical redox potential of NO₃⁻/NH₃ and the Fermi level potential. E_C and E_V express the conduction band minimum and valence band maximum of semiconductor, respectively.

Figure S11. (a) Interfacial thermal equilibrium process when MoS₂ contact with the NO₃⁻ contained solution. Downward band bending at solid-liquid (S-L) junction under (b) no potential and (c) reduction potential.

Corresponding revision:

- Figure 1 has been revised (Page 6, Revised manuscript).
- Figure 3h has been revised (Page 13, Revised manuscript).
- Figures S10-11 have been revised (Pages 19-20, Revised supplementary information).
- The corresponding content “The typical 2D p-type Ag-MoS₂ exhibits abundant unsaturated coordinated edge S sites. As the p-type semiconductors have the feature of hole (hole as the positive charge) conduction, the edge S sites can act as the traps or surface states to capture holes (*Sci. Rep.* 2021, 11, 3958; *ACS Appl. Mater. Interfaces*

2018, 10, 12, 10580–10586; *Nano Lett.* 2017, 17, 12, 7962–7967). These holes bounded by the surface states may exist in the outermost layer of the catalyst and be in contact with the electrolyte. Meanwhile, the Fermi level potential of Ag-MoS₂ (1.06 V vs. RHE) is much higher than the redox potential of NO₃⁻/NH₃ pair (0.27 V vs. RHE), and this potential difference drives the hole transfer from inner Ag-MoS₂ to the surface, which intensified the accumulation of trapped holes on the electrode/electrolyte interface. Thus, the abundant holes can attract the NO₃⁻ anion under the electrostatic interaction, to favor the enrichment of reactant NO₃⁻.” has been added (Page 9, Revised supplementary information).

Comment 6: *Figure S24 show very different concentration distribution for three samples. However, No range normal to the surface was indicated. It is hard to have an idea of length. Are they at equilibrium or please indicate the time at which the distribution was shown.*

Response: Thank you for the kind suggestions. We have added a scale bar in the updated Figure S28 to indicate the distance. The presented simulation results are based on the steady-state solver. In addition, according to reviewer #3’s suggestion (Comment 5), we have re-performed the COMSOL simulation. Specifically, the NO₃⁻ concentration on Ag-MoS₂ surface is still higher than those of MoS₂ and W-MoS₂ counterparts, which matches the experimental results demonstrated.

Figure S28. The NO₃⁻ distribution at the solid-liquid interface of (a) Ag-MoS₂, (b) MoS₂, and (c) W-MoS₂ catalysts through COMSOL simulations.

Corresponding revision:

➤ Figure S28 has been updated (Page 37, Revised supplementary information).

➤ The corresponding content has been revised to “The presented simulation results are based on the steady-state solver.” (Page 4, Revised supplementary information).

Comment 7: In line 190, severe downward shift of E_f into the valence band of Ag-MoS₂ (Figure S23). The figures are not related to the description. The authors used the diameter of semicircle to derive that negative polarization leads to the formation of degenerate semiconductor. However, the voltage is too low too negative. Is that the bubbling effects which stirred the solution.

Response: We thank the reviewer for the comment. To address the reviewer’s concerns, we have supplemented the EIS fitting results and the schematic diagram of the degenerate semiconductor.

a) EIS analysis at different potentials is performed, to illustrate the optimal NO₃⁻ adsorption at the potential of -0.6 V vs. RHE. As seen from the Figure S27a, the semicircle at high-frequency region can be ascribed to the electron transfer within the electrocatalyst, while the semicircle at low-frequency region can reflect the electron transfer information at electrode/electrolyte interface (*Appl. Catal. B* 2017, 218, 570-580; *ACS Catal.* 2015, 5, 5292-5300.). To better support the conclusion, the fitting results for resistance under different potentials are supplemented (Table S3). At the potential of -0.6 V, the smallest arc radius at low-frequency region suggested the fastest NO₃RR rate.

When increasing the potential to -0.8 and -1.0 V, we can see that the arc radius at high-frequency region become to decrease, suggesting improved conductivity of Ag-MoS₂. This can be ascribed to the excessive potential that triggered severe downward shift of E_F into the valence band, forming degenerate semiconductor with metallic property. Thus, the NO₃RR adsorption peaks at the potential of -0.6 V.

b) To better illustrate the band structure of degenerate semiconductor, we have added Figure S27b, in which the Fermi level is within the valence band. In addition, all the EIS tests were performed under static condition with no stirring. Thus, we consider that the influence of bubbling effect can be excluded.

Figure S27. (a) Nyquist plots of Ag-MoS₂ catalyst at different applied potentials from -0.2 to -1.0 V *versus* RHE. (b) Schematic diagram of the band structure for Ag-MoS₂ at excessive reduction potential (-0.8 and -1.0 V *vs.* RHE), which have become the degenerate semiconductor with metallic property.

Table S3. Fitting results for resistances of various catalysts at -0.6 V *versus* RHE in a 0.5 M K₂SO₄ with 10 mM NO₃⁻ electrolyte according to EIS equivalent circuit diagram.

Potential (V vs. RHE)	R_s (Ω cm ⁻²)	R_{bulk} (Ω cm ⁻²)	R_{ct} (Ω cm ⁻²)
-0.2	4.20	4.78	30.59
-0.4	4.08	4.83	18.73
-0.6	4.18	4.89	6.04
-0.8	4.06	3.69	10.91
-1.0	4.15	3.14	19.35

Corresponding revision:

- Figure S27 has been revised and updated (Page 36, Revised supplementary information).
- Table S3 has been supplemented (Page 72, Revised supplementary information).
- The descriptions have been revised and updated “However, excessive potential (above -0.6 V) triggered the shift of E_F into the valence band of Ag-MoS₂ (Figure S27b), forming a degenerate semiconductor. In Figure S27a, the decreased arc radius at the high-frequency region suggested promoted conductivity under the potential of -0.8 and -1.0 V, indicating their metallic properties. As a result, the hole transfer from inner catalyst to the surface would be broken, thus hindering the mass transfer of NO₃⁻ from bulk electrolyte to the IHP. The phenomena had been confirmed by the increased arc

radius under the potential of -0.8 and -1.0 V at low-frequency region (Table S3). Thus, the NO_3^- concentration peaked under the potential of -0.6 V *versus* RHE.” (Page 36, Revised supplementary information).

Comment 8: *Figure S29 show the comparison with and without NO_3^- . The LSV is like the HER. There should be a well-designed control to indicate what ratio of the HER occurred? By decreasing the concentration of NO_3^- , you should find a diffusion-limited peak which is the sign of NO_3RR . The straight LSV lines do not tell the truth.*

Response: Thanks so much for your constructive comments. We agree with your opinion that the present LSV curves is more like the HER, which lacks the reduction information of NO_3^- .

Thus, to obtain the reduction peak of NO_3^- in the LSV test, we have re-performed the tests with NO_3^- concentration of 1 mM and scan rate of 1 mV s^{-1} . The updated data are shown in Figure S33. Obvious reduction peaks of all the samples emerges at around -0.27 V vs. RHE, ascribing to the diffusion-limited peak which is the sign of NO_3RR . Compared with counterparts, the highest peak current density of Ag-MoS₂ indicated a fastest surface NO_3RR kinetics, due to the superior NO_3^- enrichment effect.

Thus, we have revised and updated Figure S33, and added corresponded descriptions, to better illustrate NO_3RR process.

Figure S33. LSV curves for catalysts in electrolytes with and without 1 mM NO_3^- at the scan rate of 1 mV s^{-1} .

Corresponding revision:

- Figure S33 has been revised and updated (Page 42, Revised supplementary information).
- The descriptions have been added “To better illustrate the diffusion-limited information NO₃RR, LSV measurements were performed with the NO₃⁻ concentration of 1 mM and the scan rate of 1 mV s⁻¹. From the applied potential from 0.25 to -1.9 V vs. RHE, obvious reduction peaks of all the samples emerges at around -0.27 V vs. RHE, ascribing to the diffusion-limited peak which is the sign of NO₃RR. Compared with counterparts, the highest peak current density of Ag-MoS₂ indicated a fastest surface NO₃RR kinetics, due to the superior NO₃⁻ enrichment effect.” (Page 42, Revised supplementary information).

Comment 9: Figure 3e indicates a y-axis title of “peal”, which may be “peak”. Figure 3c has an x-axis title of Potential?

Response: We thank the reviewer for the meticulous reading and have revised the corresponding part of the manuscript.

Corresponding revision:

- The y-axis title of “peal” in Figure 3e has been revised to “peak” (Page 13, Revised manuscript).

Figure 3. (e) Corresponding peak intensity of absorbed NO₃⁻ in Raman spectra of catalysts at various potentials.

➤ The x-axis title of “Potential” in Figure 3c has been deleted (Page 13, Revised manuscript).

Figure 3. (c) The fitted surface capacitance of the three catalysts.

We hope that our efforts fulfilled the requirements of the reviewer. Thanks so much again for your constructive comments that helped us improving the quality of the manuscript.

Reviewer #2:

The manuscript by Liao et al. presents an interesting hypothesis on the design of electrocatalysts for electroreduction of nitrate to ammonia. However, the presented data do not seem to provide a robust confirmation of the authors' interpretations. Moreover, the overall experiment design is questionable. Under these circumstances, this manuscript cannot be recommended for publication.

Response: Thanks so much for the reviewer's comment. Your comments play a crucial role in improving the quality of our paper. In response to the raised questions, we have made extensive revisions to the article, including **1**) rewritten the background and the significance of our work, **2**) supplemented series control experiments (influence of Ag nanoparticles and buffer solution on the activity, additional electrochemical tests), **3**) added corresponded discussions (explanations of ACV, CV, and EIS results) within the whole article. We hope our modifications can fulfill the requirements of the reviewer.

Comment 1: Introductory explanations of the importance of the nitrate-to-ammonia reduction technology should be revised, as this process cannot be considered neither as a sustainable source of ammonia nor as an alternative to the Haber-Bosch process in the way it is presented [10.1038/s41929-023-01060-w].

Response: Thanks for raising the question about the statement related to the importance of the nitrate-to-ammonia reduction technology. We modified the statement according with the information suggested. We have added more related information and cited the reference (10.1038/s41929-023-01060-w, #9).

As proposed in this authoritative article, the availability in global NO_3^- resources is insufficient, and there are obstacles such as limited NO_3^- quantity, low concentration, and separation, which would have a significant effect on ammonia production. Considering the substantial global demand, the Haber–Bosch process is still primary ammonia production route in the near future. Hence, we agree with the reviewer's opinion that nitrate-to-ammonia electroreduction technology cannot currently be considered as a sustainable source of ammonia or as an alternative to the Haber-Bosch

process. We have revised the corresponding descriptions and cited the reference (10.1038/s41929-023-01060-w, #9) in the introduction part.

On the other hand, continuous research is crucial to explore the possible approaches to meet the diverse requirements of ammonia production in the future energy landscape. Electrocatalytic methods emerge the unique advantages, including operation at room temperature, flexible scalability to various infrastructure sizes and the ability to be powered by decentralized renewable electricity. Currently, various electrochemical pathways for ammonia synthesis have been explored, including electrochemical N₂ reduction reaction (NRR), lithium-mediated NRR and nitrate reduction reaction (NO₃RR). Among them, the present NH₃ yield via NO₃RR surpasses those of NRR by two to three orders of magnitude, mainly due to the relatively lower dissociation energy of the N=O bond (204 kJ mol⁻¹) of nitrate anion (NO₃⁻) compared to the N≡N bond (941 kJ mol⁻¹). Besides, NO₃⁻ exhibits a ubiquitous presence within contaminated groundwater and industrial effluent, with their availability further augmented through the oxidation process of atmospheric N₂ as the source of nitrogen. Consequently, the NO₃RR pathway is clean route for renewable NH₃ production. We have also revised and added more related information in the introduction part.

Corresponding revision:

➤ The corresponding content in the introduction has been revised to “Ammonia (NH₃), one of the most common industrial chemicals, is crucial for the production of agricultural fertilizers and holds immense potential as a green hydrogen-rich fuel¹⁻³. The current global ammonia demand exceeds 180 million tons annually, primarily fulfilled through industrial scale production of energy intensive Haber-Bosch (H-B) process^{1,4}. Within this process, steam-reformed hydrogen (H₂) undergoes reaction with nitrogen (N₂) under elevated temperature (~500 °C) and pressure (>100 atm)^{5,6}, which not only accounts for approximately 1.4% of global carbon dioxide (CO₂) emissions, but also necessitates the consumption of 2% of the world's total energy supply^{7,8}. Recently, electrocatalytic methodologies have surfaced as a viable clean energy

pathway for decentralized ammonia synthesis at room temperature, accommodating a range of infrastructure scales and potentially powered by locally sourced renewable energy sources^{9,10}. Despite the substantial global demand that may sustain the traditional ammonia production route in the near future, the electrochemical ammonia synthesis can be promising complementary process to the Haber–Bosch technology for contributing to decarbonizing ammonia production^{11,12}.

Recently, diverse electrochemical approaches have been explored to address the varied demands for NH₃ production in the future energy landscape, encompassing the electrochemical N₂ reduction reaction (NRR)¹³⁻¹⁵, lithium-mediated NRR¹⁶⁻¹⁹ and nitrate reduction reaction (NO₃RR)²⁰⁻²². Among them, the present NH₃ yield via NO₃RR surpasses those of NRR by two to three orders of magnitude, mainly due to the relatively lower dissociation energy of the N=O bond (204 kJ mol⁻¹) of nitrate anion (NO₃⁻) compared to the N≡N bond (941 kJ mol⁻¹)^{23,24}. Besides, NO₃⁻ exhibits a ubiquitous presence within contaminated groundwater and industrial effluent^{25,26}, with their availability further augmented through the oxidation process of atmospheric N₂ as the source of nitrogen^{9,27,28}. Consequently, the NO₃RR pathway has emerged as one of the most potential in renewable NH₃ production.” (Pages 3-4, Revised manuscript).

***Comment 2:** Directly linked to the comment above is the questionable purpose of this study. What is the practical reason for the transformation of dilute nitrate solutions into dilute solutions of similarly toxic ammonium? The authors put a lot of emphasis on remediation and achieving WHO drinkable limits. However, if this would be the aim of the work, the target product should be genuinely benign N₂, not ammonium.*

Response: We thank the reviewer for raising the concern about the purpose of this study. We have revised the descriptions about WHO drinkable limits, and explained the significance for NO₃RR under low concentration and the recovery of ammonia as a valuable product.

a) A considerable portion of the groundwater, stemming from both agricultural and industrial sources (*Joule* 2021, 5, 290-294; *Nat. Commun.* 2023, 14, 7368; *Water Res.*

2017, 120, 1-11), has a diluted nitrate concentration, necessitating a platform suitable for operation at low concentration. Also, the removal process of NO_3^- continuously reduces its concentration, inevitably reaching the low concentration regime with reduced performance. Hence, we focus on NO_3RR under low concentration.

b) We agree with the reviewer's opinion that both nitrate and ammonium are toxic in water bodies, in which the product should be N_2 for water remediation. While, the aim of this work is to obtain the efficient ammonia production via NO_3RR , rather than the denitrification of wastewater. Hence, we have deleted and revised the description about the remediation of contaminated water in the manuscript.

c) In addition to generating ammonia via the NO_3RR process, we further conducted supplementary experiments to separate and collect ammonia from the aqueous system (Figures 4g-h). Consequently, the nitrate is converted, and the generated ammonium is recovered as a potential valuable product from the aqueous system.

Figure 4. g) Schematic of the ammonia product synthesis process from 100 mM NO_3^- electrolyte to NH_4Cl at -1.0 V *versus* RHE. h) The conversion efficiency of different steps for the ammonia product synthesis process. Numbers on the *x*-axis indicated the corresponding conversion steps in panel (g). Error bars indicate the relative standard deviations of the mean ($n = 3$).

Corresponding revision:

➤ The corresponding content in the abstract has been revised to “This high selectivity enables over 99.3% NO_3^- -to- NH_3 conversion efficiency, reducing NO_3^- concentrations from 100 mM to low levels of 0.54 mM.” (Page 2, Revised manuscript).

➤ The corresponding content in the introduction has been revised to “Ammonia (NH_3), one of the most common industrial chemicals, is crucial for the production of agricultural fertilizers and holds immense potential as a green hydrogen-rich fuel¹⁻³. The current global ammonia demand exceeds 180 million tons annually, primarily

fulfilled through industrial scale production of energy intensive Haber-Bosch (H-B) process^{1,4}. Within this process, steam-reformed hydrogen (H₂) undergoes reaction with nitrogen (N₂) under elevated temperature (~500 °C) and pressure (>100 atm)^{5,6}, which not only accounts for approximately 1.4% of global carbon dioxide (CO₂) emissions, but also necessitates the consumption of 2% of the world's total energy supply^{7,8}. Recently, electrocatalytic methodologies have surfaced as a viable clean energy pathway for decentralized ammonia synthesis at room temperature, accommodating a range of infrastructure scales and potentially powered by locally sourced renewable energy sources^{9,10}. Despite the substantial global demand that may sustain the traditional ammonia production route in the near future, the electrochemical ammonia synthesis can be promising complementary process to the Haber–Bosch technology for contributing to decarbonizing ammonia production^{11,12}.

Recently, diverse electrochemical approaches have been explored to address the varied demands for NH₃ production in the future energy landscape, encompassing the electrochemical N₂ reduction reaction (NRR)¹³⁻¹⁵, lithium-mediated NRR¹⁶⁻¹⁹ and nitrate reduction reaction (NO₃RR)²⁰⁻²². Among them, the present NH₃ yield via NO₃RR surpasses those of NRR by two to three orders of magnitude, mainly due to the relatively lower dissociation energy of the N=O bond (204 kJ mol⁻¹) of nitrate anion (NO₃⁻) compared to the N≡N bond (941 kJ mol⁻¹)^{23,24}. Besides, NO₃⁻ exhibits a ubiquitous presence within contaminated groundwater and industrial effluent^{25,26}, with their availability further augmented through the oxidation process of atmospheric N₂ as the source of nitrogen^{9,27,28}. Consequently, the NO₃RR pathway has emerged as one of the most potential in renewable NH₃ production.” (Pages 3-4, Revised manuscript).

➤ The corresponding content in the introduction has been revised to “On the other hand, a considerable portion of the groundwater, stemming from both agricultural and industrial sources³⁵⁻⁴⁰, has a diluted concentration, necessitating a platform suitable for operation at low concentration. More importantly, the removal process of NO₃⁻ continuously reduces its concentration, inevitably reaching the low concentration regime with reduced performance.” (Page 4, Revised manuscript).

➤ The corresponding content in the introduction has been revised to “Furthermore, the unprecedented NO_3^- -to- NH_3 conversion efficiency (99.3%) enables the removal of nitrate from 100 mM to low levels of 0.54 mM within 3 hours, in sharp contrast with the reference (still relatively high level under 48 hours operation).” (Page 5, Revised manuscript).

➤ The corresponding content has been revised to “Impressively, the NO_3RR on the three samples followed typical quasi-first-order kinetics, and the NO_3^- to NH_3 selectivity of Ag- MoS_2 was up to 99.3% within only 3 hours of electrolysis at -0.6 V versus RHE (NO_3^- concentration decreased sharply from 100 mM to 0.54 mM, Figures 4e and S46), manifesting that nearly all the N sources were converted into NH_3 (Figure S47). Comparatively, the NO_3^- to NH_3 conversion efficiency of W- MoS_2 and MoS_2 were as low as ~16% and ~30%, and the residual NO_3^- still remained a relatively high level of 4.64 mM and 11.78 mM even after 48 hours (Figure S48).” (Pages 15-16, Revised manuscript).

➤ The corresponding content in the conclusion has been revised to “By virtue of the unparalleled performance at ultra-low concentrations ($\sim 20 \text{ mg h}^{-1} \text{ cm}^{-2}$, FE of nearly 100% under concentration of 10 mM), the catalyst with S-L junction succeeded in removing NO_3^- to low level within 3 hours, while the counterpart without the junction cannot meet the level after treatment of 48 hours.” (Page 20, Revised manuscript).

Comment 3: Further, the choice of conditions used to study the nitrate-to-ammonia are very hard to understand. Why using non-buffering K_2SO_4 electrolyte, which provides essentially no control over the reaction conditions and is very challenging to translate into a practical technology? Practical nitrate to ammonia reduction requires the use of alkaline conditions, which can be integrated into a PEM-based device as recently demonstrated [10.1038/s41929-024-01200-w].

However, the use of alkaline conditions for a Mo-based electrode might not be a fruitful idea due to their limited stability under these conditions. This further questions the impact of the presented results.

Response: We thank the reviewer for raising the concern about the choice of electrolyte and clarified it as the following four parts:

a) The significance of alkaline NO₃RR. The alkaline NO₃RR process is favorable for inhibiting the competitive HER. Besides, the elevated pH in the cathode chamber after catalysis creates a higher pH environment, which helps to reduce the energy consumption associated with air stripping of NH₃(g) products recovery. Thus, we agree with reviewer's opinion that practical nitrate to ammonia reduction requires the use of alkaline conditions. Also, we have cited the references (10.1038/s41929-024-01200-w, #10) in the revised manuscript.

b) The significance of neutral NO₃RR. According to the reported literatures, compared with nuclear waste with the highest level of nitrate, a more widely available source of NO₃⁻ is from groundwater, for example from agricultural runoff (*Joule* 2021, 5, 290-294; *Nature* 1985, 313, 214–216; *Water Res.* 2005, 39, 3-16). Usually, the mean pH of groundwater is within 6-8 (*Water Res.* 2010, 44, 1062-1071; *Sci. Rep.* 2021, 11, 2598), which approaches neutrality. Base on that, we chose a neutral electrolyte as the research conditions.

c) The reason for choosing K₂SO₄+KNO₃ electrolyte. Cation effect is crucial for the NO₃RR process, in which K⁺ exhibits a higher NO₃RR performance than those of Na⁺, Cs⁺, and Li⁺ (*Angew. Chem. Int. Ed.* 2024, 63, e202408382). Therefore, the commonly utilized K⁺ is chosen as the cation in electrolyte. Meanwhile, SO₄²⁻ exhibits high conductivity, and K₂SO₄+KNO₃ system has been widely used for NO₃RR process under neutral condition (*Adv. Sci.* 2024, 2410763; *Angew. Chem. Int. Ed.* 2024, e202416910; *Angew. Chem. Int. Ed.* 2022, 61, e202204117.).

d) NO₃RR performance comparison in different electrolyte. To probe the NO₃RR performance in buffer solution, we have prepared a 0.5 M PBS solution for all the test (Figure S44a). The NH₃ yield rate of Ag-MoS₂ in PBS solution is 13.4 mg h⁻¹ cm⁻², with the FE of 83%, which is lower than that in K₂SO₄ electrolyte (Figure 4a). The NO₃RR performance of MoS₂ and W-MoS₂ in PBS electrolyte are also inferior to that

in K_2SO_4 electrolyte. This may be ascribed to the secondary ionization and the relatively low conductivity of PBS (*CRC Handbook of Chemistry and Physics*, 2016).

Figure 4a. NH₃ FE and NH₃ yield rate of the three samples in 0.5 M K₂SO₄ system.

Figure S44. (a) NH₃ FE and NH₃ yield rate of the three samples in 0.5 M PBS system. (b) The pH of K₂SO₄ and PBS electrolyte before and after the NO₃RR process.

Besides, we have measured the pH of both PBS and K₂SO₄ electrolyte before and after NO₃RR. As seen from Figure S44b, the pH of K₂SO₄ electrolyte varies from 6.71 to 11.87, while the pH of PBS electrolyte changes slightly from 7.02 to 7.08. Within the NO₃RR process, the hydrogenation of N-contained intermediates leads to the continuous consumption of H⁺. Thus, the neutral K₂SO₄ system changes to alkaline after performing NO₃RR tests. This phenomenon had been reported by the related literatures (*Nat. Catal.* 2024, 7, 1032–1043; *Adv. Funct. Mater.* 2023, 33, 2300512). The continuous increased pH can not only benefit the inhibition of competitive HER, but also favor the recycle of ammonia products.

While, the KH_2PO_4 and K_2HPO_4 are the main components in PBS buffer solution. The secondary dissociation characteristic can counteract the influence of H^+ consumption on the acidity or alkalinity of the solution, thereby maintaining a relatively stable pH value of the solution. Under this condition, the negligible pH change (7.02 to 7.08) may not that effective to inhibit the undesirable HER, leading to the reduction of FE. These results suggest that K_2SO_4 could be available as the supporting electrolyte.

Corresponding revision:

- Figure S44 has been added (Page 53, Revised supplementary information).
- The corresponding content has been revised to “To probe the NO_3RR performance in buffer solution, we have prepared a 0.5 M PBS solution for all the test (Figure S44a). The NH_3 yield rate of Ag- MoS_2 in PBS solution is $13.4 \text{ mg h}^{-1} \text{ cm}^{-2}$, with the FE of 83%, which is lower than that in K_2SO_4 electrolyte (Figure 4a). The NO_3RR performance of MoS_2 and W- MoS_2 in PBS electrolyte are also inferior to that in K_2SO_4 electrolyte. This may be ascribed to the secondary ionization and the relatively low conductivity of PBS.

Besides, we have measured the pH of both PBS and K_2SO_4 electrolyte before and after NO_3RR . As seen from Figure S44b, the pH of K_2SO_4 electrolyte varied from 6.71 to 11.87, while the pH of PBS electrolyte changed from 7.02 to 7.08. Within the NO_3RR process, the hydrogenation of N-contained intermediates led to the continuous consumption of H^+ . Thus, the neutral K_2SO_4 system changed to alkaline at the end of NO_3RR . The continuous increased pH can not only benefit the inhibition of competitive HER, but also favor the recycle of ammonia products (alkaline environment was conducive to the escape of NH_3).

While, the KH_2PO_4 and K_2HPO_4 are the main components in PBS buffer solution. The secondary dissociation characteristic can counteract the influence of H^+ consumption on the acidity or alkalinity of the solution, thereby maintaining a relatively stable pH value of the solution. Under this condition, the unchanged pH (7.02 to 7.08) may not that effective to inhibit the undesirable HER, leading to the reduction of FE.

These results suggested that K_2SO_4 could be available as the supporting electrolyte.”

(Page 53, Revised supplementary information).

➤ The corresponding content has been revised to “In addition, from the NO_3RR performance at a wide range of NO_3^- concentrations (Figure S42), the optimal Ag-MoS₂ in K_2SO_4 system (Figures S43-44) exhibited its broad adaptability and achieved an excellent NH₃ FE (> 90%, in Figure S45), even at ultra-low concentration (1 mM).”

(Page 15, Revised manuscript).

Comment 4: It is important to recognise that MoS₂ produces only ~33% less ammonia than Ag-MoS₂, i.e. differences in the rate of the target process are not substantial. Hence, even if the proposed effects are real (see next comment), they are not sufficiently strong to form a basis for a strategy to design effective electrocatalysts for the nitrate-to-ammonia reduction reaction.

Response: Thanks for raising the concern about the NH₃ yield rate. In the electrocatalytic nitrate reaction system, two reference samples are compared with Ag-MoS₂ catalyst that exhibits S-L junction with strong built-in electric fields (Figures S10 and S14). These references include W-MoS₂, which does not possess a significant S-L junction (Figure S20), and MoS₂, which has a weaker built-in electric field at S-L junction (Figure S11).

By evaluating the NO_3RR performance (Figure 4a), Ag-MoS₂ displays the superior performance, with a near unit NH₃ Faradaic efficiency (FE) of 99.7% at -0.6 V *versus* RHE and an optimal NH₃ yield rate of 20.1 mg h⁻¹ cm⁻² at -1.0 V *versus* RHE. The optimal NH₃ yield rate of Ag-MoS₂ at -1.0 V *versus* RHE is about 3.4-folds and 1.7-folds those of W-MoS₂ and MoS₂, respectively. Similarly, at -0.6 V *versus* RHE, the peak NH₃ FE of Ag-MoS₂ is 3.2-folds and 1.8-folds those of W-MoS₂ and MoS₂, respectively, with corresponding ammonia yield rate being 3.6-folds and 2.1-folds than those of W-MoS₂ and MoS₂. These results display the superior NO_3RR performance of Ag-MoS₂ compared to its counterparts. Moreover, the performance of Ag-MoS₂ emerges remarkable advantage over the most state-of-the-art NH₃ activity ever reported

at a low NO_3^- system (Figure 4c and Table S5). Consequently, Ag-MoS₂ with reinforced S-L junction could promote NO_3^- adsorption to benefit NO_3RR process.

Figure S10. (a) Interfacial thermal equilibrium process when the Ag-MoS₂ contact with the NO_3^- contained solution. Downward band bending at solid-liquid (S-L) junction under (b) no potential and (c) reduction potential. E_{redox} and E_F represent the theoretical redox potential of $\text{NO}_3^-/\text{NH}_3$ and the Fermi level potential. CBM and VBM express the conduction band minimum and valence band maximum of semiconductor, respectively.

Figure S14. Calculated electric field intensity within the S-L junction of the as-prepared samples.

Figure S20. Interfacial thermal equilibrium process when W-MoS₂ contacting with NO_3^- electrolyte.

Figure S11. (a) Interfacial thermal equilibrium process when MoS₂ contact with the NO₃⁻ contained solution. Downward band bending at solid-liquid (S-L) junction under (b) no potential and (c) reduction potential.

Figure 4. (a) NH₃ yield rate and NH₃ FE of catalysts in 10 mM NO₃⁻ electrolyte. (c) NO₃RR performance comparison of reported electrocatalysts (NO₃⁻ concentration ≤ 10 mM).

Table S5. Comparison of NO₃RR performance of Ag-MoS₂ with reported works at low nitrate concentration system (≤20 mM).

Catalyst	Electrolyte	NH ₃ FE (%)	NH ₃ production rate (mg h ⁻¹ cm ⁻²)	Current density (mA cm ⁻²)	Ref.
Ag-MoS ₂	10 mM KNO ₃ + 0.5 M K ₂ SO ₄	~100	~20	~200	This work
FeNi ₅₀₀ /FF	14.3 mM KNO ₃ + 0.05 M Na ₂ SO ₄	65.2	~0.27	~10	Nat. Water 2023, 1, 1068-1078
Co-Fe@Fe ₂ O ₃	500 ppm NO ₃ ⁻ -N + 0.1 M Na ₂ SO ₄	85.2	0.885	~17	PNAS 2022, 119, e2115504119
Cu-PTCDA	8.1 mM KNO ₃ + 1 M Na ₂ SO ₄	85.9	0.44	15	Nat. Energy 2020, 5,

	PBS				605-613
Pd	20 mM NaNO ₃ + 0.1 M NaOH	35	0.34	4.25	ACS Catal. 2021, 11, 12, 7568-7577
Cu@C	1 mM KNO ₃ + 1 M KOH	72	0.47	5.5	Adv. Mater. 2022, 34, 2204306
CuCl ₂ _BEF	7.1 mM KNO ₃ + 0.5 M Na ₂ SO ₄	98.6	1.82	62	Angew. Chem. Int. Ed. 2021, 60, 22933-22939
Cu/Cu ₂ O nanowires	3.2 mM NaNO ₃ + 0.5 M Na ₂ SO ₄	95.8	4.1	120	Angew. Chem. Int. Ed. 2020, 59, 5350-5354
a-RuO ₂	3.2 mM NaNO ₃ + 0.5 M Na ₂ SO ₄	97.46	2.0	60	Angew. Chem. Int. Ed. 2022, 134, e202202604
O-SiNW/Au	10 mM HNO ₃ + 0.5 M K ₂ SO ₄	95.6	4.4	—	Angew. Chem. Int. Ed. 2022, 61, e202204117

Corresponding revision:

➤ The corresponding content has been revised to “Of these, Ag-MoS₂ displayed the superior performance (Figures 4a-b), with a near unit NH₃ Faradaic efficiency (FE) of 99.7% at -0.6 V *versus* RHE (~200 mA cm⁻²) and an optimal NH₃ yield rate of 20.1 mg h⁻¹ cm⁻² at -1.0 V *versus* RHE (~340 mA cm⁻²). The optimal NH₃ yield rate value was about 3.4-folds and 1.7-folds than those of W-MoS₂ and MoS₂, respectively. Similarly, the peak NH₃ FE of Ag-MoS₂ was approximately 3.2-folds and 1.8-folds than those of W-MoS₂ and MoS₂, respectively. These results displayed the superior NO₃RR performance of Ag-MoS₂ compared to its counterparts.” (Page 14, Revised manuscript).

Comment 5: *The evidence for the increase in the local, near-electrode concentration of nitrate due to the Ag doping of MoS₂ is unconvincing:*

5.1. Comprehensive control data for high-surface area Ag nanoparticles deposited on a different support, e.g. carbon black, are needed. Probably, simple silver nps would produce even better results?

Response: Thanks so much for your constructive comment. To address your concerns, we have prepared comparative samples including Ag nanoparticles on carbon black (Ag NP/C, Figure S38) and Ag nanoparticles on MoS₂ (Ag NP/MoS₂, Figure S39), and confirmed their successful preparation by series characterizations.

Figure S38. (a, b) SEM images, (c) XRD pattern, and (d) XPS spectra of Ag 3d orbit in Ag NP/C catalyst.

Figure S39. XPS spectra of (a) Ag 3d, (b) Mo 3d, and (c) S 2p orbits in Ag NP/MoS₂ catalyst. (d, e) SEM images of Ag NP/MoS₂ sample.

The NH_3 yield rate and FE of the two samples are tested (Figure S41). NH_3 yield rate and FE of Ag NP/C is $2.88 \text{ mg h}^{-1} \text{ cm}^{-2}$ and 12.5% respectively, much lower than that of Ag-MoS₂ catalyst (NH_3 yield rate: $20.06 \text{ mg h}^{-1} \text{ cm}^{-2}$, FE: 99.6%). The Ag NP/MoS₂ samples also exhibit inferior NO₃RR performance (NH_3 yield rate: $11.7 \text{ mg h}^{-1} \text{ cm}^{-2}$, FE: 59.7%) than that of Ag-MoS₂ and comparable to MoS₂ (Figure 4a). Thus, the possible contribution of Ag nanoparticles on the NO₃RR performance can be eliminated.

Figure S41. NH_3 yield rate and FE of Ag NP/C, Ag NP/MoS₂ and Ag-MoS₂ samples.

Figure 4a. NH_3 yield rate and NH_3 FE of catalysts in 10 mM NO_3^- electrolyte.

Corresponding revision:

➤ Figures S38-39 and S41 have been added (Pages 47-48 and 50, Revised supplementary information).

➤ The descriptions have been added “Besides, the possible contribution of Ag nanoparticles on the NO_3^- enrichment and NO_3RR performance of MoS_2 has been eliminated (Figures S38-41).” (Page 14, Revised manuscript).

➤ The descriptions have been added “As seen from Figures S38a-b, the Ag nanoparticles distribute on the carbon black evenly. In the XRD pattern, besides the characteristic peaks of carbon substrate, obvious peaks at 38.1° and 77.4° can be indexed to the (111) and (311) planes of cubic Ag. In high-resolution XPS spectra, the binding energies of 373.69 and 367.67 eV are assigned $3d_{3/2}$ and $3d_{5/2}$ orbits of metallic Ag. These results verified the successful preparation of Ag nanoparticles on carbon black (Ag NP/C).” (Page 47, Revised Supplementary information).

➤ The descriptions have been added “According to the high-resolution XPS spectra, the presence of MoS_2 and metallic Ag can be confirmed. MoS_2 exhibits the nanoflower morphology. After Ag modification, Ag nanoparticles (NP) distribute on the surface of MoS_2 , further demonstrating the successful preparation of Ag NP/ MoS_2 control samples.” (Page 48, Revised Supplementary information).

➤ The descriptions have been added “The NH_3 yield rate and FE of the Ag NP/C, Ag NP/ MoS_2 and Ag- MoS_2 samples are summarized and displayed in Figure S41. For Ag NP/C, the FE is as low as 12.5%, with the NH_3 yield of $2.88 \text{ mg h}^{-1} \text{ cm}^{-2}$, indicating the poor NO_3RR performance of Ag NP/C.

When anchoring the Ag NP on MoS_2 nanoflower (Ag NP/ MoS_2), the NH_3 yield rate and FE are $11.7 \text{ mg h}^{-1} \text{ cm}^{-2}$ and 59%, respectively. While, the NO_3RR performance is still much lower than that of Ag- MoS_2 and comparable to MoS_2 . Thus, the promoted NO_3RR activity of Ag- MoS_2 can be ascribed to the Ag dopant induced surface NO_3^- enrichment effect.” (Page 50, Revised supplementary information).

5.2. Increased Raman signals on Ag/ MoS_2 can be just a plasmonic, SERS, effect.

Response: Thanks so much for your comment. According to the characterizations of Ag- MoS_2 catalyst, there exist no metallic Ag in the catalyst, in which the Ag acts as the dopant in the materials (Figures 2e-f, S5c, and S6).

Figure 2. e) XRD patterns of Ag-MoS₂ and MoS₂. f) Fourier transformed k^2 -weighted EXAFS spectra of Ag-MoS₂ and reference samples.

Figure S5. (c) Ag 3d XPS spectra of catalysts.

Figure S6. (a) Mo K-edge XANES spectra and (b) Ag K-edge XANES spectra of Ag-MoS₂ and reference samples.

To further eliminate the potential influence of SERS effect on the increased Raman signals on Ag-MoS₂, we treat the Ag-MoS₂, Ag NP/MoS₂, and Ag NP/C catalysts with

the RhB solution (concentration of 5 μM), respectively. The Raman test is carried out with the excitation wavelength of 532 nm. As seen from Figure S40, for Ag NP/C and Ag NP/MoS₂ samples, characteristic Raman peaks of RhB emerge. Specifically, the bands center at 1198 and 1525 cm^{-1} are ascribed to aromatic C-H bending in RhB molecule, the peak at 1280 cm^{-1} is the result of C-C bridge-bands stretching, the bands of 1360, 1507, 1560, and 1645 cm^{-1} originate from the aromatic C-C stretching modes, while the peak at 1597 cm^{-1} is attributed to C-H stretching, those are all characteristic vibration modes of RhB dye. The intensity of these RhB characteristic peaks on Ag NP/MoS₂ and Ag NP/C are as high as 10^5 level.

While, no obvious Raman peaks are observed of Ag-MoS₂ sample, which further verify that no metallic Ag exists in Ag-MoS₂, and the Ag-MoS₂ sample has no SERS effect. The increased Raman signal of NO₃⁻ on Ag-MoS₂ can be ascribed to the enrichment effect (Figures 3d-e), which benefits the NO₃RR performance.

Figure S40. Raman spectra of Ag-MoS₂, Ag NP/MoS₂, and Ag NP/C samples under the treatment of RhB on the surface.

Figure 3. (d) *In situ* Raman spectra of Ag-MoS₂. (e) Corresponding peak intensity of absorbed NO₃⁻ in Raman spectra of catalysts at various potentials.

Corresponding revision:

➤ Figures S40 has been added (Page 49, Revised supplementary information).

➤ The descriptions of “To further eliminate the potential influence of SERS effect on the increased Raman signals on Ag-MoS₂, we treated the Ag-MoS₂, Ag NP/MoS₂, and Ag NP/C catalysts with the RhB solution (concentration of 5 μM), respectively. The Raman test was carried out with the excitation wavelength of 532 nm. As seen from Figure S40, for Ag NP/C and Ag NP/MoS₂ samples, characteristic Raman peaks of RhB emerged. Specifically, the bands center at 1198 and 1525 cm⁻¹ are ascribed to aromatic C-H bending in RhB molecule, the peak at 1280 cm⁻¹ is the result of C-C bridge-bands stretching, the bands of 1360, 1507, 1560, and 1645 cm⁻¹ originate from the aromatic C-C stretching modes, while the peak at 1597 cm⁻¹ is attributed to C-H stretching, those are all characteristic vibration modes of RhB dye. The intensity of these RhB characteristic peaks on Ag NP/MoS₂ and Ag NP/C are as high as 10⁵ level.

While, no obvious Raman peaks are observed of Ag-MoS₂ sample, which further verify that no metallic Ag exists in Ag-MoS₂, and the Ag-MoS₂ sample has no SERS effect. The increased Raman signal of NO₃⁻ on Ag-MoS₂ can be ascribed to the enrichment effect, which benefits the NO₃RR performance.” have been added (Page 49, Revised supplementary information).

5.3. Increased NO₃⁻ adsorption can be just due to the preferential adsorption of nitrate on the surface of silver nanoparticles, which has nothing to do with localised concentration increase.

Response: Thanks so much for your comment. As evidenced in the above analysis, no metallic Ag exists in the optimized Ag-MoS₂ catalyst. Also, the Ag-MoS₂ catalyst exhibit bare SERS effect. Besides, we have carried out the NO₃⁻ adsorption experiment on Ag NP/C, Ag NP/MoS₂ and Ag-MoS₂ samples different applied potentials (Figure S41a). Clearly, the adsorbed NO₃⁻ concentrations on Ag NP/C and Ag NP/MoS₂ at each potential are inferior to that on Ag-MoS₂ catalyst.

Therefore, we consider that the NO_3^- enrichment on the Ag-MoS₂ surface is due to the formation of robust solid-liquid junction, rather than the contribution of Ag nanoparticles.

Figure S41. (a) The concentration of NO_3^- adsorbed on Ag NP/C, Ag NP/MoS₂, and Ag-MoS₂ at different applied potentials in NO_3^- containing electrolyte, tested by ion chromatography.

Corresponding revision:

- **Figures S41** has been added (Page 50, Revised supplementary information).
- The descriptions of “We have carried out the NO_3^- adsorption experiment on Ag NP/C, Ag NP/MoS₂ and Ag-MoS₂ samples different applied potentials (Figure S41a). Clearly, the adsorbed NO_3^- concentrations on Ag NP/C and Ag NP/MoS₂ at each potential are inferior to that on Ag-MoS₂ catalyst.” have been added (Page 50, Revised supplementary information).

5.4. If the differences in the local NO_3^- concentrations were as substantial as presented, why the rates of the nitrate reduction are essentially the same for MoS₂ and Ag-MoS₂, the difference being only some preference of the latter for the formation of ammonium?

Response: Thanks so much for your comment. The questions raised by the reviewer are answered as follows.

Firstly, in **Figure S42** (original **Figure 4d**) and **Figure 4e**, the reduction rate of Ag-MoS₂ under low concentration condition is fast than those of MoS₂ and W-MoS₂. Under the NO_3^- concentration of 1, 5, and 10 mM at -0.6 V vs. RHE, the NH_3 yield rate of Ag-MoS₂ is 2.5, 2.1, and 2.1 times than that of MoS₂, and is 2.9, 3.1, and 3.3 times than

that of W-MoS₂. Under low-concentration condition, the NO₃⁻ mass transfer/diffusion can be the rate determining step that limits the NO₃RR performance, and the superior enrichment effect of Ag-MoS₂ benefits the NH₃ yield rate.

When under high-concentration condition (especially at 1000 mM), abundant NO₃⁻ distribute within the electrolyte, no matter in bulk solution or at the electrode surface. Thus, even with negative polarization, the surface NO₃⁻ concentration is sufficient enough for NO₃⁻ reduction without causing concentration polarization. This further highlights the significance for investigating the NO₃RR under low concentration condition.

Figure S42. NH₃ yield rate of catalysts in different NO₃⁻ concentrations at -0.6 V *versus* RHE. Insert is the NH₃ yield rate of catalysts in low NO₃⁻ concentration (≤ 10 mM) from the dark blue region.

Figure 4. (e) NO₃⁻ removal in initial 0.5 M K₂SO₄ with 100 mM NO₃⁻ electrolyte at -0.6 V *versus* RHE in H-cell reactor. After 3 hours of electrolysis, only 0.54 mM of NO₃⁻-N. Insert is the enlarged vision within 3 hours.

Overall, none of the methods employed probes the phenomenon in question. The reviewer recognises that this is exceptionally hard, but this does decrease the current very high level of speculations in the presented interpretations.

Response: To address the reviewer's concerns, we have prepared series control samples (Ag NP/C and Ag NP/MoS₂) and supplemented corresponding characterizations (including SERS, NO₃⁻ adsorption, and NO₃RR performance, etc.). We hope that our efforts are sufficient to answer the questions of the reviewer.

Comment 6: Standard potential for the nitrate-to-ammonia redox couple is approximately 0.7 V vs. SHE, not 0.3 V vs. RHE at pH 7.

Response: Thanks so much for your comment. In response to your concerns, we have carefully searched the related literatures, and revised the corresponding descriptions.

Specifically, literatures have reported that the most widely accepted reaction of NO₃RR is $NO_3^- + 9H^+ + 8e^- \rightarrow NH_3 + 3H_2O$ (*Nat. Catal.* 6, 2023, 402-414; *Chem. Soc. Rev.* 2021, 50, 6720; *Electrochem. Commun.* 2021, 129, 107094; *Chem. Eng. J.* 2024, 495, 153108), in which nine H⁺ and eight electrons are participated. This equation not only underscores the reaction stoichiometry but also the electrochemical potential required for the reduction of NO₃⁻ to NH₃ under standard conditions (*Chem. Eng. J.* 2024, 495, 153108). The standard potential E^θ of this equation is: $E^\theta = -0.12 \text{ V vs. SHE}$. In the present work, the pH of the mixed K₂SO₄ and KNO₃ electrolyte is 6.71. After considering pH, the standard potential can be converted to 0.27 V vs. RHE. Thus, we have revised the descriptions and added corresponding discussions.

Corresponding revision

➤ The corresponding content has been revised and updated “the theoretical redox potential of NO₃⁻/NH₃ (E_{redox} , 0.27 V versus RHE under neutral condition (pH 6.71))⁵⁰⁻⁵³” (Page 9, Revised manuscript).

Comment 7: *How can the authors explain the difference in the double-layer capacitance measured by ACV and dc voltammetry, which show opposite trends?*

Response: Thanks so much for your constructive comment, and the questions raised by the reviewer are answered by the following two parts:

a) The capacitance behavior difference between DC and AC input signals. First, in the ACV and EIS results, we find the capacitance variation trend of the three samples are consistent, that is, $\text{Ag-MoS}_2 < \text{MoS}_2 < \text{W-MoS}_2$ (Figures 3b-c). As the input signal of both ACV and EIS techniques are alternating current (AC), this consistent capacitance variation trend is reasonable. For the capacitance under AC condition, due to the continuous changes in voltage, capacitors will constantly charge and discharge, and there will always be current flowing in the circuit.

While, the cyclic voltammetry test is performed using direct current (DC). For the capacitance under DC condition, once the charging process is completed, the capacitive region is fully charged and behaves like an open circuit.

Figure 3. b) The non-Faradaic capacitance-potential curves for the diverse catalysts. The potential of zero charge (PZC) describes the condition when the capacitance on a surface is minimal. c) The fitted surface capacitance of the three catalysts.

b) Re-perform the cyclic voltammetry under revised potential range. We acknowledge that the potential range for DC voltammetry test is not appropriate. Within the potential range of 0.92-1.02 V vs. RHE, this much positive potential may cause the partial oxidation of the three samples, leading to the influence of Faradaic current on the measured capacitance. Thus, we have re-performed the CV test within the potential range of 0.62-0.72 V vs. RHE, which is consistent with the potential range of ACV test

(Figure S35). Specifically, the calculated C_{dl} of Ag-MoS₂, MoS₂, and W-MoS₂ are 1.8, 2.7, and 3.2 mF. The numerical sorting of C_{dl} is as follows: Ag-MoS₂ < MoS₂ < W-MoS₂. Under this condition, the variation trend is consistent with that of ACV. Correspondingly, we have revised the corresponding experiments and discussions.

Figure S35. Cyclic voltammetry curves of (a) Ag-MoS₂, (b) MoS₂, (c) W-MoS₂ catalysts at different scan rates, and their (d, e, f) current density differences at 0.66 V versus RHE against scan rates to calculate C_{dl} .

Corresponding revision:

- Figure S35 has been revised and updated (Page 44, Revised supplementary information).
- The descriptions of have been revised “The electrochemical double-layer capacitance (C_{dl}) of the materials was tested to determine their electrochemical surface area (ECSA) using the cyclic voltammetry (CV) in non-faradic regions with diverse scan rates ranging from 20 to 100 mV s⁻¹ between 0.62 and 0.72 V *versus* RHE. The plotted current density (difference between the anode current density and cathode current density at 0.66 V *versus* RHE) against scan rate has shown a linear relationship and its slope was twice the C_{dl} .” (Page 2, Revised supplementary information).

Comment 8: *Qualitative interpretations of the non-linear capacitance measured by ACV can hardly produce meaningful results, especially in terms of the PZC. Interpretation of the potential corresponding to the capacitance minimum in terms of*

PZC is highly questionable. Moreover, the presented data are within the experimental error given the poorly defined shape of the fundamental harmonic.

Response: We thank the reviewer for the comments, and the questions are answered from the following three parts:

a) Alternating current voltammetry (ACV) is an electrochemical method which utilize a direct current potential that scans slowly with time to the working electrode is applied, and superimposing a small signal of an AC sine wave on the DC signal as an excitation or perturbation signal. This method does not interfere with the charging current, so it has high accuracy and can reflect the substance change with the concentration as low as 10^{-7} mol/L. In the field of Lithium-ion battery, ACV is usually utilized to record the adsorption behavior change in IHP range (*Nano Lett.* 2023, 23, 7014–7022; *J. Am. Chem. Soc.* 2019, 141, 9422–9429; *Angew. Chem.* 2023, 135, e202302302). Thus, we performed the ACV test to monitor the adsorption of NO_3^- within the IHP region.

b) Within the test potential range, the potential of zero charge (PZC) refers to the potential at which no excess charge exists on surface and describes the condition when the capacitance on a surface is minimum, which is adopted as an indicator for the IHP structure evolution (*ACS Nano* 2023, 17, 24, 24619–24631). Specifically, the adsorption of anion would lead to the negative shift of PZC, whereas the cation resulted in the opposite shift (*Analyst* 2009, 134, 1608–1613).

c) In Figure 3b, we acknowledge that the signal-to-noise ratio of the three curves is relatively poor, which reduces the accuracy of PZC values. We have re-performed the ACV test under the same condition, and the updated Figure 3b show significant peaks of the three samples. The PZC values of W-MoS₂, MoS₂, and Ag-MoS₂ are 0.68, 0.64, and 0.57 V vs. RHE, respectively. The negatively shifted PZC of Ag-MoS₂ suggests that more reactant NO_3^- can be enriched within the IHP region, which benefits the NO₃RR performance.

Figure 3b. The capacitance-potential curves for the diverse catalysts. The potential of zero charge (PZC) describes the condition when the capacitance on a surface is minimal.

Corresponding revision:

- Figure 3b has been revised and updated (Page 13, Revised manuscript).
- The descriptions of “ACV was a non-destructive technique which had been widely utilized to record the adsorption behavior change in IHP range^{56,57}. Compared to MoS₂, Ag-MoS₂ exhibited negatively shifted potential of zero charges (PZC, 0.57 V versus RHE). PZC was adopted as indicator for the IHP structure evolution, and the negative PZC indicated the adsorption of anion on the IHP region of Ag-MoS₂ (Figures 3h and S10-11)^{58,59}.” have been added (Page 10, Revised manuscript).

***Comment 9:** Qualitative interpretations of impedance data in terms of the phase angle dependence only is very hard to understand and accept. EIS should be modelled with a physicochemically sensible model and only these data can provide some clues on the differences in the performance.*

Response: Thanks so much for your comment. The questions raised by the reviewer are answered by the following two parts:

a) Firstly, Figure 3f in the manuscript is the fitted Bode plots using the equivalent physical circuit diagram. The equivalent physical circuit diagram is displayed in Figure S15, in which the circuit of W-MoS₂ is different from those of Ag-MoS₂ and MoS₂. To illustrate the rationality and fitting accuracy of the model, detailed parameters for EIS fitting is supplemented (Table S6). Clearly, the relative standard deviations of all the

parameters are below 5%, suggesting the reliability of the selected equivalent physical circuit model for EIS results fitting.

Figure 3. (f) Bode plots of catalysts at the reduction potential of -0.6 V *versus* RHE in 10 mM NO_3^- electrolyte.

Figure S15. The fitted equivalent circuit diagram of (a) W-MoS₂ and (b) Ag-MoS₂/MoS₂ catalysts at -0.6 V *versus* RHE.

Table S6. Fitting results for resistances of various catalysts at -0.6 V *versus* RHE in a 0.5 M K_2SO_4 with 10 mM NO_3^- electrolyte according to EIS equivalent circuit diagram.

Sample tests	Resistance			
	R_s ($\Omega \text{ cm}^{-2}$)	R_{bulk} ($\Omega \text{ cm}^{-2}$)	R_{ct} ($\Omega \text{ cm}^{-2}$)	Warburg ($\Omega \text{ cm}^{-2}$)
W-MoS ₂	4.21 (0.71%)*	4.80 (0.88%)	47.5 (3.83%)	45.47 (1.96%)
MoS ₂	4.25 (0.76%)	4.83 (0.62%)	15.41 (4.25%)	/
Ag-MoS ₂	4.18 (0.45%)	4.89 (0.59%)	6.04 (3.17%)	/

* The content in parentheses after each value represents the relative standard deviation of the fitting result

b) According to Figure 3f, there exist significant peaks at low-frequency region for all the samples. This suggests that the electron transfer within electrode/electrolyte interface is the rate determining step for W-MoS₂, MoS₂, and Ag-MoS₂. For the peak frequency, compared with W-MoS₂ (0.05 Hz) and MoS₂ (0.08 Hz), the higher frequency of Ag-MoS₂ (0.4 Hz) implies a facilitated interfacial charge transfer process (*Phys. Chem. Chem. Phys.*, 2023,25, 10966-10976; *Appl. Catal. B* 2022, 304, 120993). Meanwhile, the phase value helps in identifying the frequency domains relevant to a capacitive behavior (*ACS Catal.* 2015, 5, 5292–5300). The lowest phase value of Ag-MoS₂ indicates a less charge accumulation within the IHP under negative polarization, which can be ascribed to its more positive holes' distribution. Thus, combined with ACV, Nyquist, and Bode analysis, we can conclude that Ag-MoS₂ exhibited more positive charges within IHP, to attract and enrich NO₃⁻ reactant.

Corresponding revision:

- Table S6 has been revised and updated (Page 75, Revised supplementary information).
- The descriptions of “The relative standard deviations (RSD) of the fitted parameters are provided. It is obvious that the RSD of R_s and R_{bulk} values are below 1%, while for R_{ct} and Warburg resistance are below 5%. These further verified the rationality of the selection of the physical equivalent circuit diagram in Figure S15.” have been added (Page 75, Revised supplementary information).
- The descriptions of “In Bode phase plots (Figure 3f), compared with W-MoS₂ (0.05 Hz) and MoS₂ (0.08 Hz), the higher frequency of Ag-MoS₂ (0.4 Hz) implied a facilitated interfacial charge transfer process, which can be ascribed to its intensified NO₃⁻ accumulation within the IHP region^{69,70}.” have been added (Page 11, Revised manuscript).

Comment 10: The apparent Warburg component for W-MoS₂ can be just an onset of another, unresolved “semicircle”. Given that the Nyquist plot is not orthonormal (as it should be), it is also hard to validate the apparent 45deg slope.

Response: Thanks so much for your comment. The frequency range for EIS test in the original version is set to 10⁵-10⁻¹ Hz. As the end frequency is not small enough, we have re-performed the EIS test, with the frequency region from 10⁶ to 10⁻² Hz.

As seen from the revised Figure 4d, Ag-MoS₂ and MoS₂ with solid-liquid junction exhibited typical two semicircle shape. Compared with pure MoS₂, Ag-MoS₂ exhibited smallest arc radius at low-frequency region, suggesting the fast surface catalytic kinetics. While, for W-MoS₂ with no solid-liquid junction, the Nyquist plot at low-frequency maintains the straight line with a slope close to 45°, even under the end frequency of 10⁻² Hz. It is reported that the low-frequency of 10⁻² Hz is low enough to monitor the electron transfer and surface catalytic kinetics within electrode/electrolyte interface (*Appl. Catal. B* 2017, 218, 570-580; *ACS Catal.* 2015, 5, 5292-5300.). Therefore, we consider that the NO₃RR on W-MoS₂ catalyst with no solid-liquid junction is limited by the NO₃⁻ mass transfer. This further confirms the advantage of the optimized Ag-MoS₂ for NO₃⁻ enrichment.

Figure 4. (d) Nyquist plots of the samples at -0.6 V versus RHE in 10 mM NO₃⁻ electrolyte.

Table S6. Fitting results for resistances of various catalysts at -0.6 V versus RHE in a 0.5 M K₂SO₄ with 10 mM NO₃⁻ electrolyte according to EIS equivalent circuit diagram.

Sample tests	Resistance			
	R _s (Ω cm ⁻²)	R _{bulk} (Ω cm ⁻²)	R _{ct} (Ω cm ⁻²)	Warburg (Ω cm ⁻²)
W-MoS ₂	4.21 (0.71%)*	4.80 (0.88%)	47.5 (3.83%)	45.47 (1.96%)

MoS ₂	4.25 (0.76%)	4.83 (0.62%)	15.41 (4.25%)	/
Ag-MoS ₂	4.18 (0.45%)	4.89 (0.59%)	6.04 (3.17%)	/

* The content in parentheses after each value represents the relative standard deviation of the fitting result

Corresponding revision:

- Figure 4d has been revised and updated (Page 17, Revised manuscript).
- Table S6 has been revised and updated (Page 75, Revised supplementary information).
- The descriptions of “The relative standard deviations (RSD) of the fitted parameters are provided. It is obvious that the RSD of R_s and R_{bulk} values are below 1%, while for R_{ct} and Warburg resistance are below 5%. These further verified the rationality of the selection of the physical equivalent circuit diagram in Figure S15.” have been added (Page 75, Revised supplementary information).

Comment 11: “*In comparison, the NO₂, -H, and NH₄⁺ signals were absent in the spectra of MoS₂ and W-MoS₂ (Figures S42-43). – how ammonium is then formed on these catalysts?*”

Response: We thank the reviewer for raising the question about the ammonium formed on MoS₂ and W-MoS₂ and clarified it as the following several aspects:

a) Although there are no signals of NO₂, -H, and NH₄⁺ in the Raman spectra of MoS₂ and W-MoS₂, this does not necessarily imply the absence of ammonia generation during the NO₃RR process on these two catalysts. This phenomenon has also been reported by related literatures (*Angew. Chem. Int. Ed.* 2024, 63, e202315109; *Nat. Commun.* 2022, 13, 7958.). This observation is correlated with factors such as the nitrate reduction rates, the concentration of involved intermediates and product ammonia. The *in situ* Raman spectra of MoS₂ and W-MoS₂ display the distinct N-H signals, indicating that the denitrification and hydrogenation process of nitrate occurs. According to the NO₃RR performance results (Figures 4a and 4e), both MoS₂ and W-MoS₂ generate ammonia, but their corresponding reaction rates and ammonia yield rates are significantly lower than that of Ag-MoS₂. Therefore, the accumulation of related intermediates and product

ammonium on the surfaces of MoS₂ and W-MoS₂ catalysts is relatively low, resulting in the absence of prominent corresponding signals in the Raman spectra.

Figure 4. (a) NH₃ yield rate and NH₃ FE of catalysts in 10 mM NO₃⁻ electrolyte.

Figure 4. (e) NO₃⁻ removal in initial 0.5 M K₂SO₄ with 100 mM NO₃⁻ electrolyte at -0.6 V *versus* RHE in H-cell reactor. After 3 hours of electrolysis, only 0.54 mM of NO₃⁻-N. Insert is the enlarged vision within 3 hours.

b) Furthermore, we employ online differential electrochemical mass spectrometry (DEMS) to detect relevant species (Figures 5b and S54-55), which has higher resolution in identifying and analyzing trace gaseous intermediates and products. We monitored the mass-to-charge ratio (*m/z*) signals of NH₃ (17), H₂ (2), N₂ (28), NO (30), NH₂OH (33), and N₂O (44) during the NO₃RR process. In the spectra of the three catalysts, signals for the product NH₃ and the intermediates H₂, N₂, NO, and N₂O are observed. Among them, the NH₃ signal intensity in MoS₂ and W-MoS₂ is lower than that in Ag-MoS₂, which is related to the enrichment of NO₃⁻ induced by S-L junction on the surface of Ag-MoS₂.

Thus, ammonia is generated during the NO₃RR process of MoS₂ and W-MoS₂.

Figure 5. (b) Online differential electrochemical mass spectrometry (DEMS) measurements of NO₃RR over Ag-MoS₂ under the potential of -0.6 V *versus* RHE.

Figure S54. Online differential electrochemical mass spectrometry (DEMS) measurements of NO₃RR over MoS₂ under the potential of -0.6 V *versus* RHE.

Figure S55. Online differential electrochemical mass spectrometry (DEMS) measurements of NO₃RR over W-MoS₂ under the potential of -0.6 V *versus* RHE.

Corresponding revision:

➤ The corresponding content has been revised to “In comparison, the NO₂, -H, and NH₄⁺ signals were not prominent in the spectra of MoS₂ and W-MoS₂ (Figures S52-53), indicating that the relatively slow reaction rates of the two catalysts resulted in less accumulation of related species on their surfaces.” (Page 18, Revised manuscript).

Comment 12: The reviewer has failed to find examples of chronoamperograms. It is not even clear what was the duration of the experiments. These data should be exemplified in the main text for the most important experiments, and the rest of the data should be presented in the SI.

Figure S30 is critically important and should be shown in the main text.

Response: Based on your kind suggestion, we have supplemented the chronoamperograms of the three catalysts (Ag-MoS₂, MoS₂ and W-MoS₂) at different potentials with 0.5 h duration (Figure S34). Within the potential range of -0.2 to -1.0 V vs. RHE, the current density gradually increased from ~100 to ~340 mA cm⁻², in which Ag-MoS₂ exhibits a relatively higher current density at applied potentials (Figure 4b). We have added the corresponding information and figures in both the manuscript and the supplementary information. We have also updated the original Figure S30 and added into the main text as Figure 4b.

Figure S34. Time-dependent current density curves of (a) Ag-MoS₂, (b) MoS₂, (c) W-MoS₂ at various potentials for 1800 s.

Figure 4b. I–V plots of catalysts in 0.5 M K₂SO₄ with 10 mM NO₃[−] electrolyte at various potentials for 0.5 h electrolysis.

Corresponding revision:

- Figure S34 has been added (Page 43, Revised supplementary information).
- The updated original Figure S30 has been added to Figure 4b (Page 17, Revised manuscript).
- The corresponding content has been revised to “Following the chronoamperometry measurements at various applied potentials tested in a flow cell reactor (Figure S34), the NO₃RR performance was assessed for the three catalysts with 0.5 h duration.” (Page 14, Revised manuscript).
- The corresponding content has been revised to “Of these, Ag-MoS₂ displayed the superior performance (Figures 4a-b), with a near unit NH₃ Faradaic efficiency (FE) of 99.7% at -0.6 V versus RHE (~200 mA cm⁻²) and an optimal NH₃ yield rate of 20.1 mg h⁻¹ cm⁻² at -1.0 V versus RHE (~340 mA cm⁻²). The optimal NH₃ yield rate value was about 3.4-folds and 1.7-folds than those of W-MoS₂ and MoS₂, respectively. Similarly, the peak NH₃ FE of Ag-MoS₂ was approximately 3.2-folds and 1.8-folds than those of W-MoS₂ and MoS₂, respectively. These results displayed the superior NO₃RR performance of Ag-MoS₂ compared to its counterparts.” (Page 14, Revised manuscript).

Comment 13: *Error bars are said to correspond to “relative standard deviations by three repeated tests”. While these repeated experiments are important, they are not sufficient to prove the reproducibility and reliability of the data. To prove that the*

trends are real, key materials (Ag-MoS₂ and MoS₂, at minimum) should be synthesised as three independent samples, each should be investigated, and the data should be reported as mean +/- standard deviation.

Response: Following this kind suggestion, we have reprepared three batches of catalysts (Ag-MoS₂, MoS₂ and W-MoS₂), respectively, and tested their NO₃RR performance at various potentials. The obtained data have been reported as mean values +/- standard deviations. Figure 4a presents the updated NH₃ yield rate and Faradaic efficiency of the three catalysts. The trends are not significantly different from the previously tested results, reflecting the reliability of the NO₃RR performance results. Also, we have made corresponding modifications to the data and descriptions in the manuscript.

Figure 4. (a) NH₃ yield rate and NH₃ FE of catalysts in 10 mM NO₃⁻ electrolyte.

Corresponding revision:

- Figure 4a has been revised and updated (Page 17, Revised manuscript).
- The corresponding content has been revised to “The three batches of catalysts (Ag-MoS₂, MoS₂ and W-MoS₂) were synthesized, respectively, and their NO₃RR performance at various potentials were tested. The error bars were the mean values standard deviation according to the obtained data.” (Page 22, Revised manuscript).
- The corresponding content has been revised to “Of these, Ag-MoS₂ displayed the superior performance (Figures 4a-b), with a near unit NH₃ Faradaic efficiency (FE) of 99.7% at -0.6 V versus RHE (~200 mA cm⁻²) and an optimal NH₃ yield rate of 20.1 mg

$\text{h}^{-1} \text{cm}^{-2}$ at -1.0 V *versus* RHE ($\sim 340 \text{ mA cm}^{-2}$). The optimal NH_3 yield rate value was about 3.4-folds and 1.7-folds than those of W-MoS₂ and MoS₂, respectively. Similarly, the peak NH_3 FE of Ag-MoS₂ was approximately 3.2-folds and 1.8-folds than those of W-MoS₂ and MoS₂, respectively. These results displayed the superior NO₃RR performance of Ag-MoS₂ compared to its counterparts.” (Page 14, Revised manuscript).

We hope that our efforts are sufficient to answer the concerns of the reviewer. Solving these questions has been very helpful to further improve the depth and scientific nature of our manuscript.

Reviewer #3:

This study explores the use of a p-type Ag-MoS₂ catalyst for the electrochemical nitrate reduction reaction (NO₃RR). Ammonia production via NO₃RR is a crucial topic for energy conversion and decarbonizing chemical manufacturing, as it plays a key role in balancing the global nitrogen cycle and reducing carbon emissions. While most previous works have studied on NO₃RR catalysts under relatively high nitrate concentrations, the Ag-MoS₂ catalyst presented here demonstrates decent performance under low nitrate concentrations, which is particularly advantageous for real wastewater treatment. However, the central claim that a p-type catalyst can enhance local nitrate concentration is not entirely reasonable, and several other concerns remain unaddressed in the manuscript. Therefore, this work is not recommended for publication in Nature Communications at least in its current form.

Response: Thanks so much for the reviewer's comment. Your comments play a crucial role in improving the quality of our paper. In response to the raised questions, we have made extensive revisions to the article, including **1)** re-performed the NO₃⁻ adsorption tests of the three samples at various applied potential, **2)** revised COMSOL simulations and the EIS tests, **3)** supplemented XANES results and analysis, **4)** and provided additional discussions about the DFT calculations. We hope our modifications can fulfill the requirements of the reviewer.

***Comment 1:** The central claim of this work—that a p-type catalyst can generate positive charges at the solid-liquid junction to attract nitrate ions, making it an effective catalyst for NO₃RR, especially under low nitrate concentrations—is highly questionable. While the authors have provided sufficient analysis showing that Ag-MoS₂ exhibits p-type characteristics and can produce more positive charges at the S-L junction compared to MoS₂ and W-MoS₂ under OCV, which I find convincing, this only implies that Ag-MoS₂ might attract more nitrate ions when all three catalysts are operated under the same reduction potential. However, this does not necessarily mean that Ag-MoS₂ will be a better NO₃RR catalyst. I would challenge the authors with the*

argument that by applying a less negative potential to MoS₂, it could achieve a similar charging environment in the IHP, thereby attracting the same amount of nitrate ions as Ag-MoS₂.

Therefore, the real question is: when all the catalysts are operated at their respective optimal potentials (where they can all attract sufficient nitrate ions for the reaction), which catalyst structure exhibits the lowest energy barrier in the 8-step electron transfer process of NO₃RR? This crucial point, however, is not discussed in the manuscript. The only relevant data provided is in Fig. 5e, but it does not include a comparison with the other two catalysts, nor does it address the optimal applied potentials required to attract sufficient nitrate or the energy barriers under those conditions.

If the authors agree with the idea above, a reasonable logical flow could be: 1) p-type/n-type catalysts create different charges at the solid-liquid junction, 2) therefore, the three catalysts require different applied biases to attract sufficient nitrate ions, 3) under the desired applied potentials, one catalyst (presumably Ag-MoS₂) has the lowest energy barrier for NO₃RR, making it the most effective catalyst among the three. The authors should restructure the whole manuscript and accordingly, with additional experimental and simulation details to support and validate these claims.

Response: Thanks so much for your professional comments and inspiring ideas. In the present work, we only measure the NO₃⁻ adsorption ability on the three catalysts under the potential of -0.6 V vs. RHE. We agree with the reviewer's opinion that the NO₃⁻ enrichment effect on MoS₂ and W-MoS₂ may not peak at this potential. Hence, we have supplemented the NO₃⁻ adsorption experiments of the three samples under different applied potentials. Meanwhile, some descriptions and discussion have been revised and added. The questions raised by the reviewer are answered by the following two parts:

a) According to your suggestion, we have supplemented the NO₃⁻ enrichment capacity of the three catalysts measured by ion chromatography (IC) at different applied potentials (from -0.2 to -1.0 V vs. RHE). The maximum adsorbed NO₃⁻ concentration of the three samples occurs at different potentials (Figure 3g), that is, -0.6 V vs. RHE

for Ag-MoS₂, -0.8 V vs. RHE for MoS₂, and -1.0 V vs. RHE for W-MoS₂. The most positive Fermi level position (E_F) lead to the highest potential difference between the E_F and the E_{redox} ($E_F - E_{\text{redox}}$) in Ag-MoS₂. Thus, a relatively low applied potential is required to achieve the surface band bending within S-L junction, thus the maximum NO₃⁻ concentration for Ag-MoS₂ (Figures S10-11).

Figure 3. (g) The concentration of NO₃⁻ adsorbed on W-MoS₂, MoS₂, and Ag-MoS₂ at various applied potentials in the containing NO₃⁻ electrolyte, tested by ion chromatography.

Figure S10. (a) Interfacial thermal equilibrium process when the Ag-MoS₂ contact with the NO₃⁻ contained solution. Downward band bending at solid-liquid (S-L) junction under (b) no potential and (c) reduction potential. E_{redox} and E_F represent the theoretical redox potential of NO₃⁻/NH₃ and the Fermi level potential. CBM and VBM express the conduction band minimum and valence band maximum of semiconductor, respectively.

Figure S11. (a) Interfacial thermal equilibrium process when MoS₂ contact with the NO₃⁻ contained solution. Downward band bending at solid-liquid (S-L) junction under (b) no potential and (c)

reduction potential.

b) While, although the maximum adsorbed NO_3^- concentration of MoS_2 and W-MoS_2 locate at the potential of -0.8 and -1.0 V vs. RHE, their absolute values are much lower than that of Ag-MoS_2 . This can be ascribed to the different carrier concentration which dominates the surface band bending degree, which ultimately influence the charging in IHP and the NO_3^- enrichment.

Specifically, the carrier concentrations of the three samples are obtained through Mott-Schottky plots (Figure 3a). As the slope of Ag-MoS_2 is much lower than those of MoS_2 and W-MoS_2 , the calculated carrier concentration of Ag-MoS_2 is $1.14 \times 10^{19} \text{ cm}^{-3}$ (Table S2), 1.8 and 3.5 times of MoS_2 ($6.3 \times 10^{18} \text{ cm}^{-3}$) and W-MoS_2 ($3.25 \times 10^{18} \text{ cm}^{-3}$). A high carrier concentration can induce intensified surface band bending when the Ag-MoS_2 is contacted with the NO_3^- contained electrolyte (*Adv. Mater.* 2024, 36, 2403253; *Angew. Chem. Int. Ed.* 2021, 60, 5505.). Thus, based on the more positive Fermi level position and higher carrier concentration of Ag-MoS_2 , it exhibits a much higher NO_3^- enrichment ability at a more positive applied potential.

Based on the above analysis, we have supplemented corresponded experimental results, and modified some descriptions and discussions within the article.

Figure 3a. Mott-Schottky plots of Ag-MoS_2 , MoS_2 , and W-MoS_2 catalysts.

Table S2. Calculated carrier concentration of the three samples by M-S plots.

Samples	Carrier concentration (cm^{-3})
W-MoS_2	3.25×10^{18}
MoS_2	6.3×10^{18}

Corresponding revision:

- Figure 3a has been revised and updated (Page 13, Revised manuscript).
- Table S2 has been added (Page 71, Revised supplementary information).
- The descriptions have been added “The concentration of NO₃⁻ adsorbed on Ag-MoS₂ presented a normal distribution trend as the potential ranging from -0.2 to -1.0 V *versus* RHE, and reached the peak at -0.6 V *versus* RHE (90.7 μg mL⁻¹). Due to the different Fermi level position, the adsorbed NO₃⁻ concentration of MoS₂ and W-MoS₂ peaked at more negative potentials of -0.8 and -1.0 V *versus* RHE, respectively. As the Ag dopant acted as the electron acceptor to increase the carrier concentration (Table S2), it induced an intensified surface band bending when the Ag-MoS₂ was contacted with the NO₃⁻ contained electrolyte. Thus, the maximum adsorbed NO₃⁻ concentration of Ag-MoS₂ was approximately ~15.8-fold and ~6.3-fold than those of W-MoS₂ (5.7 μg mL⁻¹, -1.0 V *versus* RHE) and MoS₂ (14.4 μg mL⁻¹, -0.8 V *versus* RHE), respectively.” (Pages 11-12, Revised manuscript).
- The descriptions have been added “M-S plots further revealed the Fermi level potential (E_F) of Ag-MoS₂ (1.07 V *versus* RHE) and MoS₂ (0.53 V *versus* RHE),” (Page 9, Revised manuscript).
- The descriptions have been added “Specifically, the carrier concentrations of the three samples were obtained through Mott-Schottky plots (Figure 3a). As the slope of Ag-MoS₂ is much lower than those of MoS₂ and W-MoS₂, the calculated carrier concentration of Ag-MoS₂ was 1.14×10¹⁹ cm⁻³, 1.8 and 3.5 times of MoS₂ (6.3×10¹⁸ cm⁻³) and W-MoS₂ (3.25×10¹⁸ cm⁻³). A high carrier concentration can induce intensified surface band bending when the Ag-MoS₂ was contacted with the NO₃⁻ contained electrolyte.” (Page 71, Revised supplementary information).

Some other concerns:

Comment 2: *Relative to the points above, the manuscript does not clearly identify which step in the NO₃RR process is considered the rate-determining step.*

Response: Thanks so much for your comment. We have calculated the reaction pathways of NO₃RR on the three samples, as seen from Figures S58-60. Specifically, the potential-determining step (PDS) of the three samples are all *NH₂→*NH₃, with the Gibbs free energy change (ΔG) of 1.1 (Ag-MoS₂), 0.95 (MoS₂), and 1.02 eV (W-MoS₂). The ΔG of Ag-MoS₂ is comparable to those of W-MoS₂ and MoS₂, while the optimal adsorbed NO₃⁻ concentration of Ag-MoS₂ is ~15.8 and ~6.3 times higher than those of the counterparts (Figure 3g). Thus, NO₃⁻ enrichment effect could act as the main contributor to enhance the NO₃RR performance in Ag-MoS₂. We have added corresponding descriptions about the PDS within the whole article.

Figure S58. Gibbs free energy diagram of various intermediates generated during NO₃RR over Ag-MoS₂ at pH = 7.

Figure S59. Gibbs free energy diagram of various intermediates generated during NO₃RR over MoS₂ at pH = 7.

Figure S60. Gibbs free energy diagram of various intermediates generated during NO_3RR over W-MoS_2 at $\text{pH} = 7$.

Figure 3. (g) The concentration of NO_3^- adsorbed on W-MoS_2 , MoS_2 , and Ag-MoS_2 at various applied potentials in the containing NO_3^- electrolyte, tested by ion chromatography.

Corresponding revision:

➤ The corresponding content has been revised to “All the three samples underwent the process of adsorption of NO_3^- , deoxygenation of the N species, and hydrogenation of the N species to synthesize NH_3 , in which the $*\text{NH}_2 \rightarrow *\text{NH}_3$ process was the potential-determining step (PDS). The Gibbs free energy change (ΔG) of this step was 1.1, 0.95, and 1.02 eV for Ag-MoS_2 , MoS_2 , and W-MoS_2 catalysts. The ΔG of Ag-MoS_2 was comparable to those of W-MoS_2 and MoS_2 , while the optimal adsorbed NO_3^- concentration of Ag-MoS_2 was ~ 15.8 and ~ 6.3 times higher than those of the counterparts (Figure 3g). This further verified the decisive role of surface NO_3^- enrichment effect for promoting NO_3RR activity of Ag-MoS_2 .” (Pages 18-19, Revised manuscript).

Comment 3: *In Fig. 3g, the authors compared the three catalysts only at the same voltage. It would be more informative to also test the other two catalysts across different potentials, as they may have different optimal voltages for nitrate adsorption.*

Response: Based on this inspiring suggestion, we have supplemented the nitrate adsorption of the other two catalysts across different potentials, and made corresponding modifications to the descriptions in the manuscript.

As the reviewer mentioned, with the variation of applied potential ranging from -0.2 to -1.0 V vs. RHE, the optimal potentials for nitrate adsorption are different among the three catalysts. Specifically, Ag-MoS₂ exhibits the optimal adsorption potential at -0.6 V vs. RHE, while MoS₂ and W-MoS₂ at -0.8 V and -1.0 V vs. RHE, respectively (Figure 3g).

However, the nitrate adsorption concentrations of MoS₂ and W-MoS₂ at their optimal potentials are 14.4 and 5.7 μg mL⁻¹, respectively, which are substantially lower than the peak nitrate adsorption for Ag-MoS₂ (90.7 μg mL⁻¹). These phenomena originate from the differences in surface energy band bending degree among the three samples. According to the M-S plots (Figure 3a), the calculated carrier concentration of W-MoS₂, MoS₂, and Ag-MoS₂ are 3.25×10¹⁸, 6.3×10¹⁸, and 1.14×10¹⁹ cm⁻³ (Table S2), respectively. Notably, Ag-MoS₂ has a significantly higher carrier concentration compared to its counterparts (an order of magnitude greater), rendering its surface band bending more susceptible to changes under the drive of an applied potential. Although the nitrate adsorption of W-MoS₂ and MoS₂ also peaks at the more negative potential, their surface band bending degree is constrained by their relatively lower carrier concentrations. A high band bending degree contributes to intensified S-L junction, thereby promoting the enrichment of nitrate in the IHP of Ag-MoS₂. Hence, the adsorption capacity of Ag-MoS₂ for nitrate significantly exceeds that of its counterparts across various potentials.

Figure 3. (g) The concentration of NO₃⁻ adsorbed on W-MoS₂, MoS₂, and Ag-MoS₂ at various potentials, tested by ion chromatography.

Figure 3a. Mott-Schottky plots of Ag-MoS₂, MoS₂, and W-MoS₂ catalysts.

Table S2. Calculated carrier concentration of the three samples by M-S plots.

Samples	Carrier concentration (cm ⁻³)
W-MoS ₂	3.25×10 ¹⁸
MoS ₂	6.3×10 ¹⁸
Ag-MoS ₂	1.14×10 ¹⁹

Corresponding revision:

- Figure 3g has been revised (Page 13, Revised manuscript).
- The corresponding content “The concentration of NO₃⁻ adsorbed on Ag-MoS₂ presented a normal distribution trend as the potential ranging from -0.2 to -1.0 V *versus* RHE, and reached the peak at -0.6 V *versus* RHE (90.7 µg mL⁻¹). Due to the different Fermi level position, the adsorbed NO₃⁻ concentration of MoS₂ and W-MoS₂ peaked at

more negative potentials of -0.8 and -1.0 V *versus* RHE, respectively. As the Ag dopant acted as the electron acceptor to increase the carrier concentration (Table S2), it induced an intensified surface band bending when the Ag-MoS₂ was contacted with the NO₃⁻ contained electrolyte. Thus, the maximum adsorbed NO₃⁻ concentration of Ag-MoS₂ was approximately ~15.8-fold and ~6.3-fold than those of W-MoS₂ (5.7 μg mL⁻¹, -1.0 V *versus* RHE) and MoS₂ (14.4 μg mL⁻¹, -0.8 V *versus* RHE), respectively.” has been added (Pages 11-12, Revised manuscript).

- Table S2 has been added (Page 71, Revised supplementary information).
- The descriptions of “Specifically, the carrier concentrations of the three samples were obtained through Mott-Schottky plots (Figure 3a). As the slope of Ag-MoS₂ is much lower than those of MoS₂ and W-MoS₂, the calculated carrier concentration of Ag-MoS₂ was $1.14 \times 10^{19} \text{ cm}^{-3}$, 1.8 and 3.5 times of MoS₂ ($6.3 \times 10^{18} \text{ cm}^{-3}$) and W-MoS₂ ($3.25 \times 10^{18} \text{ cm}^{-3}$). A high carrier concentration can induce intensified surface band bending when the Ag-MoS₂ was contacted with the NO₃⁻ contained electrolyte.” have been added (Page 71, Revised supplementary information).

Comment 4: The authors provided only EXAFS data in Fig. 3f, while XANES analysis for the three catalysts is crucial for validating their electronic structures.

Response: Thanks for raising the concern about the XANES spectra. We have supplemented the data and added the corresponding description in the manuscript.

As shown in Mo K-edge X-ray absorption near edge structure (XANES) spectra (Figure S19a), Ag-MoS₂, MoS₂, and W-MoS₂ exhibit the higher pre-edge absorption energy than Mo foil, suggesting positively charged Mo atoms. In comparison to MoS₂, the absorption edge of Ag-MoS₂ undergoes a higher energy shift, implying decreased electron density on the Mo site due to Ag doping. Whereas, W-MoS₂ presents a lower pre-edge absorption energy, indicating increased electron density on the Mo site upon W incorporation. From the Ag K-edge XANES spectra (Figure S6b), the absorption energy of Ag-MoS₂ located between those of Ag foil and Ag₂S references, indicating the valence state of the doped Ag within 0 to 1. As shown in W L-edge XANES spectra

(Figure S19b), a higher pre-edge absorption edge energy of W-MoS₂ than W foil and WS₂ references reveals the valence state of the W species that exceeds +4. Therefore, there is a significant electronic interaction between the introduced doping species and MoS₂.

Figure S19. (a) Mo K-edge XANES spectra, (b) W L-edge XANES spectra of catalysts and reference samples.

Figure S6. (b) Ag K-edge XANES spectra of Ag-MoS₂ and reference samples.

Corresponding revision:

➤ Figures S6b and S19 have been added (Pages 15 and 28, Revised supplementary information).

➤ The corresponding content “The Mo K-edge X-ray absorption near edge structure (XANES) spectra exhibited the higher pre-edge absorption energy of Ag-MoS₂ than those of Mo foil and MoS₂ (Figure S6a), implying that the doping Ag species induced decreased electron density on the Mo site. From the Ag K-edge XANES spectra, the

absorption energy of Ag-MoS₂ located between those of Ag foil and Ag₂S references (Figure S6b), indicating the valence state of the doped Ag within 0 to 1.” has been added (Page 8, Revised manuscript).

➤ The corresponding content “As shown in Mo K-edge X-ray absorption near edge structure (XANES) spectra (Figure S19a), W-MoS₂ exhibited the higher pre-edge absorption energy than Mo foil, suggesting positively charged Mo atoms. In comparison to MoS₂, the absorption edge of W-MoS₂ underwent a lower energy shift, implying increased electron density on the Mo site upon W incorporation. From the W L-edge XANES spectra (Figure S19b), a higher pre-edge absorption edge energy of W-MoS₂ than W foil and WS₂ references reveals the valence state of the W species that exceeds +4. Therefore, there is a significant electronic interaction between the introduced doping species and MoS₂.” has been added (Page 28, Revised supplementary information).

***Comment 5:** The Comsol simulation section lacks sufficient detail and explanation. Specifically, the inputs used to simulate the surface boundary parameters for the three catalysts are not provided.*

Response: Thanks so much for your constructive comments. According to your suggestion, we have supplemented and regulated the parameters of surface boundary and length scales, etc. The revised COMSOL simulation results are updated in Figure S28.

In specific, the thickness of the diffusion layer is assumed to be 200 μm. For the ion concentration boundary condition, the ion concentration at the upper boundary of the model is set to the concentration of the native solution, and the lower boundary is set to no flux. For the electrostatic field boundary conditions, the upper boundary is set to 0 V, and the lower boundary was set to -0.6, -0.8, and -1.0 V, respectively, depending on the degree of electric field reversal potential built into different materials.

From the updated simulation results (Figure S28), the concentrations of NO₃⁻ distributed on the Ag-MoS₂, MoS₂ and W- MoS₂ catalyst surface are 5.1, 4.0, and 3.2

mM, respectively. It is evident that the concentrations of NO_3^- distributed on the Ag-MoS₂ surface is higher than that on its counterpart surfaces, which matches the experimental trends of *in situ* Raman spectroscopy, Bode analysis, elemental mapping, and ion chromatography. Correspondingly, we have added the relevant details and explanations to the manuscript.

Figure S28. The NO_3^- distribution at the solid-liquid interface of (a) Ag-MoS₂, (b) MoS₂, and (c) W-MoS₂ catalysts through COMSOL simulations.

Corresponding revision:

- Figure S28 has been updated (Page 37, Revised supplementary information).
- The corresponding content “From the simulation results (Figure S28), the concentrations of NO_3^- distributed on the Ag-MoS₂, MoS₂ and W-MoS₂ catalyst surface were 5.1, 4.0, and 3.2 mM, respectively. It was evident that the concentrations of NO_3^- distributed on the Ag-MoS₂ surface was higher than that on its counterpart surfaces, which matched the experimental trends of *in situ* Raman spectroscopy, Bode analysis, elemental mapping, ion chromatography.” has been added (Page 37, Revised supplementary information).
- The corresponding content has been revised to “The thickness of the diffusion layer was assumed to be 200 μm . For the ion concentration boundary condition, the ion concentration at the upper boundary of the model was set to the concentration of the native solution, and the lower boundary was set to no flux. For the electrostatic field boundary conditions, the upper boundary was set to 0 V, and the lower boundary was set to -0.6, -0.8, and -1.0 V, respectively, depending on the degree of electric field reversal potential built into different materials.” (Page 5, Revised supplementary information).

Comment 6: *The authors state that the Nyquist plot for W-MoS₂ “showed a straight line with a slope of 45°, suggesting that NO₃⁻ mass transfer was the rate-limiting step for NO₃RR.” This conclusion requires further clarification. Additionally, to confirm whether the plot exhibits a straight line rather than a large semicircle, a scan at lower frequencies should be conducted.*

Response: Thanks so much for your comment. The frequency range for EIS test in the original version is set to 10⁵-10⁻¹ Hz. As the end frequency is not small enough, we have re-preformed the EIS test, with the frequency region from 10⁶ to 10⁻² Hz.

As seen from the revised Figure 4d, Ag-MoS₂ and MoS₂ with solid-liquid junction exhibited typical two semicircle shape. Compared with pure MoS₂, Ag-MoS₂ exhibited smallest arc radius at low-frequency region, suggesting the fast surface catalytic kinetics. While, for W-MoS₂ with no solid-liquid junction, the Nyquist plot at low-frequency maintains the straight line with a slope close to 45°, even under the end frequency of 10⁻² Hz. It is reported that the low-frequency of 10⁻² Hz is low enough to monitor the electron transfer and surface catalytic kinetics within electrode/electrolyte interface (*Appl. Catal. B* 2017, 218, 570-580; *ACS Catal.* 2015, 5, 5292-5300.). Therefore, we consider that the NO₃RR on W-MoS₂ catalyst with no solid-liquid junction is limited by the NO₃⁻ mass transfer. This further confirms the advantage of the optimized Ag-MoS₂ for NO₃⁻ enrichment.

Figure 4. (d) Nyquist plots of the samples at -0.6 V versus RHE in 10 mM NO₃⁻ electrolyte.

Table S6. Fitting results for resistances of various catalysts at -0.6 V versus RHE in a 0.5 M K₂SO₄ with 10 mM NO₃⁻ electrolyte according to EIS equivalent circuit diagram.

Sample tests	Resistance			
	R _s (Ω cm ⁻²)	R _{bulk} (Ω cm ⁻²)	R _{ct} (Ω cm ⁻²)	Warburg (Ω cm ⁻²)
W-MoS ₂	4.21 (0.71%)*	4.80 (0.88%)	47.5 (3.83%)	45.47 (1.96%)
MoS ₂	4.25 (0.76%)	4.83 (0.62%)	15.41 (4.25%)	/
Ag-MoS ₂	4.18 (0.45%)	4.89 (0.59%)	6.04 (3.17%)	/

* The content in parentheses after each value represents the relative standard deviation of the fitting result

Corresponding revision:

- Figure 4d has been revised and updated (Page 17, Revised manuscript).
- Table S6 has been revised and updated (Page 75, Revised supplementary information).
- The descriptions of “The relative standard deviations (RSD) of the fitted parameters are provided. It is obvious that the RSD of R_s and R_{bulk} values are below 1%, while for R_{ct} and Warburg resistance are below 5%. These further verified the rationality of the selection of the physical equivalent circuit diagram in Figure S15.” have been added (Page 75, Revised supplementary information).

Comment 7: *The authors state, “We observed a nearly constant reaction rate within a NO₃⁻-N concentration from 1400 μg/mL (100 mM) to 7.6 μg/mL (0.54 mM) (indicated by the slope of the cyan curve in Figure 4e), demonstrating the excellent performance of Ag-MoS₂.” This statement is ambiguous. Most studies suggest that NO₃RR follows a pseudo-first-order reaction and is dependent on nitrate concentrations. If the reaction rate is constant, do the authors imply a zero-order reaction, suggesting that it may be independent of nitrate concentration? This conclusion should be examined more carefully.*

Response: Thanks so much for your constructive comments. We agree with the reviewer's opinion that NO₃RR follows a pseudo-first-order reaction and is dependent on nitrate concentrations. In general, as the NO₃RR reaction progresses, the reactant NO₃⁻ concentration gradually decreases, and the reaction rate reduces correspondingly. Thus, in this work, we design a p-type Ag-MoS₂ catalyst to enrich NO₃⁻ under low-concentration condition, for benefiting NO₃RR performance. In the enlarged Figure 4e, it is clear that within 3 hours, the relationship between residual NO₃⁻ concentration and treatment time still follows the pseudo-first-order kinetics, and the concentration decreases sharply from the original high level of 100 mM to low level of 0.54 mM. While for the two counterparts, the residual NO₃⁻ remains a relatively high level of 4.64 mM and 11.78 mM even after 48 hours.

Due to the superior NO₃⁻ enrichment effect of Ag-MoS₂, the fast reaction rate cause to the sharp decrease of NO₃⁻ reactant within a short period of time (3 hours). As the reviewer pointed, the NO₃RR still follows the pseudo-first-order reaction and is dependent on nitrate concentrations. The difference between the Ag-MoS₂ and the counterparts is the facilitated reaction rate.

Thus, we acknowledge that the present description may cause misunderstanding to the readers. Thus, we have revised corresponding sentences to make the conclusions more rigorous.

Figure 4. (e) NO₃⁻ removal in initial 0.5 M K₂SO₄ with 100 mM NO₃⁻ electrolyte at -0.6 V *versus* RHE in H-cell reactor. After 3 hours of electrolysis, only 0.54 mM of NO₃⁻-N. Insert is the enlarged vision within 3 hours.

Corresponding revision:

- Figure 4e has been revised and updated (Page 17, Revised manuscript).
- The corresponding content has been revised to “Impressively, the NO_3RR on the three samples followed typical quasi-first-order kinetics, and the NO_3^- to NH_3 selectivity of Ag-MoS₂ was up to 99.3% within only 3 hours of electrolysis at -0.6 V versus RHE (NO_3^- concentration decreased sharply from 100 mM to 0.54 mM, Figures 4e and S46), manifesting that nearly all the N sources were converted into NH_3 (Figure S47). Comparatively, the NO_3^- to NH_3 conversion efficiency of W-MoS₂ and MoS₂ were as low as ~16% and ~30%, and the residual NO_3^- still remained a relatively high level of 4.64 mM and 11.78 mM even after 48 hours (Figure S48).” (Pages 15-16, Revised manuscript).

We hope that our efforts can satisfy the concerns of the reviewer. Thanks so much again for your constructive comments that helped us improving the quality of the manuscript.

Manuscript number: NCOMMS-24-49780A-Z

Title: Near-Unity Nitrate to Ammonia Conversion via Reactant Enrichment at the Solid-Liquid Interface

A Point-to-Point Response to Reviewer's Comments

Dear Editor and Reviewers,

Thank you for taking time and effort to carefully examine our manuscript. The comments are highly appreciated and helpful to improve this work. We have incorporated the corresponding changes into the manuscript (highlighted in the revised manuscript) and supplementary information in order to address the editor's concerns and the requests of the reviewers. These changes are specified and discussed in a point-to-point response to the reviewers' comments, as shown below:

Reviewer #1:

After carefully reviewing the rebuttal and revisions, my concerns have grown further. The foundational aspects of the manuscript appear to be unreliable. Specifically, I have issues with the following key points: (1) the formation of band bending, (2) the misinterpretation of hole behavior and surface anion properties, and (3) the explanation of charge transfer direction and conduction mechanisms. I will outline my concerns in more detail below. Overall, I am not in agreement with the authors' proposed mechanisms or their key findings.

Response: Thank you very much for your detailed comments, which play crucial role in improving the quality of our paper. We apologize for the misunderstanding of some concepts and mechanisms in the previous response letter. Concerning the three key points you have raised and concerned, we have carried out corresponding experiments and supplemented additional explanations, hoping to provide appropriate mechanism for Ag-MoS₂ induced NO₃⁻ enrichment within IHP. The detailed descriptions and explanations are illustrated in the response for each comment you have raised.

Comment 1: The authors report work functions of 4.98 eV for Ag-MoS₂ and 4.92 eV for MoS₂ using the Kelvin probe. However, they subsequently state that the Fermi levels of Ag-MoS₂ and MoS₂ are 1.07 V and 0.53 V vs. RHE, respectively. The discrepancy between these values of two samples is too large to be physically plausible. Additionally, the authors claim that the potential difference between the Fermi level (E_f) and the redox potential (E_{redox}) triggers charge transfer from the inner Helmholtz plane to the materials. This statement forms the cornerstone of their argument for band bending. However, in the Supplementary information, the authors assert that the Fermi level (E_F) of a p-type semiconductor is higher than the redox potential of NO₃⁻/NH₃ ($E_{redox} = 0.27$ V vs. RHE), suggesting electron transfer from the electrolyte to the surface of Ag-MoS₂ and MoS₂. This raises a critical question: how can NO₃⁻ donate electrons to Ag-MoS₂ and MoS₂? The authors do not address the underlying mechanism of this electron donation, and they overlook the importance of work function differences and the oxidation of NO₃⁻ in constructing the band bending.

Response: Thanks so much for your professional and insightful comments. In response to your questions, we have supplemented corresponding experiments and added the discussions and explanations. The answers are as follows:

a) Additional experiments for KPFM measurements. The Fermi level position difference between Ag-MoS₂ and MoS₂ is 0.54 V by Mott-Schottky (M-S) plots. While, the increase in work function after Ag doping is not that obvious through previous KPFM results (from 4.92 to 4.98 eV). This may be caused by the difference in testing conditions between these two characterizations, as well as the uneven sample preparation and deviation in sample selection.

For M-S measurements, series MoS₂ electrodes are immersed in the nitrate-containing electrolyte, inducing surface band bending under the drive of potential difference between the Fermi level of semiconductor and the redox pair (NO₃⁻/NH₃) in solution. Within the M-S measurement, external applied bias is provided from the electrochemical workstation to eliminate the surface band bending (Photoelectrochemical water splitting, *New York: Springer* 2013, 344), and the Fermi

level position can be obtained based on the applied bias value (which is the intersection point of curve slope and X-axis). Under this condition, the electrode is fully contacted with the electrolyte, and the obtained Fermi level position is the macroscopic statistical results of all micro MoS₂ nanoflowers (which highly dispersed on the electrode).

While, for KPFM measurements, series MoS₂ powder catalysts are dispersed in isopropanol solution, and the solution is loaded on the conductive FTO substrate for subsequent test. This characterization for work function calculation is more microscopic and local, usually with the scale of several hundred nanometers. In the previous KPFM test, due to the uneven sample preparation and deviation in sample selection, we find that the work function of Ag-MoS₂ is only 0.06 eV greater than that of MoS₂. This result differs significantly from those of M-S and UPS. Thus, we have re-performed the KPFM test, in which the samples are highly dispersed on the conductive substrate using spin-coating method (adhere to sample preparation standards for KPFM test). According to the related literatures, the work function of pure MoS₂ is around ~5.0 eV (*ACS Appl. Mater. Interfaces* 2017, 9, 4, 3223–3245; *J. Power Sources* 2016, 319, 1–8; *ACS Nano* 2016, 10, 6, 6100–6107), which is close to the KPFM and UPS results in the present work. Then, we have retested the KPFM for three branches of Ag-MoS₂ samples. As seen from Figure S9, the work functions of the three branches of Ag-MoS₂ are 5.12 eV, 5.13 eV, and 5.15 eV, respectively. Although the work function differences by KPFM tests (0.2 eV, 0.21 eV, and 0.23 eV) are not that much with those of UPS and M-S, the increased work function can help to verify the positive effect of Ag doping, to induce the downward shift of Fermi level position.

Figure S9. (a-c) Surface potential and (d-e) corresponding line profiles along the arrow lines for the three branches of Ag-MoS₂.

Corresponding Revisions

- Revised Manuscript (Page 9, highlighted in bright yellow): “The surface potential difference between Ag-MoS₂ and FTO substrate was ~230 mV, while 20 mV for MoS₂. Correspondingly, the work functions of Ag-MoS₂ and MoS₂ were 5.13 and 4.92 eV, respectively.”.
- Revised Supplementary information (Page 17, highlighted in bright yellow): “We have tested the KPFM for pure MoS₂ sample. As seen from Figure S8, the surface potential distribution within MoS₂ and FTO substrate region differed, in which the surface potential difference between MoS₂ and FTO substrate was 20 mV. As the work function of FTO was 4.9 eV, the work function of MoS₂ obtained from KPFM technique was 4.92 eV.”.
- Revised Supplementary information (Page 18, highlighted in bright yellow): “We have also tested the KPFM for three branches of Ag-MoS₂ samples. As seen from Figure S9, the surface potential differences between Ag-MoS₂ and FTO substrate were 220, 230, and 250 mV, respectively. Thus, the work functions of the three branches of Ag-MoS₂ were 5.12 eV, 5.13 eV, and 5.15 eV. The work function difference between Ag-MoS₂ and MoS₂ was around 0.21 eV. Although the work function difference by

KPFM test was not that much with UPS and M-S plots, the increased work function can help to verify the positive effect of Ag doping to induce the downward shift of Fermi level position.”.

➤ Revised Supplementary information (Page 18, highlighted in bright yellow): Figure S9 has been revised and updated.

b) Charge transfer within the interface between semiconductor and electrolyte. To address your concerns and clearly explain the mechanism for our present work, we have systematically searched the literatures and carefully studied the charge transfer at the semiconductor-solution interface. According to the related literatures (*Chem. Rev.* 1992, 92, 411-433; *Crit. Rev. Solid State Mater. Sci.* 1980, 10, 1-41), Figure S10 displays the energy level vs. position representation of the semiconductor-electrolyte interface for p-type semiconductor electrode in contact with the electrolyte solution.

Figure S10. The energy level vs. position representation of the semiconductor-electrolyte interface for p-type semiconductor electrode (a) before and (b) after contact with the electrolyte solution.

For typical p-type catalyst, the Fermi level position is close to the valance band maximum (VBM), which exhibits more positive potential than the redox potential of $\text{NO}_3^-/\text{NH}_3$ pair. On the electrolyte solution side, the oxidized redox species (D_{ox}) occupy the region above E_{redox} , while the reduced redox species (D_{red}) occupy the region below E_{redox} . The distribution levels for the D_{red} and D_{ox} in electrolyte are similar to the

VB and CB in semiconductors.

As the p-type semiconductor exhibits the characteristic of hole conduction, when the p-type catalyst is immersed in the NO_3^- -containing electrolyte, the potential difference between E_F and E_{redox} induces holes (as the majority carriers in p-type semiconductor) transfer from catalyst surface to the D_{ox} of solution side. As a result, downward band bending occurs on the semiconductor surface, forming space charge layer with hole depletion.

Thus, we consider the charge transfer within semiconductor-electrolyte interface is dominated by the hole transfer from semiconductor surface to the electrolyte side, instead of the electron donation from NO_3^- . We found that the descriptions about the “electron transfer” at semiconductor-electrolyte interface is inappropriate, and these incorrect statements are revised within the whole article.

Corresponding Revisions

- Revised Manuscript (Page 9, highlighted in bright yellow): “Upon contact with the NO_3^- -containing electrolyte, the potential difference between the E_F and the E_{redox} ($E_F - E_{\text{redox}}$) triggered charge transfer between the catalyst surface and the inner Helmholtz plane (IHP), forming positively charged IHP on the electrolyte side (Figure S10).”.
- Revised Manuscript (Page 9, highlighted in bright yellow): “Specifically, p-type MoS_2 catalysts exhibited the characteristic of hole conduction. Under the drive of $E_F - E_{\text{redox}}$, the holes (majority carrier) in valance band would transfer from semiconductor surface to electrolyte, to induce the charge rearrangement within IHP region (Figure S10a)⁵⁴.”.
- Revised Supplementary information (Page 9, highlighted in bright yellow): “For typical p-type catalyst, the Fermi level position of Ag-MoS_2 is close to the valance band maximum (VBM), which exhibits more positive potential than the redox potential of $\text{NO}_3^-/\text{NH}_3$ pair. As the p-type Ag-MoS_2 exhibits the characteristic of hole conduction, when the Ag-MoS_2 is immersed in the NO_3^- -containing electrolyte, the potential difference between E_F and E_{redox} induces holes (as the majority carriers in p-type semiconductor) transfer from catalyst surface to the IHP of solution side. As a result,

downward band bending occurs on the semiconductor surface, forming space charge layer with hole depletion. And the positively charged IHP could attract nitrate anions under electrostatic interaction.”.

➤ Revised Supplementary information (Page 19, highlighted in bright yellow): “The energy level *vs.* position representation of the semiconductor-electrolyte interface for p-type semiconductor electrode in contact with the electrolyte solution was displayed in Figure S10. For typical p-type catalyst, the Fermi level position was close to the valance band maximum (VBM), which exhibited more positive potential than the redox potential of $\text{NO}_3^-/\text{NH}_3$ pair. On the electrolyte solution side, the oxidized redox species (D_{ox}) occupy the region above E_{redox} , while the reduced redox species (D_{red}) occupied the region below E_{redox} . The distribution levels for the D_{red} and D_{ox} in electrolyte were similar to the VB and CB in semiconductors. As the p-type semiconductor exhibited the characteristic of hole conduction, when the p-type catalyst was immersed in the NO_3^- -containing electrolyte, the potential difference between E_{F} and E_{redox} induced holes (as the majority carriers in p-type semiconductor) transfer from catalyst surface to the solution side. As a result, downward band bending occurred on the semiconductor surface, forming space charge region with hole depletion.”.

➤ Revised Supplementary information (Page 23, highlighted in bright yellow): “For Ag-MoS₂, the Fermi level potential was 1.07 V *versus* RHE, which was much greater than the redox potential of $\text{NO}_3^-/\text{NH}_3$ (E_{redox} , 0.27 V *versus* RHE, pH 7). The p-type MoS₂ exhibited the characteristic of hole conduction. When immersing Ag-MoS₂ into the NO_3^- -contained solution, the potential difference between the E_{F} of Ag-MoS₂ and the E_{redox} induced the interfacial charge transfer between semiconductor surface and electrolyte (that is, holes from Ag-MoS₂ surface to electrolyte, Figure S14a). Thus, downward band bending occurred at the solid-liquid (S-L) junction, and the formed positively charged inner Helmholtz plane (IHP) benefited the enrichment of NO_3^- anions (Figure S14b). In addition to the condition without applied potential, the band bending under the working condition (reduction potential) was also investigated (Figure S14c). As the cathode accepted the electrons from the electrochemical workstation, the E_{F} of Ag-MoS₂ should be closer to the VBM. Therefore, intensified downward band

bending should occur at the S-L junction, causing more positively charged IHP region to favor the NO_3^- enrichment at the reduction potential.”.

➤ Revised Supplementary information (Page 19, highlighted in bright yellow): Figure S10 has been added.

➤ Revised Supplementary information (Page 23, highlighted in bright yellow): Figure S14 has been revised and updated.

➤ Revised Manuscript (Page 33, highlighted in bright yellow): Reference #54 has been added.

c) Series MoS₂ samples would not oxidize nitrate in our reaction system. The oxidation state of nitrogen in nitrate is +5 (the highest theoretical oxidation state of nitrogen), making it difficult to be further oxidized. Only under extreme conditions such as high potential and strong oxidizing agent, nitrate ions may undergo the following two types of reactions: (1) direct oxidation to form peroxyxynitrate and (2) decomposition to form O₂. While, the Fermi level potential of MoS₂ and Ag-MoS₂ is not positive enough to oxidize nitrate. Thus, we consider that the oxidation of nitrate over MoS₂ surface can be excluded.

Comment 2: In line 186, the authors use EIS to measure capacitance, which they then use to quantify charge distribution. However, they fail to clarify whether the capacitance arises from the depletion region, surface adsorption, or electrochemical reactions at -0.6 V vs. RHE. This is a crucial point that needs to be addressed for a proper understanding of their results.

Response: Thanks so much for pointing out this issue. The capacitance of the catalysts which obtained from Nyquist plots at -0.6 V *versus* RHE arises from the surface NO_3^- adsorption and enrichment. The explanations are as follows:

According to the shape of Nyquist plots (which consists two semicircles), we choose the most appropriate physical equivalent circuit diagram in Figure S16. Specifically, R_s is the external circuit resistance, R_{bulk} is the bulk charge transfer resistance, and R_{ct} is the interfacial charge transfer resistance. In the diagram, R_{bulk} and

R_{ct} are paralleled to CPE_{bulk} and CPE_{ct} , respectively. CPE_{bulk} represents the charge accumulation within the bulk catalyst, while CPE_{ct} represents the charge distribution within the solid-liquid interface. In the manuscript, the obtained surface capacitance is CPE_{ct} . The CPE_{ct} of Ag-MoS₂ ($9.5 \mu\text{C cm}^{-2}$) is lower than that of MoS₂ ($32.4 \mu\text{C cm}^{-2}$), indicating significant positive charges distribution of Ag-MoS₂ (Figure 3c).

We have added the definitions of all the physical components in Figure S16, to avoid the ambiguity for understanding the explanations about Nyquist plots.

Figure S16. The fitted equivalent circuit diagram of (a) W-MoS₂ and (b) Ag-MoS₂/MoS₂ catalysts at -0.6 V *versus* RHE.

Corresponding Revisions

- Revised Manuscript (Page 10, highlighted in bright yellow): “(charge distribution within the solid-liquid interface)”.
- Revised Supplementary information (Page 25, highlighted in bright yellow): “In the equivalent circuit diagram, R_s is the external circuit resistance, R_{bulk} is the bulk charge transfer resistance, and R_{ct} is the interfacial charge transfer resistance. In the diagram, R_{bulk} and R_{ct} are paralleled to CPE_{bulk} and CPE_{ct} , respectively. CPE_{bulk} represents the charge accumulation within the bulk catalyst, while CPE_{ct} represents the charge distribution within the solid-liquid interface.”.
- Revised Supplementary information (Page 25, highlighted in bright yellow): Figure S16 has been revised and updated.

Comment 3: *In the rebuttal, the authors claim that the valence band of MoS₂ is primarily composed of S atom orbitals. This is incorrect. The valence band of MoS₂ is mainly formed by the 4d orbitals of Mo, not the orbitals of S atoms (see PHYSICAL*

REVIEW B, 2001, 64, 235305; Nature Communications 5, 4673 (2014)). The holes in the Mo 4d orbitals are responsible for conduction, while unsaturated edge S atoms contribute to surface states that narrow the bandgap at the interface, but they do not play a direct role in bulk conduction.

Response: Thank you for your professional comments. We have carefully studied the literatures you have provided, and also searched other related references (*Adv. Mater.* 2021, 33, 2007509; *Sci. Rep.* 2021, 11, 3958; *J. Mater. Chem. A* 2024, 12, 28170-28176). Combined with the density of states results in the present work, we found that the valance band of MoS₂ is formed by the hybridization of Mo-*d* orbitals and S-*p* orbitals (Figure S59). This property is also available for Ag-MoS₂. These results are similar to the above mentioned literatures.

Figure S59. The partial density of states (PDOS) of (a) Ag-MoS₂, (b) MoS₂ and (c) W-MoS₂.

Following your comments, we realize that the statement about the composition of valance band of MoS₂ in our previous response letter is indeed inappropriate. The interfacial charge transfer is driven by the potential difference between Fermi level of p-type semiconductor and redox pair in electrolyte (Figure S10). Meanwhile, the narrowed space charge layer width in Ag-MoS₂ can induce hole tunneling effect (*Chem. Rev.* 1992, 92, 411-433), which could benefit the hole transfer after the formation of downward band bending. Tunneling effect refers to the quantum behavior in which charges can penetrate or cross the energy barrier, even though the height of the energy barrier is greater than the total energy of the charge. The detailed explanation of interfacial charge transfer and the possible hole tunneling effect can be indexed to the answer of **Comment 5**.

Correspondingly, we have revised the incorrect statement about the valance band composition, and added additional descriptions. In addition, we have cited the related references you have provided (#82: *Nat. Commun.* 2014, 5, 4673; #83: *Phys. Rev. B* 2001, 64, 235305.).

Corresponding Revisions

- Revised Supplementary information (Page 68, highlighted in bright yellow): Figure S59 has been revised and updated. We have provided detailed orbital information of Mo and S atoms.
- Revised Supplementary information (Page 68, highlighted in bright yellow): “According to the PDOS of the MoS₂-based catalysts, the valance band of MoS₂ was formed by the hybridization of Mo-*d* orbitals and S-*p* orbitals. The Ag doping induced generation of band-tail states near the valance band maximum, which lead to the downward shift of Fermi level. In addition, the W doping triggered the upward shift of Fermi level away from the VBM.”.
- Revised Manuscript (Page 34, highlighted in bright yellow): References #82-83 have been cited.

Comment 4: The authors suggest that surface sulfur (S) atoms may act as hole carriers, but they incorrectly describe them as positively charged. Surface S atoms are still negatively charged as anions and are not capable of becoming positively charged. While surface S may act as a hole carrier, it remains negatively charged and cannot be assigned a positive charge. The mechanism based on this erroneous assumption does not hold up.

Response: Thanks so much for your comment. We agree with the reviewer’s opinion, and apologize for the incorrect explanation about the positively charged edge S atoms. The valance band of series MoS₂ catalyst is formed by the hybridization of Mo-*d* orbitals and S-*p* orbitals (Figure S59), while the Mo-*d* orbitals dominate the conduction band. This suggests that the edge S atoms can act as the hole carrier. While, as the reviewer pointed out, it cannot be assigned to a positive charge. We have corrected

corresponding descriptions.

Figure S59. The partial density of states (PDOS) of (a) Ag-MoS₂, (b) MoS₂ and (c) W-MoS₂.

The distribution of positive charges within IHP is based on the interfacial charge transfer (holes from semiconductor surface to electrolyte), which can be due to the potential difference between E_F and E_{redox} (Figure S10). After the formation of downward band bending, the high carrier concentration of Ag-MoS₂ (which is higher than 10^{19} cm^{-3} in Table S2) and the thin space charge layer (5.88 nm, which is below 10 nm, Table S3) collaboratively contributes to the hole tunneling effect for fast hole transfer from semiconductor surface to the IHP region. Specific explanations are provided in **Comment 5**.

Figure S10. The energy level vs. position representation of the semiconductor-electrolyte interface for p-type semiconductor electrode (a) before and (b) after contact with the electrolyte solution.

Corresponding Revisions

- Revised Supplementary information (Page 68, highlighted in bright yellow): Figure S59 has been revised and updated. We have provided detailed orbital information of Mo and S atoms.
- Revised Supplementary information (Page 19, highlighted in bright yellow): Figure S10 has been added.
- Revised Supplementary information (Page 9, highlighted in bright yellow): “For typical p-type catalyst, the Fermi level position of Ag-MoS₂ is close to the valance band maximum (VBM), which exhibits more positive potential than the redox potential of NO₃⁻/NH₃ pair. As the p-type Ag-MoS₂ exhibits the characteristic of hole conduction, when the Ag-MoS₂ is immersed in the NO₃⁻-containing electrolyte, the potential difference between E_F and E_{redox} induces holes (as the majority carriers in p-type semiconductor) transfer from catalyst surface to the IHP of solution side. As a result, downward band bending occurs on the semiconductor surface, forming space charge layer with hole depletion. And the positively charged IHP could attract nitrate anions under electrostatic interaction.”.
- Revised Manuscript (Page 10, highlighted in bright yellow): “P-type MoS₂ exhibited the characteristic of hole conduction, and the 2D structure endowed abundant edge S sites as traps to capture positive holes⁶⁰⁻⁶². Thus, the positive charges distributed within IHP can be ascribed to the surface trapped holes.” has been deleted.

Comment 5: *The statement that "the downward band bending promotes hole transfer from bulk to surface" is fundamentally incorrect. Downward band bending would push holes toward the bulk, not to the surface. This misinterpretation of band bending further undermines the manuscript's core argument.*

Response: Thanks for your professional comments. We acknowledge that the description about “the downward band bending promotes hole transfer from bulk to surface” in the previous response letter is inappropriate. As the reviewer pointed out, the downward band bending would push the holes toward bulk region. While, for the semiconductors with high carrier concentration (above 10^{18} cm⁻³) and thin space charge

layer (below 10 nm), there will be changes for the migration behavior of holes. Therefore, we have conducted thorough research and analysis on the mechanisms of band bending and holes transfer, to provide reasonable explanations for this work. Specific explanations are as follows:

a) We agree with the reviewer's opinion that the downward band bending should generate energy barrier, to hinder the transfer of holes from semiconductor surface to electrolyte solution. While, when the carrier concentration is higher than 10^{18} cm^{-3} , the thickness of the space charge layer (SCL) can be lower than 10 nm. As shown in Figure S10b, the thin SCL would induce the tunneling process for interfacial charge transfer (*Chem. Rev.* 1992, 92, 411-433).

Figure S10. The energy level vs. position representation of the semiconductor-electrolyte interface for p-type semiconductor electrode (a) before and (b) after contact with the electrolyte solution.

b) We have then calculated the carrier concentrations of Ag-MoS₂ and MoS₂ using M-S measurements. As seen from Figure 3a, the slope of Ag-MoS₂ is much lower than those of MoS₂ and W-MoS₂. The calculated carrier concentration of MoS₂ is $6.3 \times 10^{18} \text{ cm}^{-3}$, and the Ag doping significantly increased the carrier concentration to $1.14 \times 10^{19} \text{ cm}^{-3}$ (which is 1.8 times to that of MoS₂, Table S2). The high carrier concentration of Ag-MoS₂ suggests that the charge tunneling phenomenon may occur within the interface of Ag-MoS₂ and electrolyte solution.

Figure 3a. Mott-Schottky plots of Ag-MoS₂, MoS₂, and W-MoS₂ catalysts.

Table S2. Calculated carrier concentration of the three samples by M-S plots.

Samples	Carrier concentration (cm ⁻³)
W-MoS ₂	3.25×10 ¹⁸
MoS ₂	6.3×10 ¹⁸
Ag-MoS ₂	1.14×10 ¹⁹

c) To investigate whether the charge tunneling effect would occur, we have further calculated the SCL width (W_{sc}) of MoS₂ and Ag-MoS₂. In the case of a p-type semiconductor, the holes are depleted in the SCL of MoS₂ and Ag-MoS₂, and the width is calculated by the equation as follows:

$$W_{sc} = \sqrt{\frac{2\epsilon_r\epsilon_0(U-U_{FB})}{eN_d}}$$

Where ϵ_0 is the vacuum permittivity, ϵ_r is the relative dielectric constant of MoS₂, U is the applied external potential, U_{FB} is the flat band potential, and N_d is the donor density (*Chem. Eng. J.* 2021, 404, 126458; *Appl. Catal., B* 2022, 315, 121606; *Angew. Chem. Int. Ed.* 2023, 62, e202217026.).

As a result, the W_{sc} of MoS₂ and Ag-MoS₂ are 10.28 and 5.88 nm, respectively (Table S3). The Ag doping increases the carrier concentration, and the W_{sc} reduces

significantly which is less than 10 nm. Under this condition, the holes have the possibility to tunnel from the semiconductor surface into the solution. Under working condition with applied negative bias, the intensified band bending would further reduce the W_{sc} , to benefit the hole tunneling effect for fast semiconductor-electrolyte interfacial charge transfer. This would help to facilitate the continuous hole tunneling from semiconductor surface to the IHP region.

We realize that the previous description about the hole transfer mechanism is incorrect. We have corrected and revised corresponding descriptions.

Table S3. Calculated space charge layer width of the as-prepared samples.

Samples	Space charge layer width (W_{sc})
MoS ₂	10.28 nm
Ag-MoS ₂	5.88 nm

d) According to the capacitance results of ACV and Nyquist measurements, phase angle and frequency variation of Bode plots, and the quantification of surface NO₃⁻ concentration (IC, SEM, and *in situ* Raman tests) on the catalysts, we find holes could transfer from Ag-MoS₂ surface to IHP, even after the formation of downward band bending. The more positively charged IHP could attract nitrate anion under electrostatic interaction. These experimental results can provide evidences for the hole tunneling effect on Ag-MoS₂ surface.

Corresponding Revisions

➤ Revised Manuscript (Pages 9-10, highlighted in bright yellow): “Specifically, p-type MoS₂ catalysts exhibited the characteristic of hole conduction. Under the drive of $E_F - E_{redox}$, the holes (majority carrier) in valance band would transfer from semiconductor surface to electrolyte, inducing charge rearrangement within IHP region (Figure S10a)⁵⁴. Meanwhile, the high carrier concentration ($1.14 \times 10^{19} \text{ cm}^{-3}$) and thin space charge layer width (5.88 nm) of Ag-MoS₂ can contribute to the increase of

probability for hole tunneling effect (Figure S10b and Table S2-3), which benefited positive charges distribution within IHP.”.

➤ Revised Supplementary information (Page 77, highlighted in bright yellow): “To further investigate whether the charge tunneling effect would occur, we have then calculated the SCL width of MoS₂ and Ag-MoS₂. In the case of a p-type semiconductor, the holes were depleted in the SCL of MoS₂ and Ag-MoS₂, and the width is calculated by the equation as follows:

$$W_{sc} = \sqrt{\frac{2\epsilon_r\epsilon_0(U-U_{FB})}{eN_d}}$$

Where ϵ_0 is the vacuum permittivity, ϵ_r is the relative dielectric constant of MoS₂, U is the applied external potential, U_{FB} is the flat band potential, and N_d is the donor density.

As a result, the W_{sc} of MoS₂ and Ag-MoS₂ were 10.28 and 5.88 nm, respectively. The Ag doping increased the carrier concentration, and the W_{sc} reduced significantly which is less than 10 nm. Under this condition, the holes had the possibility to tunnel from the semiconductor surface into the solution. Under working condition with applied negative bias, the intensified band bending would further reduce the W_{sc} , to benefit the hole tunneling effect for fast semiconductor-electrolyte interfacial charge transfer.”.

➤ Revised Supplementary information (Page 77, highlighted in bright yellow): Table S3 has been added.

Comment 6: The authors mention a carrier concentration of 10^{18} cm^{-3} , which typically places the material in a range from semiconductor to insulator. They then claim superior conductivity based on EIS measurements, but these results are largely influenced by carbon additives and do not address the underlying issue of poor electric conductivity. Additionally, this comparison is unfair, as the voltage is primarily applied to the depletion region, and EIS does not account for overpotential losses, which are critical in this context.

Response: Thank you for your comments, and the questions raised by the reviewer are answered from the following three parts:

a) The high carrier concentrations of MoS₂ and Ag-MoS₂. In the present work, the carrier concentrations of the samples are calculated by the Mott-Schottky measurement (M-S). As seen from Figure 3a and Table S2, the calculated carrier concentration of MoS₂ and Ag-MoS₂ are 6.3×10^{18} and 1.14×10^{19} cm⁻³, respectively. Under room temperature condition, the carrier concentration of typical semiconductor is within $10^{11} \sim 10^{17}$ cm⁻³ (Photoelectrochemical water splitting, New York: Springer 2013, 344; Nanoscale, 2013, 5, 2938; J. Am. Chem. Soc., 2018, 140, 29, 9078–9082). The carrier concentrations of MoS₂ and Ag-MoS₂ are higher than most semiconductors. This can be ascribed to the relatively narrow band gap (~1.5 eV). Meanwhile, the 2D structure can promote the carrier transfer within the layer. On the contrary, the carrier concentration of insulator is usually below 10^{10} cm⁻³. Thus, we consider that the MoS₂ and Ag-MoS₂ samples exhibit high carrier concentration and conductivity.

b) Excluding the influence of carbon substrate. To eliminate the influence of carbon substrate on the conductivity, we anchor the MoS₂ samples on soda-lime glass through in situ hydrothermal process. The test is carried out under open circuit condition. The Nyquist plots are fitted using the equivalent circuit diagram in Figure S16. As seen from Figure R1 and Table R1, the charge transfer resistance within bulk Ag-MoS₂ and MoS₂ are 28.7 and 46.2 Ω cm⁻², suggesting excellent conductivity of MoS₂-based catalysts.

In addition, we have performed EIS tests of series typical p-type semiconductors, including CuO, Cu₂O, and NiO. Clearly, the semicircles in both low- and high-frequency regions are larger than those of Ag-MoS₂ and MoS₂. The charge transfer resistances within the bulk regions of CuO, Cu₂O, and NiO are 55.3, 92.5, and 185.9 Ω cm⁻², respectively. These results reveal that MoS₂-based catalysts exhibit relatively high conductivity among typical p-type semiconductors.

Figure R1. Nyquist plots of (a) Ag-MoS₂, (b) MoS₂, (c) CuO, (d) Cu₂O, (e) NiO catalysts which anchored on soda-lime glass under open circuit condition.

Table R1. Fitting results for resistances of various catalysts under open circuit condition in a 0.5 M K₂SO₄ electrolyte.

Sample tests	Resistance $R_{\text{bulk}} (\Omega \text{ cm}^{-2})$
Ag-MoS ₂	28.7
MoS ₂	46.2
Cu ₂ O	55.3
CuO	92.5
NiO	185.9

c) The main idea of this work is to construct solid-liquid junction to promote reactant enrichment. Although the conductivity plays important role in electrocatalysis, the main idea of this work is to address the problem of cathodic repulsion of nitrate anion under low concentration condition. As discussed above, p-type MoS₂ exhibits a more positive Fermi level potential than that of E_{redox} of NO₃⁻/NH₃, inducing the charge transfer within semiconductor-electrolyte interface. The formed positive IHP benefits the adsorption of NO₃⁻ cation under electrostatic

interaction, thus lead to the enrichment of NO_3^- within IHP region under low concentration condition. The Ag doping can further regulate the Fermi level potential of MoS_2 , strengthening the enrichment of NO_3^- .

Comment 7: Finally, the authors should explicitly state the dielectric constant used in the calculation of carrier concentration in the experimental section. This is a crucial parameter that needs to be clearly defined.

Response: Thank you for your comment, and we have supplemented the dielectric constant for carrier concentration calculation. Based on the SEM and TEM results, the as-prepared MoS_2 nanoflowers are bulk catalysts. According to the literatures, the dielectric constant of bulk MoS_2 is 7.6 (*Sci. Rep.* 2015, 5, 16996; *Proc. R. Soc. Lond. A* 284, 402-422), and we have supplemented this parameter in the experimental section part.

Corresponding Revisions

➤ Revised Manuscript (Pages 23-24, highlighted in bright yellow): “Mott-Schottky measurements were carried out at the frequency of 1000 Hz under dark condition to obtain the flat band potential and carrier concentration of the samples, in which the dielectric constant of MoS_2 was 7.6^{75,84}.”.

We greatly appreciate the constructive comments and feedback from the reviewer on this article. We have added experiments, revised corresponding descriptions and discussions within the article, and solving these questions is helpful to further improve the depth and scientific nature of the paper. Thank you again for your help and support of our work.

Reviewer #3:

The authors have resolved the major concerns raised, and I recommend the manuscript for publication.

Response: Thanks so much for your constructive comments, your comments play a crucial role in improving the quality of our paper. Thank you again for your help and support of our work.

Reviewer #4:

This article presents a novel Ag-doped MoS₂ catalyst (Ag-MoS₂) for the efficient conversion of nitrate (NO₃⁻) to ammonia (NH₃). The authors suggest that doping with Ag enhances the p-type characteristics of MoS₂, leveraging its hole conduction properties to generate more positive charges at the solid-liquid interface, which in turn attracts nitrate anions. This increased enrichment of reactants at the solid-liquid interface improves reaction efficiency, achieving a nitrate-to-ammonia conversion efficiency close to 100%. Three reviewers provided valuable feedback, offering insightful analyses and highly professional suggestions regarding the selection of experimental conditions, the reliability of results, the clarity of chart representations, and the experimental design. The authors addressed the reviewers' comments in detail and revised the article accordingly. After these revisions, the article is data-rich, logically coherent, and I recommend its publication. However, there are still a few minor points that need attention:

Response: Thanks so much for your recognition and providing professional comments on the present work. Following your suggestions, we have provided corresponding experiments and supplemented detailed explanations. We hope our effort can address your concerns. The detailed descriptions and explanations are illustrated in the response for each comment you have raised.

***Comment 1:** On page 11, the authors describe how they quantified the NO₃⁻ enrichment capacity based on methods outlined in the Supplementary information (SI). They measured the overall concentration of adsorbed NO₃⁻ directly. However, aside from the enhanced adsorption capacity due to the p-type properties of the catalyst improved by Ag doping, other factors such as the electrochemical active surface area (ECSA) and specific surface area of the catalyst may also influence the nitrate adsorption capacity. I believe it is insufficient for the authors to rely solely on ion chromatography to detect adsorbed nitrate. It would be beneficial to also test and compare the specific surface area and ECSA of the different catalysts coated on carbon paper.*

Response: Thanks you for your constructive comments. We agree with the reviewer's opinion that the improvement in NO_3^- adsorption capacity may not only be related to the enhanced p-type properties of the catalyst through Ag doping. This could also be associated with factors such as the electrochemical active surface area (ECSA) and specific surface area of the catalyst. To further clarify the primary factors which contributed to the increased NO_3^- adsorption capacity, we have supplemented the tests of ECSA and specific surface area for the three catalysts: Ag-MoS₂, MoS₂, and W-MoS₂. Also, we normalize the NO_3^- adsorption capacity of these catalysts with respect to their ECSA and specific surface area for comparative analysis.

As shown in Figure S27, the electrochemical active surface area (ECSA) of the catalysts is determined using the electrochemical double-layer capacitance (C_{dl}) method. The ECSA values follow the order: Ag-MoS₂ (45.0 cm²) < MoS₂ (67.5 cm²) < W-MoS₂ (80.0 cm²). By normalizing the NO_3^- adsorption capacity with respect to ECSA (Figure 3g), Ag-MoS₂ exhibits the highest NO_3^- adsorption capability at -0.6 V vs. RHE, surpassing MoS₂ and W-MoS₂ by factors of 8.8 and 27.6, respectively, at their optimal adsorption potentials. Furthermore, nitrogen adsorption/desorption isotherms (Figure S28a) reveal that the specific surface area (SSA) of the catalysts also follow the order: Ag-MoS₂ (8.3 m²/g) < MoS₂ (8.9 m²/g) < W-MoS₂ (9.0 m²/g). When the NO_3^- adsorption capacity is normalized by SSA (Figure S28b), Ag-MoS₂ demonstrates maximum NO_3^- adsorption capacities 5.8 times and 16.0 times higher than those of MoS₂ and W-MoS₂, respectively. This trend aligns with the ECSA normalization results, highlighting Ag-MoS₂'s superior intrinsic NO_3^- adsorption capability. These findings confirm that Ag doping enhances the p-type properties of the catalyst, enabling effective NO_3^- enrichment on the catalyst surface.

Figure S27. Cyclic voltammetry curves of (a) Ag-MoS₂, (b) MoS₂, (c) W-MoS₂ catalysts at different scan rates, and their (d, e, f) current density differences at 0.66 V versus RHE against scan rates to calculate C_{dl} .

Figure 3g. The ECSA normalized NO₃⁻ adsorption capacity at different applied potentials on catalysts.

Figure S28. (a) Specific surface area of Ag-MoS₂, MoS₂, and W-MoS₂ catalysts tested by nitrogen adsorption/desorption isotherms. (b) The specific surface area (SSA) normalized NO₃⁻ adsorption capacity at different applied potentials on catalysts.

Corresponding revision:

- Revised Manuscript (Page 13, highlighted in bright yellow): Figure 3g has been revised and updated.
- Revised supplementary information (Page 37, highlighted in bright yellow): Figure S28 has been added.
- Revised supplementary information (Page 37, highlighted in bright yellow): “Nitrogen adsorption/desorption isotherms (Figure S28a) revealed that the specific surface area (SSA) of the catalysts also followed the order: Ag-MoS₂ (8.3 m²/g) < MoS₂ (8.8 m²/g) < W-MoS₂ (9.0 m²/g). When the NO₃⁻ adsorption capacity was normalized by SSA (Figure S28b), Ag-MoS₂ demonstrated maximum NO₃⁻ adsorption capacities 5.8 times and 16.0 times higher than those of MoS₂ and W-MoS₂, respectively. This trend aligned with the ECSA normalization results, highlighting Ag-MoS₂'s superior intrinsic NO₃⁻ adsorption capability.”.
- Revised Manuscript (Page 12, highlighted in bright yellow): “By normalizing the NO₃⁻ adsorption capacity with respect to the electrochemical active surface area (ECSA), the concentration of NO₃⁻ adsorbed on Ag-MoS₂ presented a normal distribution trend as the potential ranging from -0.2 to -1.0 V *versus* RHE (Figures 3g and S27), and reached the peak at -0.6 V *versus* RHE (2.0 μg mL⁻¹ cm⁻²_{ECSA}). Due to the different Fermi level position, the adsorbed NO₃⁻ concentration of MoS₂ and W-MoS₂ peaked at more negative potentials of -0.8 and -1.0 V *versus* RHE, respectively. As the Ag dopant acted as the electron acceptor to increase the carrier concentration (Table S2), it induced an intensified surface band bending when the Ag-MoS₂ was contacted with the NO₃⁻ contained electrolyte. Thus, the maximum adsorbed NO₃⁻ concentration of Ag-MoS₂ surpassed MoS₂ and W-MoS₂ by factors of 8.8 and 27.6, respectively, at their optimal adsorption potentials. The NO₃⁻ adsorption concentration was also normalized by specific surface area (SSA), this trend aligned with the ECSA normalization results (Figure S28), highlighting superior intrinsic NO₃⁻ adsorption capability of Ag-MoS₂.”.

Comment 2: The authors could strengthen their work by comparing their results with existing literature, especially with other high-performance NO₃RR catalysts. Including intuitive comparison charts would help highlight the innovations and advantages of this study.

Response: We appreciate the reviewer’s attention to this aspect. We would like to clarify that the performance comparison has been included in Figure 4c and Table S6. Specifically, a comprehensive comparison between our work and recent studies has been explicitly summarized in Table S6 (≤ 20 mM NO₃⁻ conditions). As further demonstrated in Figure 4c, we compare our NO₃RR performance with representative state-of-the-art NH₃ activity ever reported (#22: *Nat. Water* 1, 1068-1078 (2023); #32: *Adv. Mater.* 34, e2204306 (2022); #34: *Nat. Energy* 5, 605-613 (2020); #71: *Angew. Chem. Int. Ed.* 61, e202202604 (2022); #72: *ACS Catal.* 11, 7568-7577 (2021).). The results show that our catalyst exhibits a significant advantage in low-concentration NO₃⁻ electrolytes.

Figure 4c. NO₃RR performance comparison of reported electrocatalysts (NO₃⁻ concentration ≤ 10 mM).

Table S6. Comparison of NO₃RR performance of Ag-MoS₂ with reported works at low nitrate concentration system (≤ 20 mM).

Catalyst	Electrolyte	NH ₃ FE (%)	NH ₃ production rate (mg h ⁻¹ cm ⁻²)	Current density (mA cm ⁻²)	Ref.
Ag-MoS ₂	10 mM KNO ₃ + 0.5 M K ₂ SO ₄	~100	~20	~200	This work
FeNi ₅₀₀ /FF	14.3 mM KNO ₃ +	65.2	~0.27	~10	Nat. Water 2023, 1,

	0.05 M Na ₂ SO ₄				1068-1078
Co-Fe@Fe ₂ O ₃	500 ppm NO ₃ ⁻ -N + 0.1 M Na ₂ SO ₄	85.2	0.885	~17	PNAS 2022, 119, e2115504119
Cu-PTCDA	8.1 mM KNO ₃ + 1 M PBS	85.9	0.44	15	Nat. Energy 2020, 5, 605-613
Pd	20 mM NaNO ₃ + 0.1 M NaOH	35	0.34	4.25	ACS Catal. 2021, 11, 12, 7568-7577
Cu@C	1 mM KNO ₃ + 1 M KOH	72	0.47	5.5	Adv. Mater. 2022, 34, 2204306
CuCl ₂ _BEF	7.1 mM KNO ₃ + 0.5 M Na ₂ SO ₄	98.6	1.82	62	Angew. Chem. Int. Ed. 2021, 60,22933- 22939
Cu/Cu ₂ O nanowires	3.2 mM NaNO ₃ + 0.5 M Na ₂ SO ₄	95.8	4.1	120	Angew. Chem. Int. Ed. 2020, 59, 5350- 5354
a-RuO ₂	3.2 mM NaNO ₃ + 0.5 M Na ₂ SO ₄	97.46	2.0	60	Angew. Chem. Int. Ed. 2022, 134, e202202604
O-SiNW/Au	10 mM HNO ₃ + 0.5 M K ₂ SO ₄	95.6	4.4	–	Angew. Chem. Int. Ed. 2022, 61, e202204117

Comment 3: Regarding Comment 2 from Reviewer #2, the authors proposed that nitrate is transformed, and the resulting ammonium is recovered as a potentially valuable product. If the production of valuable ammonium is a primary application of this research, it would be helpful for the authors to discuss the economic value of ammonium, the cost of the catalyst, the scalability of the preparation method, and potential challenges in large-scale applications. Additionally, the authors noted that one of the key applications of this work is in treating agricultural and industrial wastewater. In light of this, it is recommended that the authors simulate conditions more representative of actual wastewater treatment or ammonia production, such as using more complex electrolyte compositions.

Response: Thank you for your professional comments, and the questions raised by the reviewer are answered from the following three parts:

a) The economic value of ammonia and the cost of the catalyst. Following your suggestions, we have evaluated the economic value of ammonia and calculated the catalyst cost. The present price of industrial grade ammonia and the product NH_4Cl are around 2700~3000 ¥/ton and 1000~1200 ¥/ton, respectively. According to the fabrication process of Ag-MoS₂ catalyst (in experimental section), the preparation cost of Ag-MoS₂ with 1 cm² is as low as ca. 0.475 ¥. Detailed information is shown in Table R2.

Thus, we consider the cost of the Ag-MoS₂ catalyst is relatively low. Also, the ammonia and NH_4Cl are both the products with high economic value.

Table S8. Preparation cost of Ag-MoS₂ on carbon cloth with 1 cm².

Reagents	Usage	Price (¥)
$\text{Na}_2\text{MoO}_4 \cdot 2\text{H}_2\text{O}$	0.27 mmol	0.13
$\text{CH}_4\text{N}_2\text{S}$	1.15 mmol	0.008
AgNO_3	0.013 mmol	0.027
$\text{NH}_2\text{OH} \cdot \text{HCl}$	0.65 mmol	1×10^{-5}
Hexadecyl trimethyl ammonium bromide	0.011 g	0.01
Carbon cloth	1 cm ²	0.3
Total		0.475

Corresponding revision:

- Revised Supplementary information (Page 82, highlighted in bright yellow): “The present price of industrial grade ammonia and the product NH_4Cl were around 2700~3000 ¥/ton and 1000~1200 ¥/ton, respectively. According to the fabrication process of Ag-MoS₂ catalyst (in experimental section), the preparation cost of Ag-MoS₂ with 1 cm² was as low as ca. 0.475 ¥.”.
- Revised Supplementary information (Page 82, highlighted in bright yellow): Table S8 has been added.

➤ Revised Manuscript (Page 18, highlighted in bright yellow): “Also, we have evaluated the preparation cost of Ag-MoS₂ catalyst, whose price with 1 cm² was as low as 0.475 ¥ (Table S8).”.

b) The scalability of the preparation method, and potential challenges in large-scale applications. Series MoS₂ catalysts are prepared through a facile hydrothermal process (*Mater. Lett.* 2012, 86, 9-12). Specifically, Na₂MoO₄ and CH₄N₂S react with each other to form Na₂MoS₂ in the first stage. Then, Na₂MoS₂ could be readily reduced to MoS₂ nanoparticles with the help of NH₂OH·HCl. Subsequently, the nanoparticles start to assemble together and spontaneously transform to MoS₂ nanosheets. Finally, MoS₂ nanoflowers are formed from several MoS₂ nanosheets with the help of CTAB surfactant. In the present work, carbon cloth with an area of 4 × 4 cm² is immersed into the precursor with the addition of AgNO₃ as Ag source for the preparation of Ag-MoS₂ catalyst. According to the SEM images (Figure 2a and S2), the Ag-MoS₂ nanoflowers disperse evenly on the carbon cloth substrate. Since the hydrothermal reaction is facile and controllable, we consider that the preparation of Ag-MoS₂ with larger scale is feasible.

In addition, potential challenges in large-scale applications are as follows: 1) Within the large-scale preparation process of Ag-MoS₂, the area of carbon cloth should match the shape and size of Teflon-lined stainless-steel autoclave, since the curling of carbon cloth may lead to the aggregation and uneven distribution of Ag-MoS₂ nanoflower on the carbon cloth substrate. 2) Meanwhile, for Teflon-lined stainless-steel autoclave with larger capacity, the even temperature distribution of the heating device (usually air blast drying oven) is important. 3) Furthermore, for industrial scaling up experiments, the high current (above 10 A) is usually utilized. Under this condition, Ag-MoS₂ catalyst may dissolve in to the electrolyte. These are the challenges that we should overcome for future practical large-scale applications.

Corresponding revision:

➤ Revised Supplementary information (Page 10, highlighted in bright yellow): “Series MoS₂ catalysts were prepared through a facile hydrothermal process.

Specifically, Na_2MoO_4 and $\text{CH}_4\text{N}_2\text{S}$ reacted with each other to form Na_2MoS_2 in the first stage. Then, Na_2MoS_2 could be readily reduced to MoS_2 nanoparticles with the help of $\text{NH}_2\text{OH}\cdot\text{HCl}$. Subsequently, the nanoparticles started to assemble together and spontaneously transform to MoS_2 nanosheets. Finally, MoS_2 nanoflowers were formed from several MoS_2 nanosheets with the help of CTAB surfactant. In the present work, carbon cloth with an area of $4 \times 4 \text{ cm}^2$ was immersed into the precursor with the addition of AgNO_3 as Ag source for the preparation of Ag- MoS_2 catalyst. According to the SEM images (Figure 2a and S2), the Ag- MoS_2 nanoflowers dispersed evenly on the carbon cloth substrate. Since the hydrothermal reaction is facile and controllable, we consider that the preparation of Ag- MoS_2 with larger scale is feasible.”.

c) NO_3RR performance in simulated wastewater. Following your insightful suggestions, we have performed the NO_3RR over Ag- MoS_2 in simulated nitrate-containing wastewater. Usually, the real wastewater consists of NO_3^- , SO_4^{2-} , Na^+ , K^+ , CO_3^{2-} , Cl^- , etc. (*Nat. Sustain.* 2024, 7, 1251-1263.). Thus, we have prepared the mixed electrolyte containing NO_3^- , SO_4^{2-} , Na^+ , K^+ , CO_3^{2-} , and Cl^- to simulate the wastewater environment. The optimal NH_3 Faradaic efficiency over Ag- MoS_2 in simulated wastewater is 76.7% at -0.6 V *versus* RHE, while the NH_3 yield rate reaches $16.02 \text{ mg h}^{-1} \text{ cm}^{-2}$ at -1.0 V *versus* RHE (Figure S54a). The decreased NO_3RR performance can be ascribed to the presence of interfering ions that influence the targeted adsorption of NO_3^- . Although the NO_3RR performance in wastewater decreased to a certain extent, the NH_3 yield rate and Faradaic efficiency still maintained 80% and 76% of the optimal performance in simple electrolyte ($\text{K}_2\text{SO}_4+\text{KNO}_3$).

We have also utilized the air-stripping method for the recovery of high-purity ammonia products from simulated nitrate-containing wastewater. Approximately 77.8% of the NH_3 vapor is successfully stripped out from the electrolyte (Figure S54b). Subsequently, around 75.1% of the outflowing NH_3 gas is collected in an HCl solution, and approximately 73.9% of NH_4Cl powder is finally obtained after rotary evaporation. These results suggest that the Ag- MoS_2 has the potential for rapid conversion of NO_3^- to ammonia product in actual industrial and agricultural wastewater.

Figure S54. (a) NH₃ yield and Faradaic efficiency of Ag-MoS₂ catalyst under different applied potentials in simulated nitrate-containing wastewater. (b) The conversion efficiency of different steps for the ammonia product synthesis process.

Corresponding revision:

➤ Revised Manuscript (Pages 17-18, highlighted in bright yellow): “Next, to investigate the practical application prospect, complex electrolyte (which included CO₃²⁻, Na⁺, Cl⁻, K⁺, NO₃⁻, and SO₄²⁻) was prepared to simulate the nitrate-containing wastewater condition⁷⁵. The NH₃ Faradaic efficiency over Ag-MoS₂ in simulated wastewater was 76.7%, while the NH₃ yield rate reached 16.02 mg h⁻¹ cm⁻² (Figure S54a). The decreased NO₃RR performance can be ascribed to the presence of interfering ions that influenced the targeted adsorption of NO₃⁻. Although the NO₃RR performance in wastewater decreased to a certain extent, the NH₃ yield rate and Faradaic maintained 80% and 76% of the optimal performance for simple electrolyte. In the recovery process, approximately 73.9% of NH₄Cl powder was finally obtained (Figure S54b). Also, we have evaluated the preparation cost of Ag-MoS₂ catalyst, whose price with 1 cm² was as low as 0.475 ¥ (Table S8). These results suggested that the Ag-MoS₂ has the potential for the conversion of NO₃⁻ to ammonia product in actual industrial and agricultural wastewater.”.

➤ Revised Supplementary information (Page 63, highlighted in bright yellow): Figure S54 has been added.

➤ Revised Supplementary information (Page 63, highlighted in bright yellow): “We have also utilized the air-stripping method for the recovery of high-purity ammonia products from simulated nitrate-containing wastewater. Clearly, approximately 77.8%

of the NH_3 vapor was successfully stripped out from the electrolyte (Figure S54b). Subsequently, around 75.1% of the outflowing NH_3 gas was collected in an HCl solution, and approximately 73.9% of NH_4Cl powder was finally obtained after rotary evaporation. Although the conversion efficiency decreased in complex wastewater environment, this approach still holds the potential for practical application.”.

Comment 4: Regarding Comment 3 from Reviewer #2, the reviewer pointed out that K_2SO_4 , which does not facilitate ion exchange, cannot effectively control the reaction conditions. The authors confirmed through verification experiments that the pH changes significantly before and after the reaction when K_2SO_4 is used as the electrolyte. In response, the authors suggested that the ionic effect and the good conductivity of SO_4^{2-} contribute to better performance, whereas PBS has poorer conductivity. However, could it be possible to prepare an electrolyte composed of both K_2SO_4 and PBS? By using PBS to control the pH during the reaction process, K_2SO_4 could potentially enhance the solution's conductivity.

Response: We sincerely appreciate the reviewer's careful evaluation and valuable suggestions. In response to the insightful comments on the electrolyte system, we provide the following clarification and discussion:

a) Reason for selecting K_2SO_4 as the supporting electrolyte. The choice of K_2SO_4 as the supporting electrolyte is motivated by the ionic composition of real nitrate-contaminated water bodies (e.g., agricultural runoff and industrial wastewater), which are typically neutral or weakly alkaline and contain coexisting ions such as NO_3^- , SO_4^{2-} , Na^+ , K^+ , CO_3^{2-} , Cl^- , etc. Notably, sulfate (SO_4^{2-}) is one of the main components in nitrate wastewater (*Water Res.* 2009, 43, 17, 4430-4440; *Water Res.* 1998, 32, 10, 3080-3084.). K_2SO_4 is chosen not only for its high conductivity but also to simulate the ionic composition of real wastewater, enabling an assessment of catalyst performance under conditions closely resembling practical scenarios. In contrast, phosphate-buffered saline (PBS), despite its ability to stabilize pH (~ 7.0), is less relevant due to the near absence of its primary components (H_2PO_4^- , HPO_4^{2-}) in actual wastewater, which lack inherent buffering capacity in such systems.

b) NO₃RR performance in PBS + K₂SO₄ hybrid electrolyte. Following the reviewer's suggestion, we experimentally investigated the NO₃RR performance in a hybrid electrolyte composed of PBS and K₂SO₄ (Figure S46). As shown in Figure S46b and S46d, the optimal catalytic NH₃ yield rate and FE (16.0 mg h⁻¹ cm⁻², 92.4%) in the hybrid system is intermediate between those observed in pure PBS (13.4 mg h⁻¹ cm⁻², 83.4%) and pure K₂SO₄ (20.1 mg h⁻¹ cm⁻², 99.7%). Post-reaction pH analysis (Figure S46c) reveals that the hybrid electrolyte exhibits a pH increase from 7.10 to 8.62, a change more pronounced than in pure PBS (7.02→7.34) but less drastic than in pure K₂SO₄ (6.71→11.87). The elevated alkalinity can help to suppress the competitive hydrogen evolution reaction, enhancing both activity and selectivity for NH₃ synthesis via NO₃RR, which explains the superior performance in pure K₂SO₄ systems.

Figure S46. NH₃ FE and NH₃ yield rate of the three samples in (a) PBS electrolyte and (b) PBS + K₂SO₄ hybrid electrolyte. (c) The pH values before and after the NO₃RR process in different electrolytes. (d) NH₃ FE and NH₃ yield rate of Ag-MoS₂ in the three electrolytes.

Corresponding revision:

➤ Revised supplementary information (Page 55, highlighted in bright yellow): “As seen from Figure S46b and S46d, the NH₃ yield rate and FE (16.0 mg h⁻¹ cm⁻², 92.4%)

in the hybrid PBS+K₂SO₄ system was intermediate between those observed in pure PBS (13.4 mg h⁻¹ cm⁻², 83.4%) and pure K₂SO₄ (20.1 mg h⁻¹ cm⁻², 99.7%).”.

➤ Revised supplementary information (Page 55, highlighted in bright yellow): “In addition, the hybrid electrolyte exhibited a pH increase from 7.10 to 8.62, a change more pronounced than in pure PBS (7.02→7.34) but less drastic than in pure K₂SO₄ (6.71→11.87).”.

➤ Revised supplementary information (Page 55, highlighted in bright yellow): Figure S46 has been revised and updated.

c) Current research focus and future optimization directions. While alkaline conditions significantly enhance NO₃RR performance, long-term catalyst stability remains a critical challenge. Future studies will prioritize real-time monitoring and precise control of solution pH to maintain a moderately alkaline environment, thereby improving catalyst durability. The current work focuses on band structure engineering to redistribute interfacial charge at the inner Helmholtz plane (IHP), addressing the electrostatic repulsion of low-concentration nitrate ions at the cathode surface. The important role of electrolyte composition in electrocatalytic reactions is underscored by experimental data provided in the supplementary information (Figure S46). Future investigations will systematically explore the effects of electrolyte engineering (e.g., pH change, ionic strength, coexisting anions) on reaction kinetics and selectivity, aiming to refine catalyst design for practical applications. We deeply appreciate the reviewer’s insightful critique, which has guided us toward critical future investigations.

Comment 5: Regarding Comment 1 from Reviewer #3, although the authors have supplemented their analysis with data on the different adsorption capacities of the three samples for NO₃⁻ at various potentials, the core issue raised by Reviewer #3—the comparison of energy barriers for different catalysts at their respective optimal potentials—has not yet been fully addressed. The authors should provide a more detailed discussion of this issue, for example, by calculating the changes in reaction free energy at different potentials to clearly demonstrate the advantages of Ag-MoS₂.

Response: Thank you for your kind comments. In response to the reviewer's suggestion, we performed density functional theory (DFT) calculations to compare the Gibbs free energy change of elementary steps in the NO₃RR over Ag-MoS₂, MoS₂, and W-MoS₂ catalysts at their respective optimal NO₃⁻ adsorption potentials. The key findings are summarized as follows:

As shown in Figure R2, the adsorption energy of NO₃⁻ on Ag-MoS₂ (-2.42 eV) is stronger than those on MoS₂ (-2.25 eV) and W-MoS₂ (-2.28 eV), indicating that Ag doping favors NO₃⁻ adsorption. At their respective optimal potentials (Ag-MoS₂: -0.6 V vs. RHE, MoS₂: -0.8 V vs. RHE, W-MoS₂: -1.0 V vs. RHE, Figures S61 and S64-65), the Gibbs free energy change of the potential-determining step (PDS) on Ag-MoS₂ (1.1 eV) is higher than those of MoS₂ (0.20 eV) and W-MoS₂ (0.14 eV). This can be due to the fact that a higher applied potential usually reduces the free energy barrier for intermediate conversion. When unified at -0.6 V vs. RHE (Figures S61-63), the PDS Gibbs free energy change of all three catalysts is comparable (Ag-MoS₂: 1.1 eV, MoS₂: 1.08 eV, W-MoS₂: 1.12 eV), suggesting that Ag doping has little contribution to the PDS free energy change.

On the other hand, comprehensive analyses through band structure measurements, interfacial charge regulation studies, NO₃⁻ enrichment quantification, and NO₃RR performance evaluations collectively demonstrate that Ag-MoS₂ in the reaction system establishes solid-liquid (S-L) junction, which induces positive charge accumulation in the inner Helmholtz plane (IHP). This charge redistribution enhances NO₃⁻ anion concentration of Ag-MoS₂ by 8.8 and 27.6 folds to those of MoS₂ and W-MoS₂ (Figure 3g), thereby dramatically promoting electrocatalytic conversion of low-concentration NO₃⁻ to NH₃ (Figure 4a). These findings conclusively establish that the NO₃⁻ enrichment effect serves as the predominant factor in boosting the NO₃RR performance of Ag-MoS₂.

Figure R2. The comparison of calculated adsorption energy of NO_3^- on Ag-MoS₂, MoS₂ and W-MoS₂ at their optimal NO_3^- adsorption potentials.

Figure S61. Gibbs free energy diagram of various intermediates generated during NO_3RR over Ag-MoS₂ at the potential of -0.6 V vs. RHE for pH = 7.

Figure S62. Gibbs free energy diagram of various intermediates generated during NO_3RR over MoS₂ at the potential of -0.6 V vs. RHE for pH = 7.

Figure S63. Gibbs free energy diagram of various intermediates generated during NO_3RR over W-MoS_2 at the potential of -0.6 V vs. RHE for $\text{pH} = 7$.

Figure S64. Gibbs free energy diagram of various intermediates generated during NO_3RR over MoS_2 at the potential of -0.8 V vs. RHE for $\text{pH} = 7$.

Figure S65. Gibbs free energy diagram of various intermediates generated during NO_3RR over W-MoS_2 at the potential of -1.0 V vs. RHE for $\text{pH} = 7$.

Corresponding revision:

- Revised supplementary information (Pages 70-72, highlighted in bright yellow): Figures 61-63 have been revised and updated.
- Revised supplementary information (Pages 73-74, highlighted in bright yellow): Figures 64-65 have been added.
- Revised Manuscript (Pages 19-20, highlighted in bright yellow): “The Gibbs free energy change (ΔG) of this step was 1.1, 0.95, and 1.02 eV for Ag-MoS₂, MoS₂, and W-MoS₂ catalysts at -0.6 V *versus* RHE. The ΔG of Ag-MoS₂ was comparable to those of W-MoS₂ and MoS₂, suggesting that Ag doping had little contribution to the ΔG of PDS. Although MoS₂ and W-MoS₂ exhibited reduced ΔG of PDS at corresponding optimal NO₃⁻ adsorption potentials (Figures S64-65), while, their NO₃RR performance were far below Ag-MoS₂ (Figure 4a). These results further indicated the decisive role of Ag doping-induced surface NO₃⁻ enrichment effect in promoting NO₃RR activity in low concentration system.”.

Comment 6: Aside from the few minor issues mentioned above, I believe the authors have responded to the reviewers' comments thoroughly. The additional data provided is reasonable and effective, and after making the minor revisions, this manuscript can be recommended for publication.

Response: We greatly appreciate the constructive comments and positive feedback from the reviewer on this article. We are grateful for your acknowledgment that our revisions have adequately addressed the previous concerns. In response to your comments, we have revised corresponding descriptions and discussions within the article, and solving these questions is helpful to further improve the depth and scientific nature of the paper. It is our sincere hope that the modifications implemented have now met with your full approval. Thank you again for your help and support of our work.

Manuscript number: NCOMMS-24-49780B

Title: Near-Unity Nitrate to Ammonia Conversion via Reactant Enrichment at the Solid-Liquid Interface

A Point-to-Point Response to Reviewer's Comments

Dear Editor and Reviewers,

Thank you for taking time and effort to carefully examine our manuscript. The comments are highly appreciated and helpful to improve this work.

Reviewer #1:

The authors addressed all the questions. It is ready for acceptance now.

Response: Thank you very much for your support and approval of our work. Solving these questions is helpful to improve the depth and scientific nature of the paper.

Reviewer #4:

The authors have done all reviewer's comments. This manuscript can be considered for publication.

Response: Thanks so much for your constructive comments, your comments play a crucial role in improving the quality of our paper. Thank you again for your help and support of our work.